# *Arabidopsis* RETINOBLASTOMA RELATED directly regulates DNA damage responses through functions beyond cell cycle control

Beatrix M Horvath[1,2,*,†] iD, Hana Kourova[3,†], Szilvia Nagy[4], Edit Nemeth[1] iD, Zoltan Magyar[5], Csaba Papdi[1] iD, Zaki Ahmad[1] iD, Gabino F Sanchez-Perez[6], Serena Perilli[6], Ikram Blilou[6], Aladár Pettkó-Szandtner[5], Zsuzsanna Darula[7], Tamas Meszaros[4,8], Pavla Binarova[3], Laszlo Bogre[1,‡] & Ben Scheres[2,6,‡,**] iD

## Abstract

The rapidly proliferating cells in plant meristems must be protected from genome damage. Here, we show that the regulatory role of the *Arabidopsis* RETINOBLASTOMA RELATED (RBR) in cell proliferation can be separated from a novel function in safeguarding genome integrity. Upon DNA damage, RBR and its binding partner E2FA are recruited to heterochromatic γH2AX-labelled DNA damage foci in an ATM- and ATR-dependent manner. These γH2AX-labelled DNA lesions are more dispersedly occupied by the conserved repair protein, AtBRCA1, which can also co-localise with RBR foci. RBR and AtBRCA1 physically interact *in vitro* and *in planta*. Genetic interaction between the RBR-silenced *amiRBR* and *Atbrca1* mutants suggests that RBR and AtBRCA1 may function together in maintaining genome integrity. Together with E2FA, RBR is directly involved in the transcriptional DNA damage response as well as in the cell death pathway that is independent of SOG1, the plant functional analogue of p53. Thus, plant homologs and analogues of major mammalian tumour suppressor proteins form a regulatory network that coordinates cell proliferation with cell and genome integrity.

**Keywords** *Arabidopsis*; BRCA1; DNA damage response; E2FA; RETINOBLASTOMA RELATED
**Subject Categories** Cell Cycle; DNA Replication, Repair & Recombination; Plant Biology
**The EMBO Journal (2017) 36: 1261–1278**

See also: **S Biedermann et al** (May 2017)

## Introduction

The continuous post-embryonic growth of plants is supported by rapidly proliferating cells in meristems. Protection against the accumulation of mutations in dividing cells is not only important to maintain cellular functions, but additionally to maintain the source for generative cells throughout plant life (Scheres, 2007; Hu *et al*, 2016). The *Arabidopsis* RETINOBLASTOMA RELATED (RBR) is a conserved regulator of cell proliferation, differentiation, and stem cell niche maintenance (Harashima & Sugimoto, 2016). RBR regulates cell proliferation by restraining E2F-dependent transcription of cell cycle genes (Magyar *et al*, 2005; Gutzat *et al*, 2012; Kobayashi *et al*, 2015; Harashima & Sugimoto, 2016). Mitogenic signals promote RBR phosphorylation by cyclin-dependent kinases (CDKs) in association with D-type cyclins, the best characterised being CYCLIN D3;1 (CYCD3;1) (Dewitte *et al*, 2003; Magyar *et al*, 2012). Upon this RBR phosphorylation, the E2FB transcription factor is released and promotes cell cycle gene expression and cell proliferation, while E2FA remains associated with RBR and maintains meristems through repression of differentiation (Harashima *et al*, 2013; Kuwabara & Gruissem, 2014; Magyar *et al*, 2012; Polyn *et al*, 2015). The developmental role of RBR is best understood in the root meristem, where slowly dividing quiescent centre (QC) cells maintain surrounding root stem cells that divide more frequently. The low rate of cell division in the QC protects cells against DNA damage while surrounding stem cells are more sensitive (Fulcher &

1  School of Biological Sciences, Centre for Systems and Synthetic Biology, Royal Holloway, University of London, Egham, UK
2  Department of Molecular Genetics, Utrecht University, Utrecht, The Netherlands
3  Institute of Microbiology CAS, v.v.i., Laboratory of Cell Reproduction, Prague 4, Czech Republic
4  Department of Medical Chemistry, Molecular Biology and Pathobiochemistry, Semmelweis University, Budapest, Hungary
5  Institute of Plant Biology, Biological Research Centre, Szeged, Hungary
6  Department of Plant Sciences, Wageningen University Research Centre, Wageningen, The Netherlands
7  Laboratory of Proteomic Research, Biological Research Centre, Szeged, Hungary
8  Technical Analytical Research Group of HAS, Budapest, Hungary
   *Corresponding author. Tel: +31 317 313734; E-mail: beatrix.horvath@rhul.ac.uk
   **Corresponding author. Tel: +31 317 481 166; E-mail: ben.scheres@wur.nl
   †These authors contributed equally to this work as first authors
   ‡These authors contributed equally to this work as senior authors

Sablowski, 2009; Furukawa *et al*, 2010). RBR, in complex with the transcription factor SCARECROW, was shown to regulate specific stem cell divisions but also impose quiescence, which is important to protect against replication stress-induced cell death (Cruz-Ramirez *et al*, 2012, 2013). RBR is also required during meiosis for chromosome condensation and synapsis of homologous chromosomes, but not for introducing DSBs for homologous recombination (Chen *et al*, 2011).

DNA damaging environmental factors, such as ionising radiation, ultraviolet light, excess of metalloid elements (Br, Al) and internal damage generated spontaneously during DNA metabolism, can all impact on genome integrity (Hoeijmakers, 2009). To counteract the consequences of DNA lesions, organisms have evolved DNA damage response pathways (DDR). The recognition of DNA damage by sensor proteins initiates a network of molecular events that recruit the DNA repair machinery, regulate transcription, control cell cycle progression, eliminate damaged cells by cell death and enter into terminal differentiation or senescence (Su, 2006; Cools & De Veylder, 2009; Ciccia & Elledge, 2010; Sherman *et al*, 2011; Hu *et al*, 2016).

Depending on whether DNA damage results in exposed single-strand (SS) or double-strand breaks (DSBs), different signalling pathways are induced, involving alternative sets of sensors, mediators and effectors (Ciccia & Elledge, 2010). The central components are largely conserved among yeasts, animals and plants, although kingdom-specific proteins are also involved (Harper & Elledge, 2007; Waterworth *et al*, 2011; Amiard *et al*, 2013; Yoshiyama *et al*, 2013b). The conserved DNA damage sensing kinase ATAXIA-TELANGIECTASIA MUTATED (ATM) is activated by double-strand DNA breaks (DSBs) and acts during G1/S and G2/M checkpoints; its role recently was also implicated in the regulation of oxidative stress (Shiloh & Ziv, 2013; Shiloh, 2014). The ATAXIA-TELANGIECTASIA-AND-RAD3 RELATED (ATR) mainly responds to free single-stranded DNA, formed during processing of blocked replication forks, at G1/S and intra-S checkpoints (Culligan *et al*, 2006; Cimprich & Cortez, 2008; Culligan & Britt, 2008; Flynn & Zou, 2011; Amiard *et al*, 2013).

In mammalian systems, the ATM kinase phosphorylates the histone variant H2AX (γH2AX) upon activation by DSB and initiates a cascade of events through recruiting numerous signalling proteins and DNA repair proteins, such as the breast and ovarian cancer type1 susceptibility protein, BRCA1. Single-stranded DNA, the signal for replication stress, is sensed and bound by the mammalian replication protein A (RPA) to form a complex. The resulting complex activates ATR leading to the phosphorylation of the tumour suppressor protein, p53 and delay of S-phase, allowing the recovery of collapsed replication forks (Ciccia & Elledge, 2010).

No direct homologs of p53 have been identified in plants (Yoshiyama *et al*, 2013b), but the plant-specific transcription factor, SUPPRESSOR OF GAMMA RESPONSE 1 (SOG1) is considered to be a functional analogue of p53 (Cimprich & Cortez, 2008; Yoshiyama *et al*, 2013b). SOG1 is directly phosphorylated and activated by ATM. Active SOG1 induces transcription of genes related to DNA damage response and genes that impose cell cycle checkpoint or repair (Culligan *et al*, 2006; Ricaud *et al*, 2007; Yoshiyama *et al*, 2009). Upon DNA damage, ATM and ATR activate the WEE1 kinase, which mainly controls the replication checkpoint (De Schutter *et al*, 2007; Dissmeyer *et al*, 2009; Cools *et al*, 2011). The G2/M DNA damage checkpoint is controlled by the CDKA;1 inhibitors, SIAMESE

RELATED 5 and 7, direct targets of phosphorylated SOG1 upon DNA damage (Yi *et al*, 2014).

Here we show that RBR, besides its well-known function during cell cycle, maintains genome integrity in root meristematic cells. During DNA damage response, RBR together with E2FA accumulates at distinct heterochromatic foci labelled by γH2AX in an ATM/ATR-dependent manner. AtBRCA1 is generally recruited to numerous γH2AX-labelled foci upon damage, but less frequently it also co-localises with RBR. Co-immunoprecipitation and bimolecular fluorescence complementation (BiFC) studies show that these two proteins can interact and genetic data support that they act together in protecting the genome. In addition, RBR/E2FA acts as a transcriptional repressor of *AtBRCA1* transcription in parallel to the SOG1-governed transcription of DDR genes.

# Results

## The role of RETINOBLASTOMA RELATED in mediating maintenance of genome integrity is separable from its function in cell cycle regulation

Reduced RBR levels in the quiescent centre lead to extra cell divisions and sensitivity to genotoxic agents (Cruz-Ramirez *et al*, 2013). To investigate whether the observed cell death was associated with S-phase progression, we quantified DNA synthesis using 5-ethynyl-2′-deoxyuridine (EdU) incorporation and cell death in two Col-0 transgenic lines with reduced RBR levels; the $35S_{pro}$:*amiGORBR* (*amiRBR*) line, in which an artificial miRNA against RBR is expressed constitutively (Cruz-Ramirez *et al*, 2013), and the *RCH1::RBR* RNAi (*rRBr*) line, in which an antisense RNA is expressed locally in the root meristem (Wildwater *et al*, 2005). Both lines conferred similar phenotypes in the root with respect to extra stem cell divisions and increased S-phase labelling (Fig 1A and C for *amiRBR*; Fig EV1A and B for *rRBr*), which correlated with accumulating cell death both in the root tip of *amiRBR* (Fig 1B and D) and in *rRBr* (Fig EV1C).

To investigate whether cell death upon RBR silencing was due to a general deregulation of cell cycle entry, or reflected a specific role of RBR in cell viability, we analysed *CYCD3.1* overexpression, which promotes cell cycle progression through RBR phosphorylation (Dewitte *et al*, 2003, 2007; Magyar *et al*, 2012; Nowack *et al*, 2012) and *E2FA* and *E2FB* overexpression, which act downstream of RBR (De Veylder *et al*, 2002; Magyar *et al*, 2005, 2012). For proper comparison of accessions, the Ler line named G54, overexpressing *CYCD3.1* (Riou-Khamlichi *et al*, 1999; Dewitte *et al*, 2003) was introgressed into Col-0 (Appendix Supplementary Methods). The introgressed line showed increased EdU labelling and cell division compared to Col-0 similar to *amiRBR* (Fig 1A and C). In contrast, no cell death was observed upon *CYCD3.1* overexpression (Fig 1B and D).

Similar to *CYCD3.1*, the overexpression of *E2FA-DPA* (De Veylder *et al*, 2002) or *E2FB-DPA* (Magyar *et al*, 2005; Appendix Supplementary Methods) in transgenic Col-0 lines led to extra cell divisions in the stem cell niche and EdU labelling compared to wild-type controls (Fig EV1D). However, the cell death response remained comparable to Col-0 (Fig EV1E). Our *CYCD3.1* and *E2F* overexpression results indicated that the cell death response is not the consequence of deregulated cell proliferation by the RBR pathway but specifically linked to reduced RBR levels.

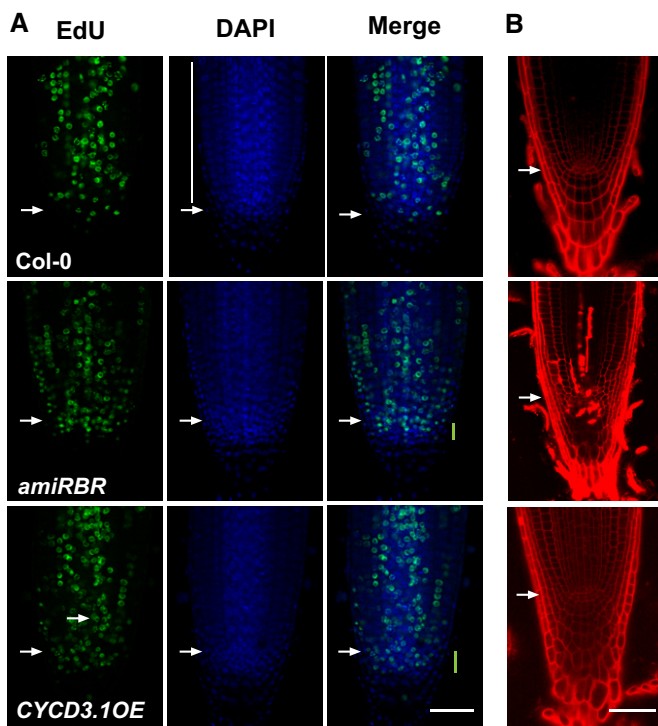

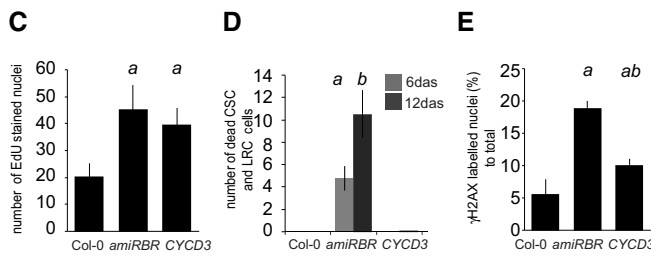

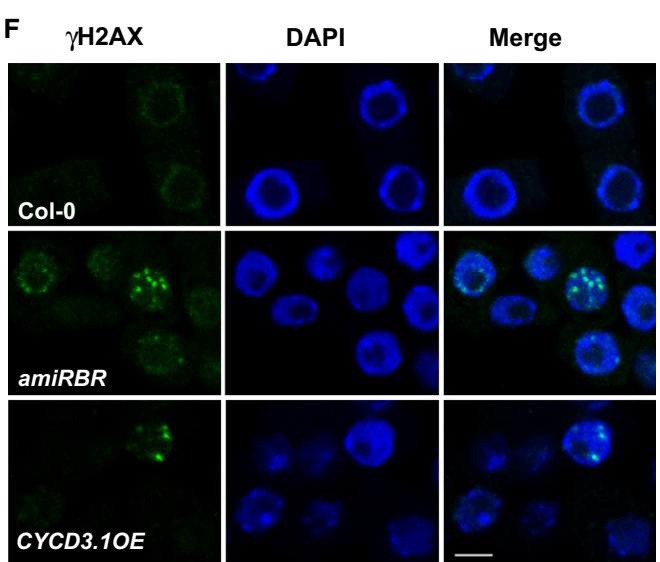

**Figure 1.  Silencing of RBR and *CYCD3.1* overexpression both promote S-phase entry but affect cell death response and DNA damage accumulation differently.**

A  Representative confocal laser scanning microscopy (CM) images of whole mount EdU-labelled roots from 6-day-old (das) seedlings of Col-0, *amiRBR* and Col-0(*CYCD3.1OE*) lines with EdU (green) and DAPI (DNA, blue) staining. In the *amiRBR* and Col-0(*CYCD3.1OE*) lines, the region of extra columella stem cell layers is labelled with a green bar in merged images. White vertical bar shows the region of cells where EdU counting was carried out. Images were taken in single median section, scale bar: 50 μm, arrow: QC position in each image.

B  CM images of propidium iodide (PI)-stained root tips from 12 das seedling; genotypes indicated as in (A). Images were taken in single median section, scale bar: 50 μm, arrow: QC position in each image.

C  Number of EdU-labelled cells as shown in (A) was counted in the epidermis, cortex and endodermis cell layers on both sides of the root. In each case, 10 roots (6 das) were quantified.

D  Cell death response in 6 and 12 das seedlings, total number of dead columella stem cells (CSC) and lateral root cap initials (LRC) and their descendants were counted in median sections as shown in (B), *n* > 2, *N* > 15. Note that in Col-0(*CYCD3.1OE*) only 1–2 dead cells were detected in the analysed population. Quantification of the dead cell area in *amiRBR* is shown in Fig 2C.

E  Frequency of γH2AX-labelled nuclei per total number of DAPI-positive nuclei (%), *n* = 2, *N* > 6 root of 6 das seedlings, analysed nuclei > 1,000.

F  Representative CM images (single section) of γH2AX immune-labelled cells of root tips from Col-0, *amiRBR* and Col-0(*CYCD3.1OE*). DAPI (blue), scale bar: 5 μm.

Data information: Values represent means with standard deviation (SD). In (C–E), *a* indicates significant difference around 1% confidence using Student's *t*-test comparing *amiRBR* and Col-0(*CYCD3.1OE*) to Col-0. In (D), *b* indicates 99% significance (*P* < 0.01) between time points and in (E) *ab* indicates 99% significance (*P* < 0.01) to Col-0 and *amiRBR*. *n* = biological repeat, *N* = sample per biological repeat.

Cell death upon RBR silencing might be a consequence of replication stress-mediated DNA damage. To visualise DNA damage, we followed the accumulation of the phosphorylated H2AX (γH2AX) histone variant. As shown above, the extent of EdU incorporation was comparable between *amiRBR* and Col-0*(CYCD3.1OE)*, but the frequency of nuclei with γH2AX foci was around 4 times higher in *amiRBR* (~19%) and twice as much in Col-0*(CYCD3.1OE)* (~10%) compared to Col-0 (~5.5%; Fig 1E and F). Collectively, our data indicated that increased DNA damage upon reduction in RBR levels is separable from cell cycle regulation and associated with cell death.

Because RBR silencing led to spontaneous DNA damage and cell death, we tested whether the *amiRBR* line showed increased sensitivity to genotoxic stresses conferred by the DNA cross-linker mitomycin (MMC), double-strand break inducer zeocin, and replication stress inducer hydroxyurea (HU) (Hu *et al*, 2016). Cell death response both in *rRBr* and *amiRBR* lines were stronger than in Col-0 upon MMC and zeocin treatments (Fig 2A–C), indicating that genotoxic stress-induced cell death response is suppressed by RBR. In contrast, HU treatment neither triggered cell death in Col-0 nor increased the response in *amiRBR* (Fig 2D). In line with the cell death response, the number of γH2AX-positive nuclei upon MMC treatment increased further in the *amiRBR* line compared to Col-0 (Fig 2E and F).

### DNA stress recruits RBR to γH2AX-labelled heterochromatic foci

The role of RBR in maintaining genome stability and repressing genotoxic stress-induced DNA damage might involve recruitment of

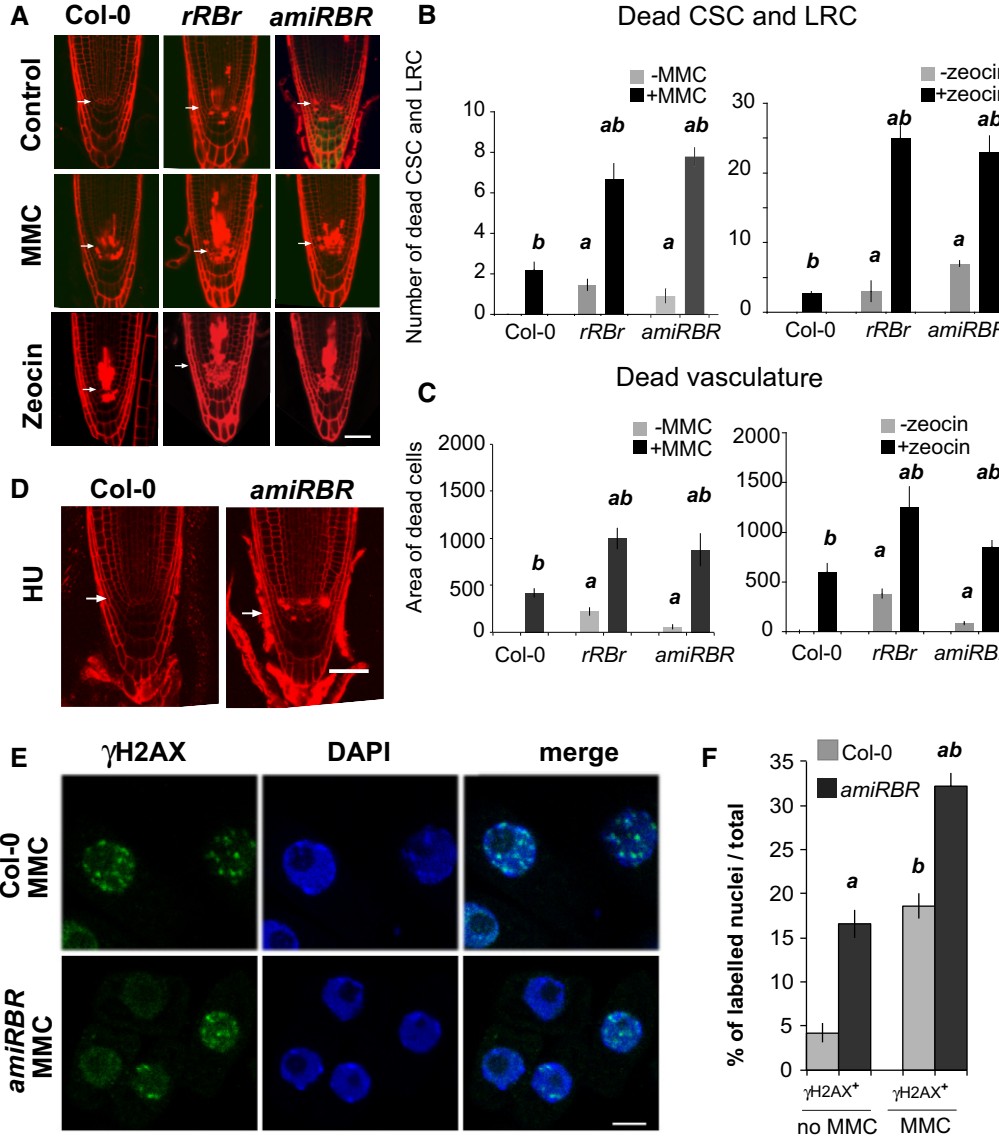

**Figure 2.  Genotoxic stress upon RBR silencing leads to hypersensitive DNA damage response.**

A    Representative (CM) images of Col-0, *rRBr* and *amiRBR* root tips of 6- to 7-day-old seedlings after 16 h of mitomycin (MMC) and 20 h of zeocin treatment compared to non-treated samples (Control).

B, C  Cell death was quantified (B) by the number of the dead columella and lateral root cap stem cells (CSC, LRC) and their daughter cells, and (C) by measuring the area of dead vasculature above the QC in the presence of MMC for 16 h and zeocin for 20 h.

D    Representative (CM) images of Col-0 and *amiRBR* root tips of 6- to 7-day-old seedlings after 16 h of hydroxyurea (HU) treatment compared to non-treated samples (control shown in A).

E    Representative (CM) images of nuclei (single section) of Col-0 and *amiRBR* 6 das root tips after 16 h of MMC treatment immune-labelled for γH2AX (green). DAPI (blue), scale bar: 5 μm.

F    Frequency (%) of γH2AX foci-harbouring nuclei compared to total nuclei in 6 das Col-0 and *amiRBR* root tip after 16 h of MMC treatment compared to non-treated samples.

Data information: In (A and D), arrows indicate position of QC, scale bar: 50 μm. In (B, C and F), values represent mean with standard error, data are combined from *n* = 3 biological repeats, *N* > 15 roots for (B and C) and *N* > 5 in (F) of *amiRBR* and Col-0, total nuclei > 1,000. *a* indicates significant difference within the 5 to 1% statistical confidence interval using Student's *t*-test between *amiRBR* and *rRBr* versus Col-0, and *b* indicates significant difference between treated versus non-treated samples. *n* = biological repeats, *N* = samples per biological repeat.

RBR to DNA damage foci. Without genotoxic stress, RBR is diffusely localised within nuclei (Magyar *et al*, 2012; Fig 3A and C, control). MMC treatment (16 h) induced the accumulation of RBR in typically few large foci (1–5 foci per nucleus, Fig 3A and C, MMC). Around

17% of the examined nuclei contained RBR foci (total number of nuclei, *N* = 845, biological repeat, *n* = 3), mostly co-localised with γH2AX-positive sites (Fig 3A). 3D reconstruction of serial sections revealed a partial co-localisation of RBR and γH2AX foci with a

broad correlation range, which is typical for dynamic and transient protein interactions (Fig 3B and E). A large proportion (80%; $n = 102$) of the analysed RBR foci localised in close vicinity of heterochromatin, as confirmed by intensity profiles (Fig EV2A). The centromeric Histone 3 (CenH3) was also detected together with RBR foci upon MMC treatment (Fig EV2B).

Consistent with the localisation of tobacco E2F in genotoxic stress-induced foci (Lang *et al*, 2012), an E2FA fusion protein under its native promoter (Henriques *et al*, 2010) significantly co-localised with RBR in foci after treatment with genotoxic agents (Fig 3C and E). To test whether RBR and E2FA localisation to these foci depends on DNA damage signalling, we used inhibitors KU55933 for ATM (IATM) and VE-821 for ATR (IATR), which revealed to be effective in plants by additively inhibiting the MMC-induced cell death response (Fig 3F). The simultaneous inhibition of the ATM and ATR kinases by these drugs also abolished both RBR and E2FA focus formation (Fig 3C, IATM and IATR +MMC). In support of a role for both RBR and E2FA on DNA damage sites, RBR-E2FA foci partially co-localised at γH2AX-positive sites (Fig 3G).

### RBR silencing triggers AtBRCA1 recruitment to DNA damage foci and AtBRCA1 co-localises with RBR foci upon DNA stress

BRCA1 is a pivotal DNA repair protein of double-strand DNA damage both in mammals (Rosen, 2013) and in *Arabidopsis* (Block-Schmidt *et al*, 2011; Trapp *et al*, 2011). *Atbrca1-1* (Reidt *et al*, 2006) and *Atbrca1-3* loss of function mutants displayed hypersensitive cell death response to genotoxic stress (MMC treatment) compared to Col-0 (Figs 4A and EV3A–E). We generated a genomic AtBRCA1-GFP construct driven by the endogenous promoter (AtBRCA1-GFP) (Appendix Supplementary Methods) and transformed it into the *Atbrca1-1* line. The AtBRCA1-GFP construct complemented the cell death response of the *Atbrca1-1* mutant (Figs 4A and EV3D and E). In untreated *Atbrca1-1*(AtBRCA1-GFP) root meristems, the GFP

signal was low and diffuse in the nucleus, while the signal increased upon MMC treatment and accumulated in pronounced speckles of an increasing number of meristematic nuclei in *Atbrca1-1*(AtBRCA1-GFP) (Fig 4A). Upon root meristem-specific silencing of RBR in the *rRBr* line, AtBRCA1-GFP also accumulated in nuclear speckles in and around the stem cell niche area indicating that the localisation of AtBRCA1 is induced by RBR reduction and is not critically dependent on RBR (Fig 4B).

AtBRCA1-GFP nuclear speckles co-localised with γH2AX foci, after MMC treatment (Fig 4C), and thus, we investigated whether RBR is co-recruited with AtBRCA1 at γH2AX foci by triple immunocolocalisations of RBR, AtBRCA1-GFP and γH2AX in the *Atbrca1-1* (BRCA1-GFP) line (Fig 4C). Similar proportions of γH2AX-positive nuclei showed co-localisation of γH2AX foci either with AtBRCA1-GFP or RBR (25 and 27%, respectively; Table 1). The AtBRCA1- and γH2AX-overlapping foci were small and numerous in most nuclei and well distinguishable from the large and sparse RBR-γH2AX co-labelled foci. The two different classes of foci rarely coexisted within the same cell (Fig 4C, Table 1). Foci with RBR and AtBRCA1 together at γH2AX sites appeared at lower frequency (10% of the γH2AX[+] nuclei, $N = 452$, $n = 3$; Table 1) and their appearance resembled the large RBR-γH2AX foci. RBR and AtBRCA1 co-localised only in the presence of γH2AX. When ATM and ATR kinase inhibitors were applied simultaneously with MMC, these inhibitors reduced the number of nuclei with γH2AX and AtBRCA1-GFP foci and abolished the formation of RBR foci (Fig 4C).

To test whether RBR can be recruited to γH2AX foci in the absence of AtBRCA1, we monitored RBR and γH2AX foci upon MMC treatment in the *Atbrca1-1* mutant. We observed co-localisation of RBR and γH2AX in large and sparse foci as in the control, suggesting that RBR recruitment is independent of AtBRCA1 (Fig 4D).

To study whether AtBRCA1 and RBR proteins might physically interact, we translated both proteins *in vitro* in wheat germ extract and performed co-immunoprecipitations (Appendix Supplementary Methods). RBR specifically interacted with AtBRCA1, but was

---

**Figure 3.** RBR and E2FA nuclear focus formation depends on ATM/ATR kinases and coincides with γH2AX-positive sites upon MMC and zeocin treatments.

A   Representative CM images (single section) of nuclei with RBR foci at the γH2AX-positive sites in Col-0 upon 16 h of MMC and 3 h of zeocin treatment (white arrowheads); diffuse nuclear RBR signal is shown in the untreated control (RBR: green, γH2AX: red, DAPI: blue).

B   Partial co-localisation of RBR and γH2AX foci shown on Imaris section of the nucleus (RBR: green, γH2AX: red). Main panel (z) shows a single z-stack of the nucleus, right panel (y-z) shows cross section by y plane perpendicular to z plane in the main panel, lower panel (x-z) illustrates cross section by x plane perpendicular to z plane in the main panel. Scale bar: 1 μm; scale bar of magnified insets: 0.5 μm.

C   Representative CM images (single section) of nuclei showing accumulation of RBR (red) and E2FA-GFP (green) in the same nuclear foci (white arrowheads) after 16 h of MMC and 3 h of zeocin treatment (RBR: red, E2FA-GFP: green, DAPI: blue). RBR and E2FA-GFP focus formation was not detected in untreated cells (control) or upon inhibition of ATM and ATR kinases (IATM+IATR+MMC). The activity of IATM and IATR inhibitors was followed on cell death response.

D   Imaris section of a nucleus showing co-localisation of RBR (red) and E2FA-GFP (green) in foci (DAPI: blue). Main panel (z) shows a single z-stack of the nucleus, right panel (y-z) shows cross section by y plane perpendicular to z plane in the main panel, and lower panel (x-z) illustrates cross-section by x plane perpendicular to z plane in the main panel. Scale bar: 1 μm; scale bar of magnified insets: 0.5 μm.

E   The range of Pearson correlation coefficients (PCCs) of RBR/E2FA- and RBR/γH2AX-positive foci formed after 16 h of MMC treatment. PCCs are visualised in quartiles of ranked data ($n = 30$). While RBR/E2FA co-localised in foci with high mean value of PCCs = 0.82, the RBR/γH2AX in foci showed PCCs ranging from 0.1 (side by side co-localisation) to 0.75 (partial co-localisation).

F   The effect of IATM and IATR inhibitors on cell death response upon 16 h of MMC treatment in Col-0 was quantified by the number of columella stem cells (CSC) and lateral cap stem cells (LRC) and their descendants. Values represent mean with standard deviation, $n = 2$, $N > 15$ roots for each, *a* indicates significant difference within the 5 to 1% statistical confidence interval using Student's *t*-test comparing samples treated with inhibitors (single or combined) and MMC to MMC only.

G   Representative CM image (single section) of a nucleus shows localisation of RBR and E2FA to a γH2AX-positive site after 16 h of MMC treatment (white arrowheads, RBR: violet, γH2AX: red, E2FA-GFP: green, DAPI: blue).

Data information: In the intensity profiles (A, C and G), the x-axis shows length in μm measured from 1 and y-axis illustrates relative intensity. Scale bars: 2 μm.

*n* = biological repeats, *N* = samples per biological repeat.

weaker than the positive control, E2FA (Fig 5A). The observed direct interaction between RBR and AtBRCA1 was confirmed by bimolecular fluorescence complementation (BiFC) assays in young, growing tobacco leaves in the presence or absence of MMC. RBR–SCARECROW complex formation (Cruz-Ramirez et al, 2012) served as positive control and AtBRCA1–SCARECROW interaction as

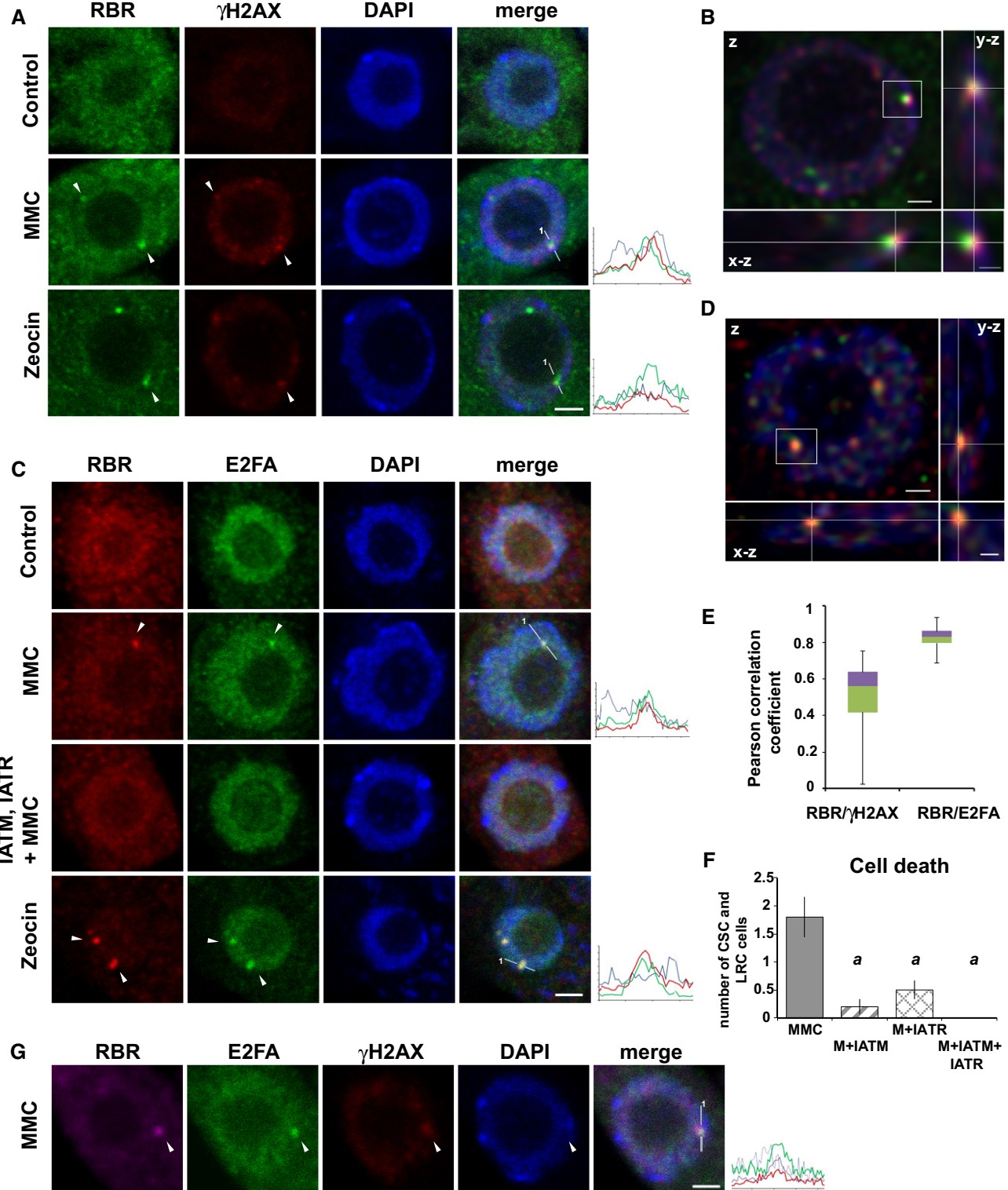

**Figure 3.**

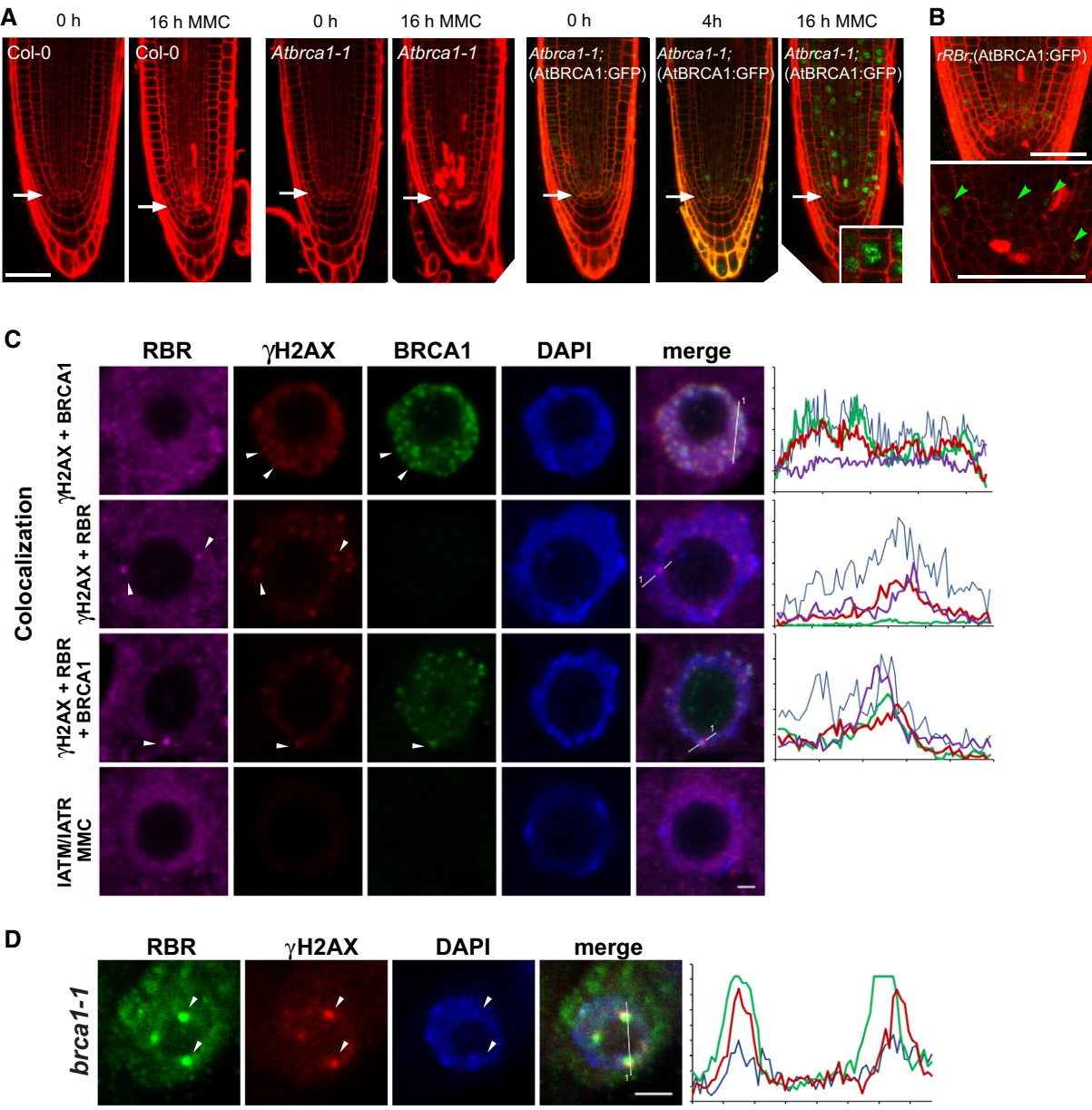

**Figure 4. AtBRCA1 and RBR are recruited to γH2AX foci and partially co-localise upon genotoxic stress, and locate to foci independent of each other.**

A   Representative CM images of PI-stained root tips of Col-0, *Atbrca1-1* (0, 16 h) and *Atbrca1-1*(*AtBRCA1pro:AtBRCA1gen:GFP*) seedlings after 0, 4 and 16 h of MMC treatment. Arrows indicate position of QC, scale bar: 50 μm. Inset in the last image illustrates an enlarged nucleus with pronounced speckles.

B   CM images of PI-stained root tips of *rRBr;*(AtBRCA1:GFP) showing AtBRCA-GFP accumulation into foci in QC and the stem cell niche labelled with green arrowheads. Top and bottom images represent different root tips. Scale bar: 50 μm.

C   Representative CM images of nuclei (single section) with triple immunolabelling for RBR (violet), γH2AX (red) and AtBRCA1 (green) and stained for DAPI (blue) showing co-localisation of AtBRCA1-GFP with γH2AX (arrowheads), RBR with γH2AX (arrowheads) and RBR, γH2AX and BRCA-GFP (arrowheads) after 16 h of MMC treatment. In the presence of ATM and ATR inhibitors (IATM+IATR+MMC), the γH2AX and AtBRCA1-GFP nuclear signals and RBR foci formation were abolished. See also Table 1 for statistics.

D   Representative CM image (single section) of RBR foci localised with γH2AX-positive sites (arrowheads) in nuclei of *Atbrca1-1* root meristematic cells after 16 h of MMC treatment (RBR: green, γH2AX: red, DAPI: blue).

Data information: In (C and D) intensity profiles: *x*-axis shows length in μm measured from 1; *y*-axis shows relative intensity. Scale bars: 2 μm. *N* > 3, *n* = 3.
*n* = biological repeats, *N* = samples per biological repeat.

negative control. RBR and SCARECROW formed a complex within 36 h after infiltration, while RBR and AtBRCA1 complex formation could be detected after 48 h in the nucleus (Fig 5B). In rare cases, the interaction was observed in foci. These data indicated that RBR and AtBRCA1 are independently recruited to DNA damage foci but have the ability to interact.

**Table 1. Number and ratio of nuclei showing co-localisation of γH2AX, RBR and/or AtBRCA1.**

|  | Number of nuclei | | | | | Ratio | |
|---|---|---|---|---|---|---|---|
|  | Root 1 | Root 2 | Root 3 | Mean | SD | Mean (%) | SD |
| γH2AX (total) | 156 | 144 | 152 | 151 | 6.1 | 100% |  |
| γH2AX+ RBR | 37 | 56 | 45 | 46 | 9.5 | 27% | 3% |
| γH2AX+AtBRCA1 | 32 | 42 | 38 | 37 | 5.0 | 25% | 4% |
| γH2AX+AtBRCA1+RBR | 12 | 14 | 15 | 14 | 1.5 | 9% | 1% |

### RBR and AtBRCA1 genetic interaction suggests common roles in maintaining genome integrity

To study whether RBR and AtBRCA1 might function together, we studied their genetic interaction. Based on genotyping and segregation analysis of linked resistance markers, the *amiRBR;Atbrca1-1* cross was homozygous for both loci, yet around half of the seedlings showed strong developmental abnormalities, such as mis-positioned or missing organs or seedling lethality (Fig EV4A and B), indicating a variably penetrant window of sensitivity for the lack of AtBRCA1 and compromised RBR level during embryogenesis. The *amiRBR; Atbrca1-1* seedlings that looked largely normal displayed extra stem cell divisions and increased S-phase entry in the root meristem, both phenotypic confirmations for the effective RBR silencing (Fig EV4D–G), and the *AtBRCA1* expression could not be induced in the introgressed line confirming the presence of the mutant *AtBRCA1-1* allele (Fig EV4C). The frequency of γH2AX-positive nuclei in the *Atbrca1-1* and *amiRBR* parents and the *amiRBR;Atbrca1-1* cross was similar (Fig 6A and B), which is consistent with a scenario where RBR and AtBRCA1 act together in a common pathway to maintain genome integrity.

We also studied whether AtBRCA1 function is required for the cell death response observed in the *amiRBR*, and found that both in the *amiRBR,Atbrca1-1* and *amiRBR;Atbrca1-3* crosses, the cell death was substantially suppressed (Fig 6C), as quantified in the distal stem cell niche (Fig 6D). The lack of AtBRCA1 function had no substantial influence on other RBR-regulated processes such as columella stem cell division or S-phase entry (Fig EV4D–G).

To test whether *AtBRCA1* expression is sufficient to induce cell death, we expressed a *myc*-tagged genomic *AtBRCA1* fusion under the control of the GVX β-estradiol-inducible promoter (*GVX1090$_{pro}$: AtBRCA1$_{gen}$:10xmyc*) in the *Atbrca1-1* mutant. After 24 h of induction, no cell death developed, indicating that elevation of *AtBRCA1* transcription cannot trigger cell death on its own (Appendix Fig S1 and Appendix Supplementary Methods). These observations indicate that AtBRCA1 is required but not sufficient to trigger a cell death response when RBR cannot maintain genome integrity.

### RBR regulates DDR gene transcription through E2FA

The observed recruitment of RBR together with E2FA as a complex at DNA lesions might start the signalling process for the

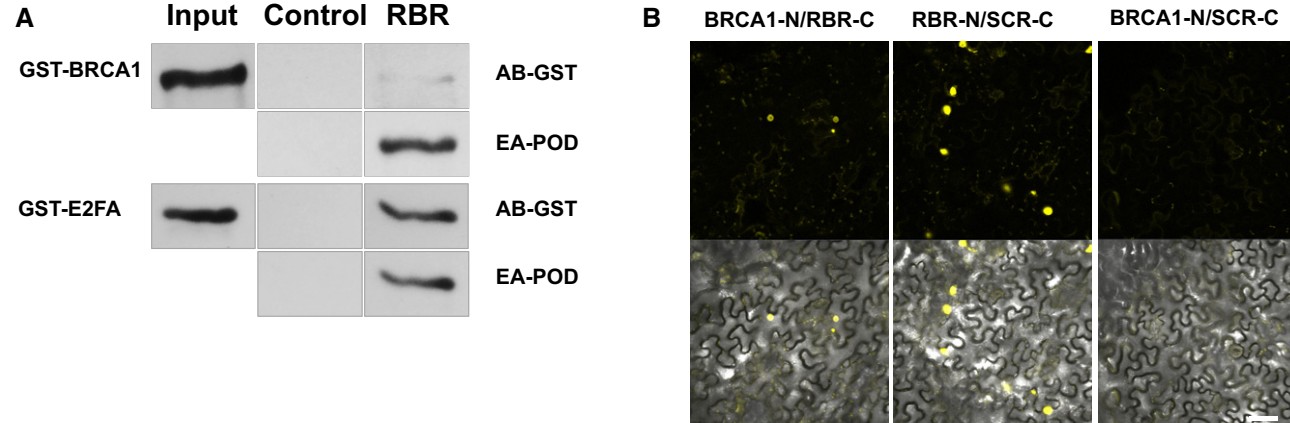

**Figure 5. RBR and AtBRCA1 proteins can physically interact.**

A Co-immunoprecipitation of RBR with AtBRCA1 and E2FA proteins. Control: streptavidin beads, RBR: streptavidin beads bound with RBR-biotin, AB-GST: GST (anti-glutathione-S-transferase) antibody, EA-POD: Extravidin-POD (peroxidase-conjugated streptavidin) labelling RBR-biotin-containing complexes, GST-BRCA1: GST-labelled AtBRCA1, and GST-E2FA: GST-labelled E2FA proteins, in the input of the wheat germ extract.

B BiFC assay *in planta* reveals physical interaction between AtBRCA1 and RBR (BRCA1-N/RBR-C). The RBR-N/SCR-C pair was used as a positive control, and BRCA1-N/ SCR-C pair as a negative control. Young, growing tobacco leaves were infiltrated and analysed 36–48 h after infiltration. Scale bar: 50 μm, SCR: SCARECROW transcription factor.

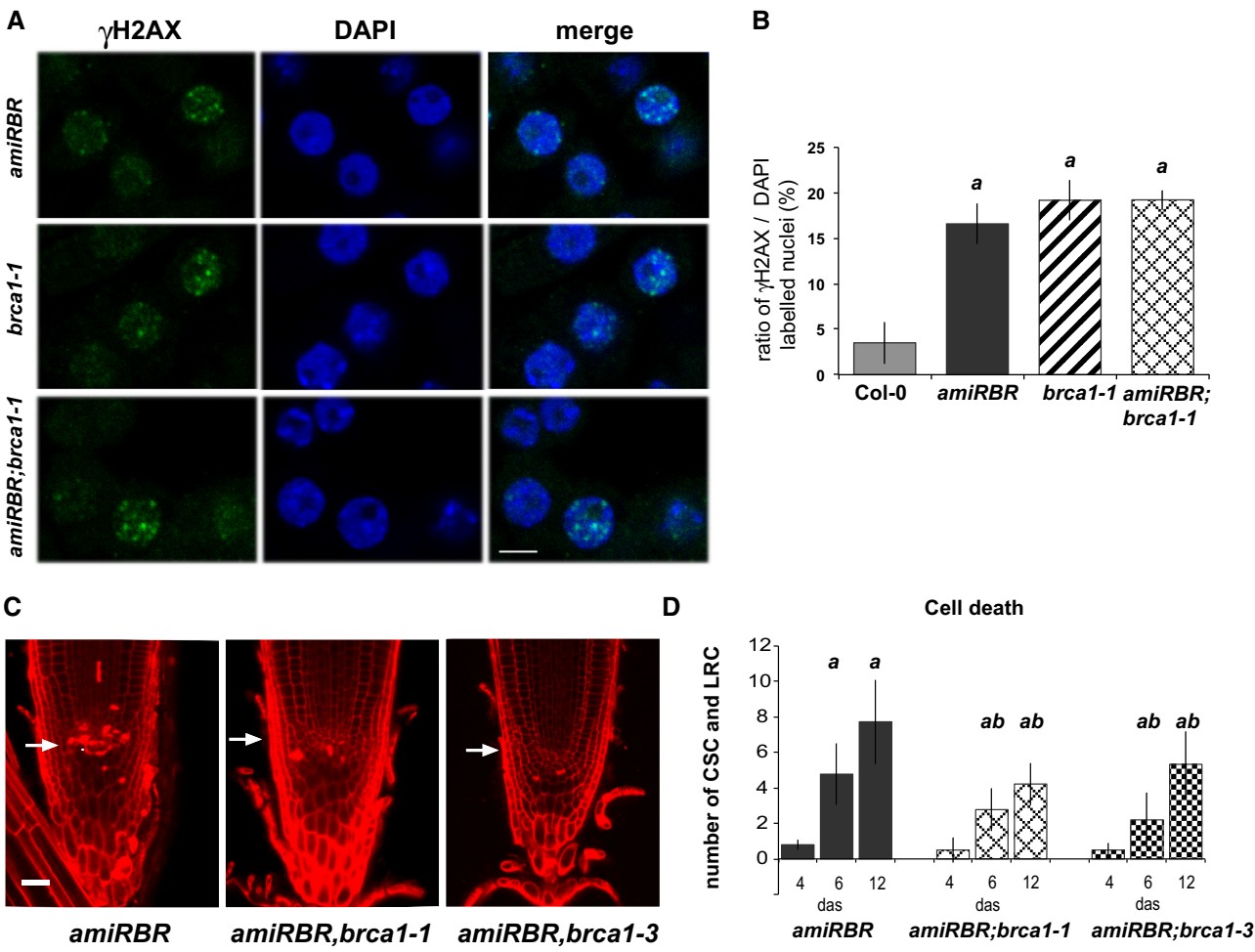

**Figure 6.   RBR and AtBRCA1 may act in a common process during DDR.**

A   Representative (CM) images of nuclei (single section) of *amiRBR*, *Atbrca1-1* and *amiRBR;Atbrca1-1* 6 das root tips immunolabelled for γH2AX (green) and DAPI (blue). Scale bar: 5 μm.

B   Frequency (%) of γH2AX-labelled nuclei to total DAPI-stained nuclei in Col-0, *amiRBR*, *Atbrca1-1*, *amiRBR;Atbrca1-1* grown under normal conditions. Values represent means with SD, *n* = 3, and total nuclei > 1,000. *a* indicates significant difference within the 1% statistical confidence interval using Student's *t*-test between *amiRBR*, *Atbrca1-1, amiRBR;Atbrca1-1* versus Col-0.

C   CM images of PI-stained root tips from *amiRBR*, *amiRBR;brca1-1* and *amiRBR;brca1-3* of 12 das seedlings. Scale bar: 20 μm, arrow: QC position in each image.

D   Cell death response of *amiRBR*, *amiRBR;brca1-1*, *amiRBR;brca1-3* seedlings at 4, 6 and 12 das. Values represent means with SD, *N* > 15 for each mutant and Col-0 (*n* = 3–4) *a*: *P* < 0.01 between the given genotype and *Atbrca1-1*, which did not develop cell death at any time point. *b*: *P* < 0.01 comparison between cross and *amiRBR*. The total number of dead columella stem and daughter cells (CSC), lateral root cap initials and their descendants (LRC) were counted in median sections as shown in (C).

Data information: *n* = biological repeats, *N* = samples per biological repeat.

transcriptional regulation of DNA damage response genes. To investigate transcriptional responses to RBR down-regulation in the root tip, we performed genomewide transcriptome profiling of the meristematic region (representative root tips of each time point are shown in Fig EV1A) in three independent biological replicates (Appendix Supplementary Methods). We identified 99 differentially expressed genes between *rRBr* and Col-0 root tips, of which 82 genes were up- and 17, including *RBR,* were down-regulated (Appendix Table S1). Gene ontology (GO) analysis revealed significant enrichment for genes encoding nuclear proteins functionally related to three major processes: (i) nucleosome and chromosome assembly and maintenance; (ii) replication and cell cycle checkpoint control; (iii) DNA damage response and repair (Fig 7A, Appendix Table S1 and Appendix Supplementary Methods). The transcriptional changes in a set of genes representing different functional and co-expressional categories (Appendix Table S2, Appendix Fig S2 and Appendix Supplementary Methods) were confirmed by qRT–PCR both in *rRBr* root tips, where *RBR* is silenced in root meristems (Fig 7B and C), and in seedlings from *amiRBR* where post-embryonic RBR levels are reduced constitutively using the *35S* promoter (Fig 7D). The transcriptional changes were comparable in *rRBr* and *amiRBR* lines and in full agreement with the micro-array data. Importantly, *AtBRCA1* was among the DDR targets that were up-regulated upon RBR silencing.

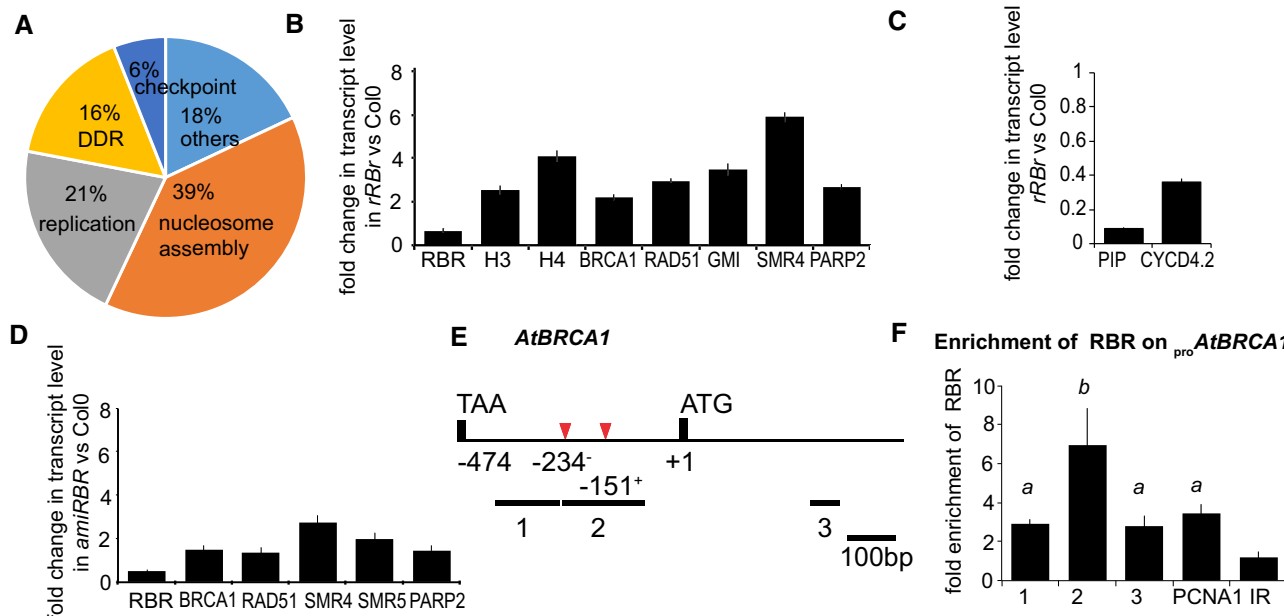

**Figure 7. Genes regulated by RBR are annotated to nucleosome assembly, replication and DDR; RBR protein is enriched on the *AtBRCA1* promoter.**

A   The pie chart represents the major processes regulated by RBR.

B, C   Validation of transcriptome analysis for a selected set of up- and down-regulated genes upon RBR silencing, respectively, using qRT–PCR on dissected root tips of 4-day-old *rRBr* and Col-0 seedlings.

D   Genes showing differential expression upon local RBR silencing are also de-repressed in the constitutively silenced *amiRBR* line. Graph represents qRT–PCR on 4-day-old seedling material.

E   Schematic representation of the *AtBRCA1* promoter; black lines with numbers indicate the position and length of the amplified regions by qPCR analysis, the position of the start codon (ATG), the stop codon of the upstream neighbouring transcript and the position of putative E2F elements (red arrowheads) on the + and − strand, at positions −234 and −151, respectively, are indicated. Position of amplified regions: 1: −383 to −248; 2: −238 to −78; and 3: +313 to +455; positions are numbered from ATG (+1).

F   Chromatin immunoprecipitation (ChIP) using RBR antibody; the graph shows fold enrichment calculated as a ratio of chromatin bound to the numbered section of the promoter with or without antibody. Values represent mean of three biological replicates with standard error, a: $P < 0.01$ compared to the negative control and b: $P < 0.01$ compared to the positive control using Student's *t*-test. *PCNA1* promoter was used as a positive control and IR (an intergenic region between At3g03360-70) as a negative control. The enrichment on IR was arbitrarily set to 1. Numbers 1, 2 and 3 on the *x*-axis refer to the regions labelled in (E).

Data information: In (B–D), values represent mean of fold change normalised to values of the relevant genes from Col-0, and error bars indicate ± SD, $n = 2$, $N > 100$. All of the values were in the 1% statistical confidence interval using Student's *t*-test. Abbreviations of genes are available in Appendix Table S1 and primers used in this study in Appendix Table S3. Data information: $n$ = biological repeats, $N$ = samples per biological repeat.

The presence of canonical E2F binding sites in the 1-kb promoter region of the differentially expressed genes (53 out of 99, summarised in Appendix Table S1 column *N*, based on Naouar *et al*, 2009) suggested that RBR exerts its repressive activity through E2F proteins. Chromatin immunoprecipitation (ChIP) assays were carried out on root tissues using RBR-specific antibody (Horvath *et al*, 2006) in three independent biological repeats followed by qRT–PCR on the promoter of the *AtBRCA1* (Fig 7F). Significant enrichment of RBR was detected on distinct regions of the *AtBRCA1* promoter. The enrichment on fragment 2 that contained two putative E2F binding motifs (Fig 7E; $−234^+$:ggggcaa and $−151^−$:tttg-gcgc) exceeded the enrichment detected on the *PCNA1* promoter used as a positive control (Fig 7F). A reduced level of enrichment (± 3 times) was also observed in neighbouring regions lacking putative binding sites, which may be attributed either to the heterogeneous size of sonicated fragments (± 300–500 bp) or to E2F binding to non-consensus sequences.

To address which of the activator E2Fs might partner with RBR to regulate DDR gene expression, we quantified transcription of *AtBRCA1* in *e2fa-1*, *e2fa-2*, *e2fb-1* (MPIZ_244, GABI-348E09, SALK_103138, respectively; Berckmans *et al*, 2011b) and *e2fb-2* (SALK_120959) mutants. Similar to *amiRBR*, *AtBRCA1* expression increased in *e2fa-1* but not in *e2fa-2* mutants nor in any of the *e2fb* mutants when compared to Col-0 (Fig 8A). The difference in the *AtBRCA1* expression in the two *e2fa* lines likely relates to the different sites of insertion in the two alleles. Both *e2fa* mutant alleles are predicted to encode truncated E2FA proteins that lack the trans-activation and the canonical RBR binding domains, but retain the DNA-binding and dimerisation domains. In contrast to *e2fa-2*, the *e2fa-1* allele also lost the putative "marked box" domain (Fig EV5A), which was described in mammalian E2Fs to provide a second interaction interface with Rb's C-terminal domain (Ianari *et al*, 2009; Dick & Rubin, 2013). *AtBRCA1* derepression in *amiRBR; e2fa-1* and *amiRBR;e2fa-2* double homozygous lines did not exceed the derepression seen in *amiRBR*, further validating that RBR represses *AtBRCA1* through the DNA-binding E2FA transcription factor (Fig 8A). The level of RBR silencing in the double mutants is shown in Fig EV5B. Among the RBR-repressed DDR-related genes

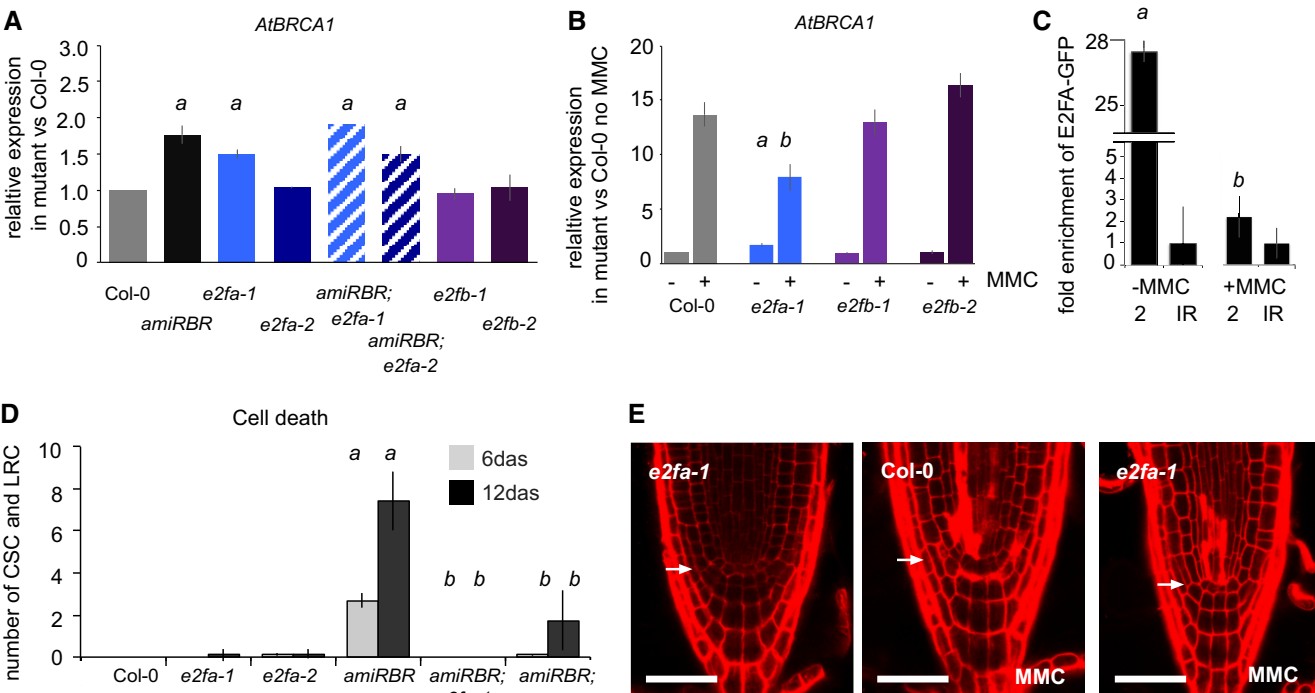

**Figure 8. Spontaneous cell death upon RBR silencing is suppressed by E2FA and DNA damage response upon genotoxic stress is dependent on E2FA.**

A  Relative transcript level of *AtBRCA1* in *amiRBR*, *e2fa-1*, *e2fa-2*, their double mutants and *e2fb-1*, *e2fb-2* compared to Col-0, where the level of expression was set arbitrarily to 1.

B  Relative transcript level of *AtBRCA1* in Col-0, *e2fa-1*, *e2fb-1* and *e2fb-2* without and upon 16 h of MMC treatment. All the values are compared to the expression level measured in non-induced Col-0 which was set to 1.

C  ChIP using GFP antibody to chromatin isolated from Col-0(AtE2FA-GFP) seedlings; the graph shows fold enrichment on the *AtBRCA1* promoter region 2 without and upon genotoxic treatment (MMC, 16 h). The graph illustrates a representative experiment. *a*: P < 0.01 without MMC, *b*: P < 0.01 in MMC compared to the non-treated and IR control using Student's *t*-test. The enrichment on IR was arbitrarily set to 1.

D  Quantitative analysis of cell death response in Col-0, *e2fa-1*, *e2fa-2*, *amiRBR*, *amiRBR;e2fa-1* and *amiRBR;e2fa-2* mutants at 6 and 12 das. Values represent mean ± SD, at least two biological replicates testing more than 20 seedlings for each mutant. Note the absence and insignificant number of spontaneous cell death in the distal stem cell niche in Col-0 and *e2fa* mutants, respectively, at these time points. *a*: P < 0.05 significance comparing single mutant to Col-0 and *b*: P < 0.05 comparing double mutants to *amiRBR* using Student's *t*-test. CSC: columella stem cells, LRC: lateral root cap initials and their descendents.

E  CM images of PI-stained root tips in non-treated *e2fa-1* mutant, and MMC-treated Col-0 and *e2fa-1* (6 das). Images were taken in median section, scale bar: 50 μm. Arrow: QC position in each image.

Data information: In (A and B), values represent mean ± SD, n > 2, N > 100 in each experiment. *a*: P < 0.05 comparing single mutant to Col-0 and in (B) *b*: P < 0.05 comparing values upon MMC treatment using Student's *t*-test. n = biological repeats, N = samples per biological repeat.

tested, only *SMR4* was similarly regulated as *AtBRCA1* by RBR and E2FA (Fig EV5E). Interestingly, MMC-induced *AtBRCA1* expression was suppressed in *e2fa-1* but not in *e2fb* mutants (Fig 8B), suggesting that E2FA is specifically required for genotoxic stress-induced *AtBRCA1* expression.

**RBR represses E2FA activity to inhibit cell death response**

To test whether E2FA can directly bind to the *AtBRCA1* promoter, we performed ChIP analysis using *AtE2FA_pro_:AtE2FA_gen_:GFP* seedlings and *35S_pro_:GFP* controls (Magyar *et al*, 2012). E2FA-GFP was highly enriched on the segment of the *AtBRCA1* promoter containing two putative E2F binding sites (Fig 8C) and the enrichment was reduced when seedlings were treated with MMC (Fig 8C), indicating that, upon genotoxic stress, RBR-E2FA-mediated repression of *AtBRCA1* is released. To investigate whether the release may rely on a change in E2FA-RBR interaction upon genotoxic stress, we pulled

down the complex through the E2FA-GFP and quantified known complex components by label-free mass spectrometry (MS). We found that the association of RBR with E2FA and the DPs became stronger upon MMC treatment as indicated by the ratio of the quantified MS spectra of the complex components (Appendix Table S4). Interestingly, in the E2FA-GFP pull downs we could never detect any of the components of the multi-protein complex *DP*, *RB-like E2F* and *MuvB* (DREAM, Sadasivam & DeCaprio, 2013), while with E2FB-GFP, these proteins were readily pulled down (Appendix Table S5). This may suggest that E2FA functions in different complex(es) than the DREAM associated with E2FB and E2FC (Kobayashi *et al*, 2015).

As RBR repression acts through E2FA to regulate transcription of at least two DDR genes, we investigated whether this regulation functions also in the cell death response. We quantified cell death in the two *e2fa* mutant lines alone and in combination with *amiRBR*. Neither *e2fa-1* nor *e2fa-2* showed any cell death response, and root

development was also normal. Importantly, spontaneous cell death in *amiRBR* was completely suppressed in the *amiRBR;e2fa-1* and strongly delayed and reduced in the *amiRBR;e2fa-2* crosses (Fig 8D) while RBR silencing remained effective (Fig EV5B), demonstrating that the RBR silencing-induced cell death response is dependent on E2FA function. To test whether E2FA is also required for genotoxic stress-induced cell death, we treated *e2fa* mutants with MMC. Cell death upon genotoxic stress was partially suppressed in *e2fa-1* and *e2fa-3* (Xiong *et al*, 2013) lines, but not by *e2fa-2* (Fig 8E and quantified in Fig EV5C and D), confirming that the cell death is generally dependent on E2FA and is mediated through the marked box.

### E2FA and RBR are required for genotoxic stress-induced DDR in a SOG1-independent pathway

SOG1 is a pivotal transcription factor for the induction of DDR genes upon genotoxic stress. We observed significant overlap between DNA repair genes regulated by RBR (Appendix Table S1 column B) and genes with compromised induction by irradiation in the *sog1-1* mutant (Appendix Table S1 columns B and L, respectively; ratio: 8/10). The *sog1-1* mutation can fully suppress cell death response upon genotoxic stress (Yoshiyama *et al*, 2009, 2013a). Based on this

comparison, we asked whether the activation of the DNA damage response pathway upon RBR silencing is dependent on SOG1 function. Homozygous *sog1-1* plants (Preuss & Britt, 2003) were transformed with the $35S_{pro}:amiGORBR$ construct (Cruz-Ramirez *et al*, 2013), and RBR silencing was confirmed in the *amiRBR,sog1-1* line (Fig EV6C). The cell death response in the *amiRBR,sog1-1* root meristem was comparable to the *amiRBR* line (Fig 9A and B), demonstrating that cell death induced upon RBR silencing is independent of SOG1. In the *amiRBR,sog1-1* lines, RBR silencing was effective (Fig EV6C) and transcription of all the tested DDR genes also remained elevated as in *amiRBR* (Fig 9C), showing that the release of RBR-mediated transcriptional repression is also SOG1 independent. As expected, the genotoxic stress-induced DDR gene expression (Fig 9D) and cell death response by MMC (Fig EV6A and D) and zeocin (Fig EV6B) treatments were fully suppressed in the *sog1-1* plants, but not in the *amiRBR,sog1-1* lines, further confirming that RBR acts on a SOG1-independent pathway. HU does not activate SOG1 to induce DDR (Yoshiyama *et al*, 2013a), and hence accordingly did not have any effect in the *amiRBR* and *amiRBR, sog1-1* lines (Fig EV6B). Taken together, RBR regulates DDR gene transcription and cell death at least in part through a SOG1-independent pathway.

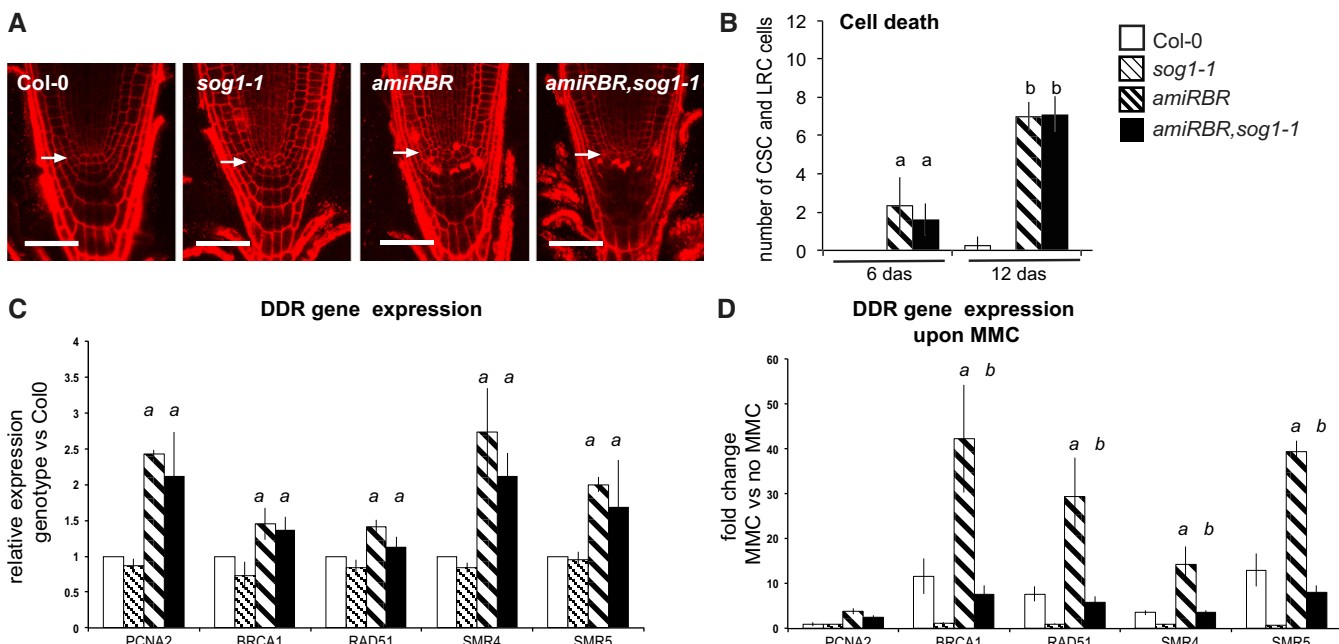

**Figure 9. DNA damage response upon RBR silencing is independent of SOG1.**

A  CM images of PI-stained root tips (12 das seedlings) from the genotypes indicated. Arrow indicates the position of the QC; scale bar: 50 μm.

B  Cell death response from 6 and 12 das seedlings, total number of dead columella stem cells (CSC), lateral root cap initials (LRC) and their descendants were counted as shown in (A). Note that neither Col-0 at 6 das nor *sog1-1* at 6 and 12 das showed cell death. *a*: P < 0.05 at 6 das, *b*: P < 0.05 at 12 das mutants versus Col-0 using Student's *t*-test. Value represents mean ± SD, n > 3, N > 15 for each mutant and Col-0. The genotype legend in the graph also holds for (C and D).

C  Relative expression level of DDR genes in *sog1-1*, *amiRBR* and *amiRBR,sog1-1* lines (6 das) compared to Col-0 (6 das), where the level of expression for each gene was set arbitrarily to 1. *a*: P < 0.05, mutants compared to Col-0 using Student's *t*-test.

D  Transcriptional induction of the indicated genes is depicted as fold change comparing MMC (16 h) to non-treated samples of Col-0, *sog1-1*, *amiRBR* and *amiRBR,sog1-1*. *a*: P < 0.05 *amiRBR* versus Col-0 and *b*: P < 0.05 *amiRBR,sog1-1* to *sog1-1* using Student's *t*-test.

Data Information: In (C) and (D), 6 das seedlings were analysed and data represent means with ± SD. At least three biological replicates were analysed, in each case around 100 seedlings for each mutant. For *amiRBR,sog1-1* the mean was calculated from the analysis of six independent transformants (T2 generation). n = biological repeats, N = samples per biological repeat.

# Discussion

Here we show that the plant Retinoblastoma homologue, RBR, has a direct role in maintaining genome integrity. RBR is recruited to a limited number of large heterochromatic DNA damage foci together with E2FA in an ATM- and ATR-dependent manner upon DNA damage. AtBRCA1 and RBR are independently recruited to these specific damage foci, they interact and our genetic study show that they partially act together to maintain genome integrity and to prevent cell death. The accompanying paper of Biedermann *et al* (2017) shows that RBR is required to localise RAD51 and that RBR and RAD51 co-localise in these large foci, corroborating a non-transcriptional role for RBR in the maintenance of plant genome integrity. We further show that RBR and E2FA also have transcriptional DNA damage response roles that act in parallel to the well-established SOG1 pathway. Below, we discuss how this evidence for a possible dual cell cycle-independent role of RBR in the meristematic DNA damage response at γH2AX foci and at target gene promoters fits in with recent evidence from plant and mammalian experimental systems.

## RBR-mediated DNA damage control at γH2AX foci

When mammalian Rb proteins are deregulated, aberrant S-phase progression can result in nucleotide pool deficiency, replication fork stalling and DNA damage (Bester *et al*, 2011). Even with haploid wild-type Rb, mammalian cells show chromosome defects and aneuploidy (Coschi *et al*, 2014; Hinds, 2014). Similarly, in plants, reduction of RBR function in the *rbr-3* mutant (Johnston *et al*, 2010) and in transgenic plants overexpressing the viral RepA protein inactivating RBR (Henriques *et al*, 2010) resulted in aneuploidy. Our data corroborate that, in plants, S-phase progression due to RBR deregulation can contribute to DNA damage but we show that this effect is separable from a direct role of RBR in DNA damage control.

Our finding that DNA damage induces ATM/ATR-dependent recruitment of RBR and E2FA to γH2AX foci suggests a direct, non-transcriptional role for RBR in DNA damage control. The reported accumulation of *Nt*E2F in γH2AX-labelled foci at the G1/S transition in tobacco cells is consistent with this notion (Lang *et al*, 2012). Also, consistent with non-transcriptional roles for RBR is the finding that, during meiosis, RBR is recruited to chromosomes in a DNA DSB-dependent manner, where it was suggested to facilitate the assembly of chromatin modifiers, repair proteins and condensin complexes for homologous recombination through their LxCxE motifs (Chen *et al*, 2011).

In mammals, Rb localises to chromatin at S-phase after DNA damage (Avni *et al*, 2003). Furthermore, E2F1 (Coschi *et al*, 2014) and E2F7 (Zalmas *et al*, 2013) have transcription-independent roles to bring protein complexes to damaged DNA. E2F1 and Condensin II are recruited by pRb to the pericentromeric region of the chromosome and to replication origins, thus facilitating correct replication, accurate chromosome condensation, and chromosome segregation (Coschi *et al*, 2014; Hinds, 2014). Rb heterozygosity leads to loss of E2F1 and Condensin II binding, accompanied by replication stress labelled by increased γH2AX foci. Recently, it was shown that Rb localises to DSBs dependent on E2F1 and ATM, to promote DSB repair through homologous recombination (Velez-Cruz *et al*, 2016). We show that a similar mechanism might operate in plants, where

RBR localisation to DSBs requires ATM and ATR activities. Further similarities to the animal scenario are that, homogenously distributed tobacco E2F partly relocalises upon genotoxic stress and forms 2–3 foci per nucleus in BY-2 tobacco cells. For this focus formation, the transactivation domain and the RBR binding site were shown to be critical. Also, the plant Condensin complex II appears to play a role in alleviating DNA damage by HR or compacting the genome in response to genomic stress (Sakamoto *et al*, 2011). In future, it will be interesting to investigate whether a similar non-transcriptional role for RBR, E2FA and Condensin II complexes in genome integrity is also operational in plants.

In mammalian cells, Rb interacts with HsBRCA1, which was suggested to be important to repress cell proliferation (Aprelikova *et al*, 1999). In *Arabidopsis,* we did not detect any cell proliferation effect either after induced *AtBRCA1* overexpression or in the *Atbrca1* mutants. In human cells, Rb was also shown to recruit HsBRCA1 in order to facilitate processing and repair of topoisomerase II-induced DSB (Xiao & Goodrich, 2005). Recently, Rb was also shown to be directly involved in DSB repair, independently of its cell cycle function, through its interaction with components of the canonical non-homologous end-joining repair pathway (Cook *et al*, 2015). It will be interesting to investigate the mechanism of RBR and AtBRCA1 interaction at these specific heterochromatic sites with damaged DNA, and their joint function in DNA damage control in *Arabidopsis*.

## RBR-mediated transcriptional responses to DNA damage

Cells with excessive damage are eliminated. The coordination of cell proliferation and apoptosis in mammalian cells relies on the formation of the Rb-E2F1 complex by interaction of Rb's carboxy-terminal domain and the marked box of E2F1 (Carnevale *et al*, 2012; Dick & Rubin, 2013). During S-phase, the phosphorylated Rb-E2F1 complex represses pro-apoptotic genes, while in response to DNA damage upon ATM-dependent phosphorylation of E2F1, this complex becomes a transcriptional repressor on the cell cycle genes and activator on the pro-apoptotic genes (Ianari *et al*, 2009; Dick & Rubin, 2013). There are indications that a similar mechanism may function in *Arabidopsis*. RBR forms a complex with E2FA, which remains stable upon CYCD3;1-CDKA phosphorylation during the cell cycle (Magyar *et al*, 2012). In animal cells, phosphorylation of Rb by CycD:Cdk4/6 kinases diversifies rather than merely inactivates Rb complexes (Narasimha *et al*, 2014). RBR phosphorylation upon CYCD3.1 overexpression in plants might similarly lead to the formation of distinct regulatory complexes with roles in activation of G1 to S transition and roles protecting against cell death or differentiation. In agreement, we find that silencing of RBR leads to a very different outcome than RBR phosphorylation. It initiates cell death response fully relying on E2FA with an intact "marked box" domain, suggesting a conserved mechanism between kingdoms. Importantly, not all the RBR-repressed DDR genes are E2FA regulated. AtBRCA1 is an essential target, as its function was required but not sufficient to induce cell death upon transcriptional derepression. As cell death response was fully suppressed in *e2fa-1*, in relation to AtBRCA1, additional genes should be involved in the induction of cell death process. Interestingly, both AtBRCA1 and E2F functions are required also for the pathogen-induced cell death during hypersensitive response in plant defence (Bao & Hua, 2015; Zebell & Dong, 2015).

Active SOG1 is the pivotal transcription factor in plant DDR upon genotoxic stress (Yoshiyama *et al*, 2013b). Here we show that E2FA also carries out this function, since MMC-induced activation of DDR genes, such as *AtBRCA1* and *SMR4*, is compromised both in *sog1-1* and *e2fa-1* mutants. The ability of E2FA to activate DDR genes is dependent on RBR levels or activity, which are responsive to intrinsic cell cycle-dependent and cell extrinsic signals in a SOG1-independent pathway.

In conclusion, RBR, mainly known as a regulator of cell cycle and asymmetric cell division in plant meristems, is also involved in maintaining genome integrity in these growth zones through two functions, (i) assembly at a limited number of γH2AX foci together with E2F and, possibly, AtBRCA1; (ii) transcriptional regulation of important DDR genes including *AtBRCA1*. It will be interesting to investigate in the future whether and how assembly of E2F-RBR complexes at particular γH2AX foci is coupled to the transcriptional role of these complexes in DDR gene regulation.

# Materials and Methods

## Plant material and growth conditions

Seeds were sterilised and grown as described earlier (Wildwater *et al*, 2005) except that seedlings used for micro-array analysis and qRT–PCR were germinated on 1.2% plant agar. *Arabidopsis thaliana* ecotype Columbia 0 (Col-0) was used as wild type; T-DNA insertion lines *Atbrca1-3* (SALK_099751) and *e2fb-2* (SALK_120959) were obtained from the Nottingham *Arabidopsis* Stock Centre. The transgenic lines *sog1-1* (Yoshiyama *et al*, 2009), *e2fa-1*, *e2fa-2* and *e2fb-1* (MPIZ_244, GABI-348E09, SALK_103138, respectively) (Berckmans *et al*, 2011b) (Berckmans *et al*, 2011a)*), e2fa-3* (Xiong *et al*, 2013), *Atbrca1-1* (Reidt *et al*, 2006), *rRBr* (Wildwater *et al*, 2005) and *amiRBR* (Cruz-Ramirez *et al*, 2012) were described earlier. The T-DNA insertions and mutations were confirmed by PCR-based genotyping or sequencing and gene silencing was demonstrated via gene expressional studies and phenotyping. To study the *amiRBR;sog1-1* phenotype, more than 20 independent transformants were generated, genotyped by sequencing the *sog1-1* locus and analysed for *RBR* silencing. The overexpression lines *E2FA-DPA* (De Veylder *et al*, 2002) and *CYCD3.1OE* (Riou-Khamlichi *et al*, 1999; Dewitte *et al*, 2003) were described earlier. The construction of *E2FB-DPA* (Magyar *et al*, 2005) is described in the Appendix Supplementary Methods.

## Chemical treatments and induction studies

To induce DNA damage response, 5- to 6-day-old seedlings were transferred to tissue culture plates (unless stated otherwise), containing fresh MS liquid medium without or with 10 μg/ml mitomycin C (MMC), 20 or 3 μg/ml zeocin or 1 mM hydroxyurea (HU) and treated for 16 h or alternatively for short treatment periods of 1–4 h. For kinase inhibitory assay, 5- to 6-day-old seedlings were pre-incubated for 2 h in ATM or ATR kinase inhibitors (Selleckchem, KU55933, VE-821, respectively) which was applied to the MS liquid medium at 10 μM final concentration, afterwards MMC was given, as described above. Appropriate controls and mutants were treated simultaneously, and all treatments were repeated at least three times

($n$ = biological repeat) with 15–20 ($N$ = sample size) replicates. Although the level of MMC induction was varied between the different experiments, the ratio between controls and treated samples were comparable. Cell death in root tips was quantified by counting the number of PI-stained cells in the columella stem cells (CSC) and lateral root cap initials (LRC) and their daughter cells, and by measuring the contiguously PI-stained cell area directly adjacent to the QC in the proximal meristematic vasculature.

## Immunofluorescence labelling and fluorescence microscopy

Root excision and slide preparation of squashed root tips and immunolabelling with *Arabidopsis* anti-γH2AX and others were performed according to Amiard *et al* (2010) and Friesner *et al* (2005) with slight modifications; 3.7% paraformaldehyde with 0.05% Triton was used for 1 h and enzyme treatment was applied on root tips transferred and attached to microscopic slides. For dilution of primary and secondary antibodies, see Appendix Supplementary Methods. 5-Ethynyl-2′-deoxyuridine (EdU) labelling was performed in whole mount preparation of root tips (for details, see also in Appendix Supplementary Methods).

For fluorescence microscopy Olympus IX-81 FV-1000 confocal imaging system was used. For details of confocal laser scanning microscopy, image acquisition and processing, see Appendix Supplementary Methods. For Imaris section, *z*-stacks were taken with 0.2 μm *z*-step. Images were de-convolved using Huygens (Scientific Volume Imaging, Hilversum, The Netherlands) to remove out-of-focus information and sectioning of gained 3D objects was performed using Imaris software (Bitplane) in the section mode. The quantitative co-localisation analysis was performed using ImageJ software with JACoP (Just Another Co-localisation Plug-in, (Bolte & Cordelieres, 2006) based on Pearson's coefficient. A region of interest was defined by a square of unified pixel size (26 × 26), and image correlation analysis was performed by combining single stacks of green and red fluorescent images. The data analysis was generated using the Real Statistics Resource Pack software (Charles Zaiontz; www.real-statistics.com).

For phenotypic analysis, roots were stained in 5 μg/ml propidium iodide (PI) and analysed on Leica SP2 or Olympus IX-81-FV1000 inverted laser scanning microscope. For qualitative and quantitative comparison, images were recorded with identical microscope settings in all cases. EdU staining of replicating cells was performed using Click-iT EdU Alexa Fluor 488 HCS Assay (Molecular Probes, Eugene, OR, USA) as described earlier (Vanstraelen *et al*, 2009).

## Bimolecular fluorescent complementation and transient transfection assay

For BiFC, AtBRCA1 cDNA was subcloned to pGEMT-easy 221 (see primers Appendix Table S3). Subcloning of SCR and RBR cDNAs were described earlier (Welch *et al*, 2007) (Cruz-Ramirez *et al*, 2012) respectively). To generate split YFP construct, the binary BiFC GATEWAY-Destination vectors were used (Gehl *et al*, 2009). Four-week-old *Nicotiana benthamiana* plants were infiltrated by *Agrobacterium tumefaciens* containing different constructs as described by (Liu *et al*, 2010). The infiltrated region of the leaf was then mounted in water and checked for expression. YFP fluorescence was

visualised using a Zeiss LSM 710 confocal laser scanning microscope, and images were processed with the confocal microscope Zeiss ZEN software. Results from at least three independent experiments, and more than 20 infiltrated leaves were visualised.

### Transcriptional profiling, expressional studies and cloning

A detailed description is provided in Appendix Supplementary Methods. The micro-array data are submitted to GEO under accession GSE47715. Link: http://www.ncbi.nlm.nih.gov/geo/query/acc.cgi?acc=GSE47715.

### Chromatin immunoprecipitation (ChIP)

ChIP was carried out on root material of 5-day-old Col-0 seedlings to study RBR enrichment and on Col-0($E2FA_{pro}:E2FA_{gen}:GFP$) (Magyar *et al*, 2012) seedlings without and with 16-h MMC treatment to analyse E2FA-GFP enrichment. Here, $35S_{pro}:GFP$ was used as a control. To determine RBR enrichment, IP was performed in the absence and presence of antibody specific for RBR protein as described by Horvath *et al* (2006). For the detection of E2FA-GFP, GFP-trap beads (Chromotek) were used as described earlier (Schepers *et al*, 2001).

Primers for quantitative RT/PCR were designed to amplify fragments between 100 to 200 bp spanning the putative promoter region of *AtBRCA1*. The negative and positive controls are described earlier (Cruz-Ramirez *et al*, 2012). Primer pairs were analysed on the same biological material, repeated three times with three technical replicates for RBR and twice for E2FA. Enrichment for RBR was calculated by comparing the PCR data derived from immunoprecipitation samples with and without antibody and for E2FA between Col-0 ($E2FA_{pro}:E2FA_{gen}:GFP$) and $35S_{pro}:GFP$ lines. Student's *t*-tests were performed to analyse statistical significance. List of primers is given in Appendix Table S3.

### *In vitro* translation and pull-down

Full-length cDNAs for AtBRCA1 and RBR were obtained from the RIKEN Plant Science Centre and recloned into the pEU3II-HLICNot vector by ligation-independent cloning. *In vitro* transcription, cell-free translation, pull-down and immunoblotting were performed as described earlier (Nagy *et al*, 2015). See also Appendix Supplementary Methods.

**Expanded View** for this article is available online.

### Acknowledgements
We thank Charles White, Simon Amiard (Clermont Université, France), Oliver Trapp, Holger Puchta (Institute of Technology, Karlsruhe, Germany), Alfredo Cruz-Ramirez (Laboratorio Nacional de Genomica para la Biodiversidad, Cinvestav Sede Irapuato, Mexico), Ann Britt (University of California, at Davis, USA), Jim Murray, Walter Dewitte (Cardiff School of Biosciences, Cardiff University, Wales, UK), Wenkun Zhou (Wageningen University, The Netherlands) for sharing material; Christian Bachem (Wageningen University, The Netherlands) and Arp Schnittger (Hamburg University, Germany) for critically reading the manuscript; Laszlo Bako and Pal Miskolci (Umeå University, Sweden) for technical advice in ChIP technology, Kinga Bachem (Wageningen University, The Netherlands) for illustration and Milian Bachem (University of Rotterdam, The Netherlands) for statistical analysis. B.M.H was funded by a Marie-Curie IEF fellowship (FP7-PEOPLE-2012-IEF 330789), and by the Netherlands Organization for Scientific Research, Spinoza grant to B.S. G.S-P. was funded by the Utrecht University Focus & Mass grant and CBSG2/NCSB. Cs.P. was funded by a Marie-Curie IEF fellowship (FP7-PEOPLE-2012-IEF 330713) and both, Cs.P and L.B. by the BBSRC-NSF grant (BB/M025047/1). P.B. was funded by Grant Agency of the Czech Republic 15-11657S P501. Z.M. and E.N. were funded by the Hungarian Academy, OTKA 107838 and with Campus Hungary, TÁMOP-4.2.4B/2-11/1-2012-0001 fellowship, respectively. Z. M. and A.P-Sz. were supported by the grants GINOP-2.3.2-15-2016-00001 and Zs.D. by GINOP-2.3.2-15-2016-00032 and GINOP-2.3.2-15-2016-00020 from the Ministry for National Economy (Hungary). A.P-Sz. was funded by János Bolyai Research Scholarship of the Hungarian Academy of Sciences. S.P. was funded by ERA-CAPS grant 2013/15548/ALW. The funders had no role in the design of the study, data collection and analysis, decision to publish, or preparation of the manuscript.

### Author contributions
BMH and BS conceived the idea of analysing RBR targets and regulation of cell death, BMH, ZM and LB the links to E2Fs and BMH link to *sog1-1*. BMH performed expressional studies, transcriptome analysis, generation and characterisation of mutants ChIP with RBR on *AtBRCA1* promoter; the immunolocalisation and microscopical studies on localisation of RBR, E2FA, γH2AX and AtBRCA1 were performed by HK and PB, the expressional and phenotypic studies by EN, experiments on RBR and AtBRCA1 binding in wheat germ system by SN and TM and using BiFC technique by IB, the analysis of the micro-array data by BMH and GFS-P. ZA performed the cell death experiments with *CYCD3.1OE* line. ChIP analysis with E2FA-GFP protein was carried out by CP and SP generated and analysed the *amiRBR,e2fa* mutant. Protein complex isolation and LC-MS/MS identification were performed by AP-S and ZD. The manuscript was written by BMH, LB, PB and BS and seen and commented by all authors.

### Conflict of interest
The authors declare that they have no conflict of interest.

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
