## [Review Process File · The EMBO Journal]

Manuscript EMBO-2016-94561

Arabidopsis RETINOBLASTOMA RELATED directly regulates DNA damage responses through functions beyond cell cycle control

Beatrix M. Horvath, Hana Kourova, Szilvia Nagy, Edit Nemeth, Zoltan Magyar, Csaba Papdi, Zaki Ahmad, Gabino F. Sanchez-Perez, Serena Perilli, Ikram Blilou, Aladár Pettkó-Szandtner, Zsuzsanna Darula, Tamas Meszaros, Pavla Binarova, Laszlo Bogre and Ben Scheres

*Corresponding authors: Beatrix Horvath, Royal Holloway, University of London;
Ben Scheres, Utrecht University*

Review timeline:

Initial Submission date:	13 August 2015
Initial Editorial Decision:	12 October 2015
Re-submission date:	19 April 2016
Editorial Decision:	27 June 2016
Revision received:	07 December 2016
Editorial Decision:	30 January 2017
Revision received:	20 February 2017
Editorial Decision:	23 February 2017
Accepted:	23 February 2017

Editor: Hartmut Vodermaier

Transaction Report:

Initial Editorial Decision

12 October 2015

Thank you again for submitting your manuscript on RBR roles in the plant DNA damage response to The EMBO Journal, and please accept my apologies for the delay in getting back to you with a decision, related to limited availability of referees with suitable expertise. We have now finally received comments from two expert reviewers, in whose light we unfortunately had to conclude that the study does currently not represent a sufficiently compelling candidate for publication in The EMBO Journal. As you will see, referee 2 raises several major questions regarding the molecular basis of RBR-mediated induction of damage response genes, which I realize can probably be addressed through further work. On the other hand, referee 1 brings up a number of more serious concerns regarding the set-up of the study, the interpretation of the results, and the experimental support for some of the key conclusions of the manuscript. I will not repeat these points in detail here in this letter, but I am afraid that in light of these well-taken criticisms, we are not convinced that your present results have already led to a sufficiently clear and insightful picture that we would expect in an EMBO Journal article.

We realize the potential interest of defining a direct RBR role in the DNA damage response and would therefore not exclude the possibility of once more looking at a refocussed and extended new version of this study at some point in the future. However, given that the outcome of such further

work and reinterpretation is currently unclear, it is only fair to say that I am currently unable to make any strong commitments on eventual publication of such a new version, which would have to be treated as a new submission and only get sent back to our referees if the novelty was still uncompromised and the key concerns would appear to have been conclusively addressed.

In any case, thank you again for having had the opportunity to consider this work for The EMBO Journal. I am sorry that the referee reports do not allow me to come to a more positive conclusion at the present stage, but nevertheless hope you will find our referees' comments helpful.

REFeree REPORTS

Referee #1:

Rb, in animals, interacts with the E2F transcription factor family to suppress cell cycle progression. Rb phosphorylation integrates a variety signals (as there are many good reasons to induce cell cycle arrest), responding to both developmental and DNA damage-related inputs. In many cases, release of an E2F from RB allows that E2F to activate replication-related genes, including cyclins and PCNA.

In Arabidopsis RBR has been shown to be important in suppressing cell division (for example, in the QC) during development. It has been previously demonstrated that seedlings expressing 35S-driven antisense to this gene display excessive cell divisions around the QC, and spontaneous cell death. Arabidopsis RBR has also been shown to form numerous foci on chromosomes after DSB-formation during meiosis, and to play a role in crossover formation (this work, published in EMBO with an associated minireview, should certainly be cited here).

Here the authors further investigate the cell death previously described in the root tip cells that surround the QC in p35S:miRBR seedlings, and go on to describe events that occur in response to the crosslinking agent MMC, in wt vs. miRBR backgrounds. They look at the formation of Rb, BRCA1, and gammaH2AX foci, at cell death, and at the levels of expression of the DSB-responsive transcripts. They investigate the dependence of these events, in miRBR and wild-type root tips, on damage-detecting kinases ATM and ATR, on the repair/response platform BRCA1, and the DSB-responsive transcription factor SOG1. This involved a lot of work: there are 35 figures in the body of the paper, plus an additional 10 in the supplement. Many of the micrographs are very dim and hard to interpret, and many of the effects, though statistically significant, are very small, and these small effects are over- or mis-interpreted. In fact, very large effects are misinterpreted (see below). But there's also some data that's clear, interpretable, and interesting. My take-home message is that this paper needs to be cleaned up, extensively rewritten, and the data interpreted more carefully.

Addressing only the statements made in the abstract:

"We show that in Arabidopsis RBR... is intimately involved in the maintenance of genome integrity..."

I agree with this statement, though I'm not sure what "intimately" means. As previously shown, miRBR results in spontaneous cell death (Fig. 2C-J). We don't know what causes this death, but the upregulation of H2AX (Fig. 1AB) is indicative of stalled replication forks or chromosome breaks. This is a new observation. Less impressive, but still possibly indicative of DNA damage, is the upregulation of "DDR responsive" genes like BRCA1, an effect which is quite subtle (1.5-2.5x, Fig. 2, Fig. 3) in comparison to the effects of MMC or other DSB inducing agents. This uptick in expression might be the passive result of the fact that G1 is abbreviated, and more cells are in S phase (as indicated by EdU staining) which is when BRCA1 and RAD51 are employed- note other factors normally expressed in S phase are also upregulated 3-4x, Fig. 2). This data on upregulation of replication-associated genes supports the notion that Rb in plants, like Rb in animals and fungi, plays a role in repressing cell cycle progression- as has been shown before. No evidence is presented to suggest that the majority of this death is programmed- it may simply be passive mitotic death due to mitosis with incomplete replication, or with partially rereplicated chromosomes.

"We show that in Arabidopsis RBR... is intimately involved in DNA damage response"

I don't agree with this statement. The DNA damage response still occurs in the miRBR mutants. DDR is in fact enhanced in the absence of RBR. The authors demonstrate that miRBR cells still respond strongly to exogenous DNA damage. Death and focus formation (two phenomena that the authors are using as readouts of DDR) are shown to result from loss of RB, not from its presence. This puts RB in the category of genes (like repair proteins, or many other proteins involved in DNA metabolism) that prevent damage, and therefore prevent the accidents that result in the induction of damage response. For example a DNA ligase IV defect would exacerbate the response to IR, but we wouldn't say that DNA ligase IV plays a role in damage response. It's not part of the signal transduction pathway between damage and the cell cycle.

"We present evidence that RBR together with E2FA... is required for the cell-death response in the root-stem cell niche"

This is in direct contradiction to their results: miRBR roots exhibit both spontaneous death and enhanced MMC-induced cell death (which seems to be a simple additive effect- the sum of MMC induced death in wt plus the spontaneous death in miRBR). How can the authors possibly conclude from this enhanced cell death that conclude that RBR is required for cell death?

"RBR regulates the expression of a cohort of DDR genes and directly represses AtBRCA1 expression". The authors do present new data that indicates that RBR is physically associated with the BRCA1 promoter in normally growing plants- just as it is associated with another replication-related gene, PCNA1.

"Upon genotoxic stress, both RBR and AtBRCA1 act downstream of ... ATM and ATR..." We know already, from several publications, that BRCA1 responds at the transcriptional level to ATM. The statement about RBR seems to be based on the fact that the slight derepression of BRCA1 observed in miRBR is largely independent of ATM or ATR (p. 17). This doesn't necessarily mean that RBR is acting downstream of ATM and ATR. RBR could simply be acting independently of these factors- perhaps through its general upregulation of cell cycle progression.

...and can be independently recruited to gH2AX foci..." The authors do demonstrate that both BRCA1 and RBR can form foci upon MMC treatment, and that these colocalize with gH2AX foci, and that they are not always there at the same time (Fig. 5). BRCA1 focus formation is clearly dependent on ATM and ATR (as is gH2AX focus formation)- Fig. 6B. However, given the very high background level of antibody binding both in the nucleus and in the cytoplasm, the RBR "foci" are hard to define. It looks like some of the foci in Fig. 6B are not even in the nuclei. The dependence- or not- of RBR foci on ATR and ATM is not well supported in this figure- unlike the BRCA1 and gH2AX result.

"During the early step of damage signaling, AtBRCA1 is essential for mediating RBR function in the maintenance of genome integrity and initiating cell-death response.". The authors do have nice data indicating that in a brca1 mutant the spontaneous cell death induced by amiRBR occurs more slowly (Fig. 7F and G). However, given that cell death still occurs, I'd drop the "essential".

In summary, there's a lot of useful data here, but I find many of the authors' statements paradoxical. I think this paper would benefit greatly from substantial revision, rewriting and rethinking. Here are some suggestions:

The authors may need to separate the MMC induction of DDR story, which is clearly and not surprisingly ATM/ATR/SOG1 dependent, from the spontaneous cell death story, a major fraction of which does seem to independent of this DDR pathway.

The figures are very difficult to read. They would benefit by being broken up into less complex figures, making it easier for the reader's eye to move between the legend to the figure. Ideally, each figure should be understandable without a legend at all. In some case the figures have titles and axes labels that do explain what is being measured, but in many cases they don't. For example, "fold change" unfortunately means different things in different parts of a figure- help us by saying "fold change +/- MMC" or "fold change miRBR/wt". And in many cases the micrographs are not convincing- they're dim and the foci are poorly defined. In addition, the number of asterix placed above a column usually indicate levels of statistical significance. Perhaps letters would do a better

job here.

smaller details:

p. 4- The Culligan and Friesner refs do not discuss formation of gH2AX in S phase- in Friesner they're visualized in isolated prophase nuclei. Friesner is therefore also not the appropriate ref for the author's protocol.

p. 7 Fig. 1- tell us what cell populations are being scored for death. Also, Fig1A+B do not describe cell size.

The term "unscheduled DNA synthesis" means something very specific in the repair world: it is the scattered incorporation of nucleotides in nuclei that is associated with repair synthesis, as opposed to the complete labeling of nuclei associated with whole-genome replication in S phase cells. The EdU labeling here is not at the resolution to distinguish between these two. The enhanced number of labeled nuclei probably reflects the fact that a greater fraction of cells are in S phase. Also, just a thought, late S nuclei can easily be distinguished from early S nuclei with EdU labeling- that might be something you might take a look at.

P10, bottom, Upregulation of genes involved in replication might indicate that more replication is going on, not that replication forks are stalled.

Referee #2:

The current manuscript describes a new and unexpected regulation in the answer of plant cells to genotoxic stress. Whereas current models postulate that DNA damage response (DDR) is at least mainly induced via an ATM SOG dependent pathway the current manuscript demonstrates that RBR together with E2FA, but independently of SOG1, is required for the cell-death response in the root-stem cell niche. Moreover the authors show that there is a kind of repression of DDR genes like AtBRCA1. In CHIPSeq assays RBR is found together with E2FA directly at the promoter region of AtBRCA1.

I don't think that the author were able to demonstrate that RBR binds to the DNA of the promoter of AtBRCA1 directly. Most likely it binds via an interaction with E2FA. An important question not addressed is of course whether this is a general property of all EF2A binding sites.

If RBR - as shown by the authors - directly interacts with AtBRCA1, the question is whether AtBRCA1 can be found at its own promoter, too. It might thus be involved in the regulation of its own gene e.g. repression after induction.

Although open questions remain the results are interesting and revise our current picture of DDR in plants.

1st Editorial Decision - Re-submission

27 June 2016

Thank you for submitting a new version of your manuscript on a direct, BRCA1-related DNA damage response role of Arabidopsis RBR. I would like to sincerely apologize for the extraordinary delay in its review process, which was in part caused by the limited number of expert referees suitable and ready to review two back-to-back submissions involving several major groups in the field, but also by additional delays on part of the referees. Given your request for co-consideration, we sent both studies for in-depth review by one of the original and two new reviewers, and we have now finally received all sets of comments.

As you will see from the reports copied below, the referees find the proposal of a direct, not gene expression-related RBR role in DNA repair and damage responses potentially highly interesting, and also appreciate various other findings in the present manuscript. However, they are not convinced that the presented data currently offer sufficiently decisive evidence for such a direct RBR role, especially in the absence of deeper insights into its molecular function at damage foci or the consequences of its absence from damage sites. In this light, we still cannot consider the study ready

for publication in The EMBO Journal at the present stage, but we also realize the independent support from the second study as well as the overall constructive and encouraging nature of the referee reports. I would therefore like to now give you a formal opportunity for a major revision along the lines suggested by the referees, focussing especially on corroborating the experimental support for a direct, cell cycle/gene expression-independent RBR role. This should entail addressing the two main 'result' concerns of referee 1, in particular the issue with a key control in the appropriate background, as well as the major experimental concerns, control and presentation issues raised by referees 2 and 3. On the other hand, while following up on the RBR-BRCA1 interaction and its functional significance would certainly also help to further strengthen the key conclusions, we understand that this may be difficult to achieve within the scope of a regular major revision and would therefore not consider such data essential at this point.

REFEREE REPORTS

Referee #1:

Retinoblastoma protein(s), and their Arabidopsis homolog RBR1, play a well-established role in the regulation of entry into S phase. These protein binding proteins recognize the E2F family of transcriptional activator/repressor(s), inhibiting their activity. When phosphorylated by activated cyclin-dependent kinases Rb proteins are inactivated, and E2Fs become available to activate progression into S. Among the many intracellular conditions required for initiation of replication is the absence of detectable DNA damage- for example DSBs or stalled RNA polymerases- which would induce a G1 checkpoint. Retinoblastoma proteins therefore have a well established role in genome maintenance in humans, and RBR1 defects have been demonstrated to result in enhanced sensitivity to DNA damaging agents in plants.

The authors hope to demonstrate that RBR1 plays additional roles in genome maintenance outside of checkpoint signal transduction, i.e., that RBR1 is required for detection of damage and/or repair of damage. This is a challenge experimentally, as an RBR1 KO is lethal, and DNA damage is difficult to measure directly.

This paper presents (perhaps too) many studies, most of which are focused on demonstrating that *rbr1* defective plants are experiencing DNA damage (assayed somewhat indirectly as increased spontaneous and induced gH2AX foci, BRCA1 foci, and cell death), are slightly upregulated for DSB-induced transcripts (which might be a result of this enhanced spontaneous damage, or might be due to the derepression of S phase, as these transcripts regulate an S/G2-phase specific repair pathway). Dependence of this effect on E2Fa, which promotes entry into S, is also consistent with this role. Similarly, they present data indicating that BRCA1 and RBR1 physically interact, which hasn't been shown in plants (it has been well characterized in humans), but again, this interaction may be related to BRCA1's Rb-dependent role in growth inhibition (in humans) (Aprelikova 1999 PNAS). This is useful (particularly the studies on spontaneous damage) but not conceptually novel given RBR1's established role, in plants and humans, in protecting the genome through its role in suppressing, at the transcriptional level, transition to S phase. In summary, there's a lot of very good data, all of which could be consistent with Rb's known role in checkpoint maintenance. The authors attempt to separate the effects of RBR1 from the consequences of unchecked cell cycle progression via a control in which S phase is deregulated through OE of *CYCD3.1*, but I have issues with this control (see below). Until this control is performed in an isogenic line I can't accept it. There is always the possibility that RBR1 might play a role in some other aspect of genome maintenance (damage recognition, chromatin modification, or repair)- and a defect in any of these processes would, like a checkpoint defect, enhance the frequency of gH2AX foci and induce DSB-related transcripts. It's up to the authors to provide evidence for this second or third role for Rb in plants. Establishment of such a role would be newsworthy and of interest beyond the plant community.

In this paper, the authors argue that they have discovered a novel role for RBR1 in genome maintenance outside of its known checkpoint function. The author's best support for this hypothesis is the fact that they observe the formation of RBR1 foci at gH2AX foci. It's hard to see how that

could be related to regulation of gene expression, so I agree with them there. However, old models die hard. I agree with the authors that most of the phenotypes presented in this paper could result from, say, a defect in repair. But 95% of the data presented here is consistent-also- with Rb's known role in cell cycle control. This paper really is more of a (well-performed, extremely thorough) survey of Rb-defect induced phenotypes in plants than evidence, as suggested in the abstract and title, for a new role for Rb in genome maintenance.

More specific comments:

Results:

p. 6 An important and interesting control done badly. A major point for this paper is that RBR1 plays a role in maintenance of genomic stability beyond control of cell cycle progression- that it also plays a role in repair. The authors attempt to provide a (conceptually very nice) control for this effect by stimulating progression into S phase, in the presence of wt RBR1, using a CYCD3.1OE line- where enhanced cell death and gH2AX focus formation is not observed. Unfortunately this control is (apparently? It's not explicitly stated) performed in a Landsberg er background, which is not only not isogenic but includes the erecta mutation that affects growth of the plant (including meristem size) and response to a variety of stresses, while the effects of amiRBR are investigated in a Col background. (Unfortunately the construction of this line is not described at all in Methods, nor is a reference provided). This control needs to be done in Col, so that we know that the difference in effect on cell death is due to the difference in RBR1 status, not the many other differences between these two ecotypes.

In this same paragraph, the authors cite the 2016 Harashima review as a source for the statement ""CYCD3.1... is known to phosphorylate and inactivate RBR as a repressor of S phase entry". Harashima and Sugimoto do not mention CYCD3.1 in this review. In fact, H+S state: "What site(s) of RBR1 is phosphorylated is not shown so far and whether the observed phosphorylation affects the structure of RBR1 proteins and thereby affects the cell cycle progression has not been established in plants".

P7 "indicating that RBR requirement could be uncoupled from the restriction of S phase entry"- The fact that a given nucleus has gH2AX foci but does not happen to be replicating in the 10 minute interval queried does not mean that the observed damage occurred independently of entry into S phase. Cells experiencing damage generated by premature entry into S phase may be exhibiting no EdU incorporation because they are in a later phase of the cell cycle (including progression through M into G1 again) or may be experiencing an intra-S- checkpoint (cells arrested in S are not actively replicating). It doesn't make sense to expect all damaged cells that have entered S phase to somehow remain in a constant state of active DNA replication. Also, direct us to a specific Figure or Table, not just "Supporting Information", especially given that you have 15 SI documents, most with multiple panels.

Writing issue:

P7 (line numbers would be helpful throughout) The authors suggest that the increased number of foci "could indicate that... RBR maintains genomic integrity". RBR does help to maintain genomic integrity, through its checkpoint function. Surely you mean something more specific. Similarly later "...possibility that in addition RBR protects DNA from damage"... yes, we already know RBR1 does this! That's what a checkpoint does- it prevents a bad situation (say, a polymerase-blocking lesion) from getting worse (becoming a double strand break during replication). In your writing, try to point out what actually is unexpected if RBR1 plays a role only in the transcriptional control of genes affecting entry into S phase. Point these out. Colocalization with gH2AX foci is an example. I am having a hard time finding a second example. Improving the CYCD3.1OE control and extensively characterizing the differences between this (= hyperphosphorylated Rb) and KD of Rb would really help.

P8 "The two different classes of foci rarely coexisted in the same cell (Fig 2D). Add Table S1 to the main text- this, not Figure 2D, has the data. Cite a paper that tells us whether the BRCA1-1 mutant is null for BRCA1 expression, and tells us that here.

P9 The authors argue that ATM and ATR inhibitors prevent- or greatly reduce- the formation of RBR, gH2AX, and BRCA1 foci formation, citing fig. S3B and C. Fig S3B is on a different topic

which seems to have been mistakenly left out of the text.

P9 (writing) again, repair is part of maintaining genome integrity. If you mean damage induced checkpoints, say so.

P10 "the presence of canonical E2F binding sites in the promoter region of the differentially expressed genes" perhaps some text was lost here? Did some fraction of genes have this sequence? Tell us about this.

P12+13 The lack of amiRBR-induced cell death in the BRCA1 mutants is interesting, particularly given BRCA1's lack of importance in damage-induced PCD. The authors include an image of a pRBR:GFP construct in the same root to give a feel for the degree of silencing induced by the microRNA. All the GFP images look overexposed- to me. Could we compare them on the same scale (gain) as a non-amiRBR plant? That will allow us to see if there's any silencing in amiRBR, and if it's maintained in the double mutants.

Similarly, the anti-cell death effect of the brca1 mutations calls into question the similar effect of the E2FA mutations. Could the authors provide a control that demonstrates that the amiRBR transgene is still functioning in these "double mutant" lines?

P13. Fig 7 Independence from SOG1- just a thought and a minor correction: SOG1 isn't required for all damage-induced cell death- its only required for the very localized (around the QC) death that occurs within hours of treatment. At later times (a day or more after treatment) SOG1 (and ATM/ATR)-independent death is observed throughout the mitotic zone. There's no evidence that this is programmed.

Why do the authors quantify cell death only in root cap cells? I suppose because they are easy to score and represent a limited population? But these particular cells spontaneously undergo cell death as the root cap is shed, don't they? So the scoring might dependent on many factors, including the rate of the cell cycle, rather than DNA damage. In contrast, in the images shown the majority of the (damage-induced) death observed occurs in the younger cells of the stele. I think it is very important that the authors not mix and match this data- they always need to make it clear in the text which effect they are observing.

In summary, I think there are a number of reasonably interesting observations here, which clearly indicate that RBR1 plays a role in genome maintenance in plants as well as in mammals. But the topic addressed in the title and abstract- that RBR1 is more than a transcriptional regulator of entry into S phase, is supported only by the observation (already made in mammals) that Rb, for unknown reasons, accumulates at gH2AX foci.

Referee #2:

The retinoblastoma protein is best known, both in animals and plants, for its role in cell cycle progression as a transcriptional repressor of E2F target genes. However, it is a multifunctional protein with other less characterized functions. Here, the authors show its participation in genome integrity and DNA damage response (DDR). They find that RBR, the Arabidopsis Rb homologue, and BRCA1 are recruited to the sites of MMC-induced DNA damage, in a subset of gamma-H2A.X foci. They confirm that BRCA1, as well as other genes that participate in DDR are E2F targets. The major claim of this study is that RBR are part of a regulatory network coordinating cell proliferation and genome integrity.

This manuscript contains a valuable set of results addressing a poorly understood function of plant RBR. The manuscript contains a large amount of data but its current structure is not very clearly justified. A significant amount of the work is devoted to demonstrate the BRCA1 is an E2F target. Another focuses on the colocalization of RBR and BRCA1, but this is not pursued to really provide details about complex formation, dynamics and function. In my opinion, the study is of high quality but needs further experiments to provide compelling support of the conclusions, and leaves several unanswered crucial points, including a demonstration of the functional relevance and/or significance of the presence of RBR in the DNA damage foci, and of the functional coordination between cell

proliferation and genome integrity. A discussion on how is RBR targeted to DNA lesions is also necessary. In addition, there are several points that need clarification and/or further experiments, as detailed below.

Some specific points.

1. page 7, three first lines. This conclusion needs to be explained.
2. Fig. 2. How is separated the function of RBR in the formation of punctate patterns from its role as a transcriptional regulator (in rRbr lines).
3. Page 9. The RBR-BRCA1 interaction seems to be rather weak. Is this functionally relevant in vivo? Is this due to transient formation of a complex? I wonder if alternative approaches, e.g. BiFC, would provide more hints into the significance of this interaction. How does the phosphorylation status of RBR affect complex formation and function?
4. Page 10. Fig. 4E. PCNA1 is used as a positive control. In that case, do fragments 1 and 3, lacking E2F sites give a positive binding? What is the profile of RBR binding after MMC challenge?
5. PAGE 11. The use of e2fa mutants is interesting but requires (i) a detailed characterization of the various alleles, (ii) are they knock-out or knock-down in terms of mRNA produced? (iii) could a truncated protein be produced? (iv) Can this explain some of the results obtained (dominant negative effects after sequestering DP? (v) What is the role of an E2F truncated protein bearing a marked box?
6. Page 13. Lines 2-4. Explain suppression of the cell death response in amiRBR by brca1 mutations.
7. Page 14. Line 5. The differences between de-repression and activation needs further justification in this context.
8. It is already known that E2F colocalizes to DNA damaged sites (also in animal cells). Based on this, finding RBR in those sites could be also expected.

Referee #3:

General comment on the two manuscripts:

The authors B. M. Horvath et al. (Scheres and Boegre labs) submitted a manuscript with the title "Arabidopsis RETINOBLASTOMA RELATED is involved in repair and DNA damage response". The authors S. Biedermann et al. (Schnittger lab) submitted a manuscript with the title "The Retinoblastoma homolog RBR1 mediates localization of the DNA repair protein RAD51 to DNA lesions". The manuscripts should be considered for back-to-back publication. While the Scheres/Boegre paper has a lot of data, the Schnittger paper is much less substantial and appears often sloppy (no NGS mRNA analysis, no co-IP interaction data; missing size bars, some statistical analysis missing). Both manuscripts emphasize that RBR not only has a function in cell cycle regulation and transcription but also a direct function in DNA repair. Both studies underpin this latter point by co-localization data between RBR, DNA damage markers (γ H2AX) and DNA repair proteins (BRCA1 or RAD51). The Scheres/Boegre study also performed an additional experiment (co-IP) to demonstrate the co-existence of RBR and BRCA1 in the same complex. Unfortunately, both studies suffer from technical short-comings related to afore mentioned key experiments (detailed evaluation below) and it remains unclear if RBR is really targeted to DNA lesions, co-localizing with DNA repair factors and if it has a direct function and not an indirect one (via control of transcription of genes encoding DNA repair proteins and cell cycle factors). The accompanying experiments (cell death studies in root tips, sensitivity assays, epistatic analyses, mRNA expression and promoter control analyses) are not discriminating between an indirect or direct contribution of RBR to DNA damage response. It is important to highlight, that a principle involvement of RBR in DDR and DNA repair is unambiguously shown in both studies. The direct involvement of RBR in DNA repair in plants has already been hypothesized earlier (in a study related to meiotic DNA repair - Chen et al 2011, EMBO J.; in a study by the Scheres lab, Cruz-Ramirez et al., 2013 PLoS Biol.) but not conclusively answered back then. Furthermore, there are conflicting data comparing the given studies and the previous Chen et al. study: now Biedermann et al. report co-localization of RBR with RAD51 in mitotic nuclei, while the previous study of Chen et al. clearly showed no co-localization of these two factors during meiosis; Horvath et al, report co-localization of RBR with BRCA1, a factor needed for DNA repair and speculate about a failure of DNA repair in the mitotic

nuclei with reduced RBR levels, yet the previous study by Chen et al. did not observe any DNA repair defects (only defects in connecting to the homologous partner). None of these conflicts are further discussed in the two given manuscripts. In this sense, the two given manuscript fail to provide a strong and non-ambiguous answer for the interesting question if RBR in plants is directly involved in DNA damage repair.

In principle the addressed questions and the submitted findings are interesting and the authors should be given a chance to address all raised points of criticism. Special attention should be given to data quality and the key question of RBR co-localization and co-existence in the same complex together with established DNA repair factors.

General comments on the B. M. Horvath et al. manuscript:

The authors B. M. Horvath et al. (Scheres and Boegre labs) submitted a manuscript with the title "Arabidopsis RETINOBLASTOMA RELATED is involved in repair and DNA damage response". The authors state that their data indicates that RBR not only is involved in cell cycle control but also in safeguarding DNA integrity. This latter insight is certainly new and has not been studied in depth before in plants. It should be noted though, that in 2011, a joint paper of the Franklin and Berger labs analyzed the importance of RBR in meiosis (Chen et al., 2011; EMBO J.). Not much reference is given to this study, yet certain key findings in the given manuscript appear not in line with the previous study (see details above and below). Furthermore, the authors emphasize that their data indicate a direct involvement of RBR in plant DNA repair, acting together with DNA repair factors, localizing to chromatin/DNA to promote DNA repair. In mammalian cells, the direct involvement of (the mammalian homologue of RBR) pRb in DNA repair has been suggested by co-IP experiments, especially highlighted in Cook et al. (2015 Cell Rep.) with evidence of pRb interacting with proteins involved in cNHEJ (Ku70/80/DNA-Pk; in Xiao and Goodrich (2005 Oncogene) with evidence of interaction between pRB and BRCA1 and Top2...etc... Conversely, Lang et al. (2012 New Phyt.) published that in Arabidopsis E2F, a binding partner of RBR involved in transcriptional control, co-localizes with γ H2AX. This latter results would rather suggest that RBR is not directly involved in DNA repair but possibly targeted together with E2F to DNA lesion sites (to integrate the DNA damage signals and release repression of genes encoding DNA repair factors globally). Indeed the authors provide very solid data on RBR dependent DNA repair gene de-repression upon genotoxic stress - rather supporting an indirect role of RBR in DNA damage response. No doubt, it is certainly intriguing to speculate about a direct role of RBR in DDR, but the data in the literature comes from different model systems, is partly conflicting and/or not convincing. In this sense, any statement on RBR's role in plant DDR has to be very solid and beyond any doubt. Unfortunately, the authors fail to make this point (see below).

Specific comments on the B. M. Horvath et al. manuscript:

Title:

Change title to "Arabidopsis RETINOBLASTOMA RELATED is involved in DNA repair and damage response"

Introduction:

The first paragraph introduces the topic of cell cycle regulation via RBR and relates to prior research performed in plants.

The second paragraph introduces DNA damage response but exclusively quotes 4 non-plant reviews published between 2006 and 2011. Please up-date the paragraph with newer references and include references to DNA repair in plants.

The third paragraph starts with: "Depending on whether DNA damage results in single- or double strand breaks, different pathways are induced....etc.." Please also include the cell-cycle state as a determinant of DNA repair pathway choice. Furthermore, please include in the third and forth paragraph plant-specific references when introducing "ATM" and "ATR". The reference to the review "Ciccica and Elledge, 2010" appears 4 times in the introduction...please find more appropriate and further references. With respect to ATM and ATR activation: the stimuli are dsDNA breaks (ATM) or free ssDNA (ATR)...pl. correct. (Certainly it is true that stalled replication forks may lead to ssDNA and activate ATR, but the context demands to name a more specific elicitor...). It may also be interesting to mention in this context that in mammalian cells there is extensive

evidence for activation of ATM by oxidative stress and that in the context of cell cycle regulation/HU stress this has also been proposed for plants (Yi et al 2014)...pl include these findings and according references.

Page 4:

"The ATM- and ATR activated WEE1 kinase mainly controls the replication checkpoint (Cools et al, 2011, Dissmeyer et al, 2009),..."

Please provide reference/evidence for WEE1 being ATM/ ATR activated...the given references are not including this information.

Page 5:

Pl. rephrase the entire last paragraph; first, as outlined above/below, the statements seem not justified; second the wording itself is problematic:

"..(i) sensing the DNA damage foci, where RBR can function together with AtBRCA1 in repair,..." . what is meant with "sensing the DNA damage foci"? This does not make sense. Furthermore, that "RBR can function together with AtBRCA1 in repair" is certainly not shown in the manuscript.

Results:

The first paragraph introduces the used RBR RNAi lines. Please clearly describe the lines used in the given study in relation to the ones used by Chen et al. (rbr-2) and by Borghi et al. (RBRi) so that in the "discussion" the discrepancies in results can be discussed.

Figure 1D: why is the signal of γ H2AX not overlapping the DAPI signal 100%? This signal should not be outside the DAPI-pos. area. Is the signal specific? Pl provide explanation or better picture.

Figure 1F: the amiRBR EDU pos/neg. cells are not shown (Suppl. Fig 1 shows only the rRBR data; overlay is also missing: pl. provide). Please provide the data and show channels individually and as an overlay.

Page 7:

"...of this category also increased in amiRBR (Figure 1F), again indicating that RBR requirement to prevent DNA damage can be uncoupled from the restriction of S-phase entry." This statement is most likely not true, since the EdU pulse has been performed for 15min only (unclear when cells have been fixed - pl. provide experimental details) and the DNA damage from a prior S-phase may still be retained and yield a cell that is EdU negative but γ H2AX positive. Indeed, accumulation of DNA damage due to premature cell division without completing S-phase is very likely to occur in the constitutively silenced amiRBR transgenic plant line.

Page 7:

"Next we studied whether lines with RBR silencing were more sensitive to genotoxic stress and found elevated γ H2AX labeling..."

Pl. rephrase: elevated γ H2AX foci are not measuring sensitivity to genotoxic stress...

"The accumulation of DNA damage foci in the RBR silenced meristematic cells could indicate that, like Rb in animal cells, plant RBR maintains genome integrity.." Pl. provide according pictures and display together with the graph shown in Fig. S2A.

The authors find more cell death and more cells with γ H2AX foci after 16h MMC treatment and conclude that DNA damage may accumulate due to compromised RBR function. The reviewer wants to emphasize that the most consistent interpretation would conversely be that in the amiRBR or rRBR lines pre-mature S-phase/cell division take place. This is concluded from the 16h time window of MMS treatment, and also from the fact that RBR depletion has been correlated (by the very same authors) with pre-mature entry into S etc... Furthermore, it should be noted that MMC acts as an intrastrand crosslinker. Only during S-phase (!) the DNA intrastrand crosslinks will compromise replication and need to be repaired and given sufficient time to do so. This has to be considered and integrated.

Figure 2A

16h MMC...see comment above.

Cytology of somatic nuclei exposed to MMC:

RBR localizes only as 1-4 large foci per nucleus but only in 17% of all cells (what about the remaining 83%(!)?). The foci areas take up about 1 micrometer in width, which is about 25% of the entire width of the somatic nuclei. Similarly, the γ H2AX foci reside as few, large foci in the nuclei (the picture provided shows 4 foci) with only about 20% of all cells showing labeling altogether (Fig. S2A) (what about the other 80%?). Interestingly, the observed RBR and γ H2AX foci appear side-by-side but not overlapping, and also side-by-side with intensely DAPI-stained bodies. The authors also provide a graph of measured fluorescence intensity in the respective channels, to underline their statement of co-localization.

The experimental section does not explain how the pictures are acquired: are these single stacks or are these (max. intensity?) projections? Why not performing a 3D re-construction with the (most likely) available z-layers. How is co-localization defined? Please provide a definition? Are these foci in the same z-level? Has the picture acquisition been done in a manner that wave length shifts has been considered (which could work potentially in favor of the author's arguments, since weirdly, the foci seem to cluster but not overlap)? Furthermore, to argue for co-localization (according to a definition yet to be provided) a statistical test and a comparison to a random situation is needed. Preferentially this test should be done in 3D (and not on a projection!) using the actually measured nuclei volume, exclude the volume of the nucleolus and use the average size of the foci volumes.....

Please include in your analysis and discuss the adjacent DAPI-stained bodies: are these centromeric regions? The reviewer points out that Coschi et al. 2014 (Cancer discovery; not a plant study) found a protein complex associating with pericentromeric repeats comprised of E2F1, condensin and pRb. If this is also true in plants, the nature of the presented staining (a few massive foci of RBR) would be in accordance with previous findings in mammalian cells. In general, the low amount of cells that show a staining altogether after 16h of MMC treatment and the low amount of γ H2AX foci in those few cells may reflect different technical short-comings: e.g. limited MMC stability and/or penetration; over-fixation of cells/proteins; limited permeability for antibodies to enter the cells/nuclei during the staining procedure...etc...

Please re-do and extend the analysis and re-write the paragraph accordingly.

In any case, please delete the following sentence:

"These observations supported a role for RBR in DNA damage sensing without excluding the possibility that in addition RBR protects DNA from damage." How can these experiments possibly probe for RBR (or any other protein) sensing DNA damage? The experiment just probe for (co-)localization of proteins in cells. Phosphorylated γ H2AX is a downstream consequence of a prior detection of DNA lesions - co-localization would merely indicate that the two proteins reside at the same locus. Moreover, the peculiar nature of the γ H2AX foci make it uncertain if these foci are actually representing DNA damage. One way to test this would be to elevate the concentration of MMC, to see a concentration dependency of the foci numbers and the number of pos. cells. Please consider that MMC will only lead to DNA damage in S-phase and therefore an accumulation of DNA damage in amiRBR or rRBr upon MMC treatment is weakening the argument that RBR may act directly on DNA lesions and not indirectly by regulating cell cycle and transcription (as elegantly shown in the following parts of the given manuscript). Using a different drug, like bleomycin or a derivative, that chemically introduces ssDNA and dsDNA lesions irrespective of the DNA replication may be helpful, especially if co-treated with EdU to filter for those cells that have not undergone replication during the experimental window.

The authors probe for BRCA1, mentioning its pivotal role in mammals, but not mentioning its modest importance in plants (please include information and reference to work from H. Puchta). The authors argue for "up-regulation" of AtBRCA:GFP in the meristematic cells of the rRBr line but more AtBRCA1:GFP foci could also represent more DNA damage after pre-mature S-phase entry and therefore more focused localization of AtBRCA1:GFP...etc.... Again, it would be interesting to test G1 cells after bleomycin treatment, and see if presence or absence of RBR makes a difference for the observed "speckles". The co-localization data suffers from the same shortcomings in technical implementation and analysis as outlined above. The quality of the cytological analysis presented in Figure 2D is poor. It is very hard to decide if the shown images represent genuine staining, technical artefacts or background (compare the different classes of γ H2AX staining: the

authors believe they are meaningful, the reviewer is not convinced.) The low staining efficiency in general (see above) and the low quality of the pictures make it impossible to decide if these pictures are meaningful.

Please provide better data/pictures also for the BRCA1 analysis.

The experiment depicted in Figure 3A, aiming to show direct interaction between RBR and AtBRCA1 is not convincing at all. Please provide more convincing interaction data and ideally corroborate with a different experimental strategy.

The experiment depicted in Figure S3C also suffers from the picture quality and the unclear interpretation of foci. Are they specific or background? Why is the partial lack of DAPI co-localization with γ H2AX shown in Figure 1 ignored but seriously interpreted with respect to RBR and DAPI in Figure S3C. Again, the settings of the microscope, the wavelength shift, the 3D vs the projection issue....etc... All this prohibits drawing convincing conclusions from these data.

The following sentence does not make sense: "If RBR function was solely required to maintain genome integrity, while AtBRCA1 was involved in repair....", pl. re-phrase.

Page 12:

AtBRCA1 genepl correct

Page 13:

Please give an explanation why the brca1-3 mutant line has been introduced.

Page 13:

While it is interesting to learn about the (partial) suppression of the amiRBR cell death phenotype by brca1 mutants it would also be interesting to learn about survival and fitness of the entire plant.

Page 13:

"We detected a remarkable overlap between genes induced upon genotoxic stress in an ATM/SOG1 dependent manner and...." Pl correct to "We observed...." and furthermore change the word "remarkable" to "significant", if true ...if so, then pl provide the numbers (statistical test against random...etc..!)

Discussion:

Pl rephrase sentence: "Cells are protected from DNA damage by imposing quiescence..."

Pl revise sentence after additional data has been added:

"We show that upon genotoxic stress RBR accumulates in large nuclear foci, which significantly overlap with DNA damage sites labelled for γ H2AX....". Define "overlap" and "co-localisation", give values for "significant"....

Delete the following sentence:

"During homologous recombination of meiotic chromosomes, RBR is recruited to DSBs,"since it is not clear if true (certainly they do not coincide with RAD51 or DMC1).

Page 18:

Pl rephrase last paragraph according to the new data. The following statement needs re-phrasing in any case: "(i) protection and sensing of DNA integrity".

RESPONSE TO REFEREES

We were very pleased with the constructive criticisms of all reviewers and have responded to all points of the referees in detail as elaborated below. We have performed additional experiments to address the two main critiques posed by the referees and we feel that this work has improved the manuscript substantially. Before dealing with the specific points raised, we summarize the main changes:

1. We have strengthened our evidence that RBR has cell-cycle independent roles in plant DNA damage response by repeating, as suggested, the experiment now using CycD3;1 overexpression in the Col-0 background. In addition, we have analysed E2F/DPA lines which again uncouple cell cycle progression and DNA damage response. Quantification of γ H2AX foci reveals that referee 1 was right to suspect some direct effect of forced cell cycle progression on DNA damage, but clarifies that this is not enough to provoke a cell death response.
2. We have improved our colocalization studies as a response to referees 2 and 3. It remains clear that RBR is recruited to specific γ H2AX foci where E2FA also accumulates, and form a subset of the larger set of AtBRCA1 positive foci. Importantly, we have extended the evidence that AtBRCA1 protein can directly bind RBR by using a split-YFP assay in plant cells. We believe that these new data together with the consolidation of cell cycle independency mentioned above under (1) and the RAD51 data from the accompanying paper make a strong case for cell cycle- and transcription independent RBR roles in DNA damage control.

Referee #1:

Retinoblastoma protein(s), and their Arabidopsis homolog RBR1, play a well-established role in the regulation of entry into S phase. These protein binding proteins recognize the E2F family of transcriptional activator/repressor(s), inhibiting their activity. When phosphorylated by activated cyclin-dependent kinases Rb proteins are inactivated, and E2Fs become available to activate progression into S. Among the many intracellular conditions required for initiation of replication is the absence of detectable DNA damage- for example DSBs or stalled RNA polymerases- which would induce a G1 checkpoint. Retinoblastoma proteins therefore have a well established role in genome maintenance in humans, and RBR1 defects have been demonstrated to result in enhanced sensitivity to DNA damaging agents in plants.

The study on silencing of RBR specifically in the QC (pWOX5::amiGORBR) (Cruz-Ramirez et al, 2013) described and connected the effect of DNA damage upon zeocin treatment to QC maintenance, and did not examine the effects of RBR silencing on DNA damage responses, such as cell death, without genotoxic agents.

The authors hope to demonstrate that RBR1 plays additional roles in genome maintenance outside of checkpoint signal transduction, i.e., that RBR1 is required for detection of damage and/or repair of damage. This is a challenge experimentally, as an RBR1 KO is lethal, and DNA damage is difficult to measure directly.

This paper presents (perhaps too) many studies, most of which are focused on demonstrating that *rbr1* defective plants are experiencing DNA damage (assayed somewhat indirectly as increased spontaneous and induced γ H2AX foci, BRCA1 foci, and cell death), are slightly upregulated for DSB-induced transcripts (which might be a result of this enhanced spontaneous damage, or might be due to the derepression of S phase, as these transcripts regulate an S/G2-phase specific repair pathway). Dependence of this effect on E2Fa, which promotes entry into S, is also consistent with this role. Similarly, they present data indicating that BRCA1 and RBR1 physically interact, which hasn't been shown in plants (it has been well characterized in humans), but again, this interaction may be related to BRCA1's Rb-dependent role in growth inhibition (in humans) (Aprelikova 1999 PNAS). This is useful (particularly the studies on spontaneous damage) but not conceptually novel given RBR1's established role, in plants and humans, in protecting the genome through its role in

suppressing, at the transcriptional level, transition to S phase. In summary, there's a lot of very good data, all of which could be consistent with Rb's known role in checkpoint maintenance.

The authors attempt to separate the effects of RBR1 from the consequences of unchecked cell cycle progression via a control in which S phase is deregulated through OE of *CYCD3.1*, but I have issues with this control (see below). Until this control is performed in an isogenic line I can't accept it.

There is always the possibility that RBR1 might play a role in some other aspect of genome maintenance (damage recognition, chromatin modification, or repair)- and a defect in any of these processes would, like a checkpoint defect, enhance the frequency of gH2AX foci and induce DSB-related transcripts. Its up to the authors to provide evidence for this second or third role for Rb in plants. Establishment of such a role would be newsworthy and of interest beyond the plant community.

In this paper, the authors argue that they have discovered a novel role for RBR1 in genome maintenance outside of its known checkpoint function. The author's best support for this hypothesis is the fact that they observe the formation of RBR1 foci at gH2AX foci. Its hard to see how that could be related to regulation of gene expression, so I agree with them there.

However, old models die hard. I agree with the authors that most of the phenotypes presented in this paper could result from, say, a defect in repair. But 95% of the data presented here is consistent-also- with Rb's known role in cell cycle control. This paper really is more of a (well-performed, extremely thorough) survey of Rb-defect induced phenotypes in plants than evidence, as suggested in the abstract and title, for a new role for Rb in genome maintenance.

We collected new data using Col-0(*CyCD3.1OE*), E2FA/DPA and E2FB/DPA overexpression lines with similar levels of extra S-phase entry compared to *amiRBR*, but without cell death response. Based on these data, we are confident to conclude that RBR has an additional role in genome maintenance beside its cell-cycle checkpoint function. This finding is supported with the data showing that cells with depletion of RBR show higher frequency of nuclei with yH2AX labeling than cells with RBR hyperphosphorylation (*CYCD3.1OE*).

As the referee points out, the observed responses could be consequences of unscheduled S-phase entry or defective repair, both resulting in DNA damage. Our new analysis shows that this is indeed partly the case, but at the same time strongly supports the cell cycle independent role.

More specific comments:

Results:

p. 6 An important and interesting control done badly. A major point for this paper is that RBR1 plays a role in maintenance of genomic stability beyond control of cell cycle progression- that it also plays a role in repair. The authors attempt to provide a (conceptually very nice) control for this effect by stimulating progression into S phase, in the presence of wt RBR1, using a *CYCD3.1OE* line- where enhanced cell death and gH2AX focus formation is not observed.

In our previous submission, we have only provided evidence that *CYCD3.1OE* in Landsberg erecta background showed extra S-phase entry without any cell death response. No data was given for yH2AX foci formation. Now these data from Col-0(*CyCD3.1OE*) are also incorporated and shown in Figure 1E and F.

Unfortunately this control is (apparently? It's not explicitly stated) performed in a Landsberg er background, which is not only not isogenic but includes the erecta mutation that affects growth of the plant (including meristem size) and response to a variety of stresses, while the effects of *amiRBR* are investigated in a Col background. (Unfortunately the construction of this line is not described at all in Methods, nor is a reference provided). This control needs to be done in Col, so that we know that the difference in effect on cell death is due to the difference in RBR1 status, not the many other differences between these two ecotypes.

The reference for the Ler(*CYCD3.1OE*) G54 line (Dewitte *et al*, 2003, Riou-Khamlichi *et al*, 1999) and the description of the introgression line (Col-0) used during resubmission is now incorporated in the Supplementary Information.

Experiments carried out on the near-isogenic line (Col-0, F3) show that the difference in effect on

cell death is due to the difference in CYCD3.1, not the other differences between these two ecotypes.

The Col-0(*CYCD3.1OE*) showed also extra S-phase entry comparable to *amiRBR* (Figure 1A, 1C). The frequency of nuclei with γ H2AX labelling increased in both mutants compared to Col-0 but it was remarkably lower in the Col-0(*CYCD3.1OE*) line (Figure 1E and F), showing that DNA damage response is not only caused by the aberrant checkpoint regulation.

In this same paragraph, the authors cite the 2016 Harashima review as a source for the statement "'CYCD3.1... is known to phosphorylate and inactivate RBR as a repressor of S phase entry". Harashima and Sugimoto do not mention CYCD3.1 in this review. In fact, H+S state: "What site(s) of RBR1 is phosphorylated is not shown so far and whether the observed phosphorylation affects the structure of RBR1 proteins and thereby affects the cell cycle progression has not been established in plants".

The reviewer is correct, the review of Harashima was too general for this statement. In plants RBR phosphorylation was studied by the P-RbS807 antibody that recognizes the corresponding P-RBRS911 (Magyar et al 2012). In a work connecting E2Fs and pathogen induced cell death the authors show through site directed mutagenesis of the RBRS911 that the P-RbS807 antibody is specific (Wang et al, 2014). The sentence referred to by the referee was deleted and the paragraph reformulated.

P7 "indicating that RBR requirement could be uncoupled from the restriction of S phase entry"- The fact that a given nucleus has γ H2AX foci but does not happen to be replicating in the 10 minute interval queried does not mean that the observed damage occurred independently of entry into S phase. Cells experiencing damaged generated by premature entry into S phase may be exhibiting no EdU incorporation because they are in a later phase of the cell cycle (including progression through M into G1 again) or may be experiencing an intra-S- checkpoint (cells arrested in S are not actively replicating). It doesn't make sense to expect all damaged cells that have entered S phase to somehow remain in a constant state of active DNA replication. Also, direct us to a specific Figure or Table, not just "Supporting Information", especially given that you have 15 SI documents, most with multiple panels.

We agree that this possibility exists. This experiment was further complicated by the fact that EdU incorporation can induce DNA damage. For this reason, the experiments and conclusion related to this comment were deleted because we have now different and stronger data for cell-cycle progression independent DNA damage.

Writing issue:

P7 (line numbers would be helpful throughout) The authors suggest that the increased number of foci "could indicate that... RBR maintains genomic integrity". RBR does help to maintain genomic integrity, through its checkpoint function. Surely you mean something more specific. Similarly later "...possibility that in addition RBR protects DNA from damage"... yes, we already know RBR1 does this! That's what a checkpoint does- it prevents a bad situation (say, a polymerase-blocking lesion) from getting worse (becoming a double strand break during replication). In your writing, try to point out what actually is unexpected if RBR1 plays a role only in the transcriptional control of genes affecting entry into S phase. Point these out. Colocalization with γ H2AX foci is an example. I am having a hard time finding a second example. Improving the *CYCD3.1OE* control and extensively characterizing the differences between this (= hyperphosphorylated Rb) and KD of Rb would really help.

The RBR/E2F pathway was deregulated through overexpression of *CYCD3.1*, as well as the two activator E2Fs, E2FA and E2FB. If DNA damage upon RBR silencing originated solely from RBR's cell cycle role, these genetic alterations should be similarly effective; but they are not. Similarly, if RBR functions solely in the DNA damage checkpoint, E2F overexpression would be expected to override this checkpoint, but it does not. Our most parsimonious model is that RBR levels affect DNA damage through diverse mechanisms that are at least partially separable from the cell cycle, or checkpoint regulatory functions. The recruitment of RBR and E2FA to damage foci and RBR interaction with AtBRCA1 indeed suggest such a direct involvement.

P8 "The two different classes of foci rarely coexisted in the same cell (Fig 2D). Add Table S1 to the

main text- this, not Figure 2D, has the data.

Data presented in Table S1 are now shown as Table 1 associated with the main text and described appropriately there.

Cite a paper that tells us whether the BRCA1-1 mutant is null for BRCA1 expression, and tells us that here.

The *Atbrca1-1* mutant was characterized earlier (Reidt *et al*, 2006) and described as a non-functional mutant. Using a primer pair to amplify the nearly full -length mRNA, the authors did not visualize any DNA fragment via agarose gel. Using C-terminal specific primer pairs, similarly to the above publication, we could also detect a minute amount of amplified fragment (Figure S4B). Upon genotoxic treatment, however, using the same primer pairs, no transcriptional induction was observed (Figure S4B). In the *Atbrca1-3* mutant however the basal level of expression was higher than in Col-0. Nevertheless, no induction was detected and the mutant showed hypersensitivity (Figure S4C and D). We have detailed this information in the Results under "RBR represses E2FA activity to inhibit the cell death response", second paragraph.

P9 The authors argue that ATM and ATR inhibitors prevent- or greatly reduce- the formation of RBR, γ H2AX, and BRCA1 foci formation, citing fig. S3B and C. Fig S3B is on a different topic which seems to have been mistakenly left out of the text.

Apologies for this. During resubmission we reorganized the figures, now the data on ATM and ATR inhibition and foci formation is included in Figure 3C (IATR+IATM+MMC). The activity of the two inhibitors, and their effect on cell death were analysed as shown in Figure S3D.

P9 (writing) again, repair is part of maintaining genome integrity. If you mean damage induced checkpoints, say so.

We do not know the exact molecular function of RBR and AtBRCA1 together that, when compromised, leads to increased γ H2AX labeling. We do not think this relates to the cell cycle checkpoint, as *Atbrca1* mutation has no effect on S-phase progression judged by EdU labeling.

P10 "the presence of canonical E2F binding sites in the promoter region of the differentially expressed genes" perhaps some text was lost here? Did some fraction of genes have this sequence? Tell us about this.

We extended the sentence to provide the precise fraction:

"The presence of canonical E2F binding sites in the promoter region of the differentially expressed genes (53 out of 99, summarized in Table S1 column N, based on (Naouar *et al*, 2009)) suggested that RBR exerts its repressive activity through E2F proteins."

P12+13 The lack of amiRBR-induced cell death in the BRCA1 mutants is interesting, particularly given BRCA1's lack of importance in damage-induced PCD.

The authors include an image of a pRBR:GFP construct in the same root to give a feel for the degree of silencing induced by the microRNA. All the GFP images look overexposed- to me. Could we compare them on the same scale (gain) as a non-amiRBR plant? That will allow us to see if there's any silencing in amiRBR, and if it's maintained in the double mutants.

Similarly, the anti-cell death effect of the *brca1* mutations calls into question the similar effect of the E2FA mutations. Could the authors provide a control that demonstrates that the amiRBR transgene is still functioning in these "double mutant" lines?

The GFP marker in these images has misled the referee and we understand this on hindsight. We have deleted them from Figure 6C. During the generation of the *35S:amiGORBR* construct, a vector encoding also GFP marker for the presence of the transgene was used. This GFP is not a

readout of an “*ami*” sensor; GFP level doesn’t correlate with the level of silencing. We now describe in the Supplementary Information how we used the marker as a primary selection criterion and with this information we have retained the GFP images in Figure S5F.

After genotyping, we have analysed the expression of *AtBRCA1* via qRT-PCR (Figure S5D). To verify RBR silencing, we scored the well-described effect of reduced level of RBR on QC and stem cell maintenance; in addition, we counted the columella cell layers on the same individual seedlings tested also for cell death response (Figure S5F and G). EdU labelling on the homozygous F3 batches also confirmed that the *35S:amiGORBR* transgene was still functional (Figure S5H). The double mutants were genotyped, and the absence of functional *AtBRCA1* was controlled by expression studies after MM treatment (Figure S5D).

The functionality of the *amiRBR* transgene in the *amiRBR;e2fa* mutants was tested by qRT-PCR (Figure S7D).

P13. Fig 7 Independence from SOG1- just a thought and a minor correction: SOG1 isn't required for all damage-induced cell death- its only required for the very localized (around the QC) death that occurs within hours of treatment. At later times (a day or more after treatment) SOG1 (and ATM/ATR)-independent death is observed throughout the mitotic zone. There's no evidence that this is programmed.

Spontaneous cell death as a result of RBR silencing was analysed without treatment in the stem cell niche area. The referee’s thoughts are consistent with our observation, according to which SOG1 is not required in the very localised area upon RBR silencing, opposite to genotoxic treatment and stress. We have carried out all our treatments for 16 hrs (or less), for the exact reason mentioned by the referee. And indeed, we have been careful and avoided the term “programed cell death”.

Why do the authors quantify cell death only in root cap cells? I suppose because they are easy to score and represent a limited population?

We have scored the number of dead cells in the QC, columella and lateral root cap stem and daughter cells. As the proximal meristem cells are hard to count reliably, we quantified the area of PI staining inside cells in the proximal meristem which is a more rough indication of the amount of dead cells (eg. Figure S2C, S4E and S7A). In the previous submission we left out these results; now we provide this additional quantification of cell death, supporting our former observations in the Supplementary Information, “Mutants and their phenotypic analysis”.

But these particular cells spontaneously undergo cell death as the root cap is shed, don't they?

In the distal root stem cell niche where the counting was carried out no shedding-associated cell death took place, see the wild-type controls.

So the scoring might dependent on many factors, including the rate of the cell cycle, rather than DNA damage. In contrast, in the images shown the majority of the (damage-induced) death observed occurs in the younger cells of the stele. I think it is very important that the authors not mix and match this data- they always need to make it clear in the text which effect they are observing.

In summary, I think there are a number of reasonably interesting observations here, which clearly indicate that RBR1 plays a role in genome maintenance in plants as well as in mammals. But the topic addressed in the title and abstract- that RBR1 is more than a transcriptional regulator of entry into S phase, is supported only by the observation (already made in mammals) that Rb, for unknown reasons, accumulates at gH2AX foci.

Referee #2:

The retinoblastoma protein is best known, both in animals and plants, for its role in cell cycle progression as a transcriptional repressor of E2F target genes. However, it is a multifunctional protein with other less characterized functions. Here, the authors show its participation in genome integrity and DNA damage response (DDR). They find that RBR, the Arabidopsis Rb homologue, and BRCA1 are recruited to the sites of MMC-induced DNA damage, in a subset of gamma-H2A.X foci. They confirm that BRCA1, as well as other genes that participate in DDR are E2F targets. The major claim of this study is that RBR are part of a regulatory network coordinating cell proliferation and genome integrity.

This manuscript contains a valuable set of results addressing a poorly understood function of plant RBR. The manuscript contains a large amount of data but its current structure is not very clearly justified.

We incorporated new data and reorganized chapters and figures. We also focused the discussion and we hope that the revised structure now clarifies the rationale and outcomes of our study better.

A significant amount of the work is devoted to demonstrate the BRCA1 is an E2F target. Another focuses on the colocalization of RBR and BRCA1, but this is not pursued to really provide details about complex formation, dynamics and function. In my opinion, the study is of high quality but needs further experiments to provide compelling support of the conclusions, and leaves several unanswered crucial points, including a demonstration of the functional relevance and/or significance of the presence of RBR in the DNA damage foci, and of the functional coordination between cell proliferation and genome integrity. A discussion on how is RBR targeted to DNA lesions is also necessary. In addition, there are several points that need clarification and/or further experiments, as detailed below.

Some specific points.

page 7, three first lines. This conclusion needs to be explained.

This conclusion was not clearly stated, and in addition new and different experiments were carried out. Hence this paragraph has now been replaced.

2. Fig. 2. How is separated the function of RBR in the formation of punctate patterns from its role as a transcriptional regulator (in rRbr lines).

We observed nuclear foci of RBR and E2FA targeted to yH2AX positive sites only after genotoxic stress (MMC and zeocin, Figure 2). Therefore, punctation is not associated with the general transcriptional regulatory role of RBR.

3. Page 9. The RBR-BRCA1 interaction seems to be rather weak. Is this functionally relevant in vivo?

This question is hard to address due to partial synthetic lethality. To show functional relevance we generated the double mutants (*amiRBR;brca1-1^{-/-}* and *amiRBR;brca1-3^{-/-}*). The single *Atbrca1* mutants are viable and do not show any developmental effect (Reidt *et al*, 2006). Similarly, plants with moderate (*amiRBR*) or locally (*rRBR*) reduced RBR level, do develop and/or grow normally. In both mutants we followed root development in great details.

Contrast to the single mutants, in the double mutant (*amiRBR;brca1-1^{-/-}*), around half of the seedlings showed strong developmental abnormalities such as mis-positioned or missing organs and did not reach the primary leaf stage, there were not vital (Figure S5A to C); indicating that the *Atbrca1-1* mutation can aggravate the effect of *amiRBR* on seedling development. Therefore, to study the effect of RBR silencing in the mutant background, we focused on the subset of seedlings with non-aberrant, normally developed roots. These seedling were compromised as well, cell death response characteristic for *amiRBR* was suppressed by both, the *Atbrca1-1* and *Atbrca1-3* mutants.

Is this due to transient formation of a complex?

The low detection level of the interaction indeed could be due to transient complex formation.

However, we hesitate to speculate on this issue; ideally, specific point mutant in *AtBRCA1* and RBR that disrupt the interaction should address this point, but these mutants are not available.

I wonder if alternative approaches, e.g. BiFC, would provide more hints into the significance of this interaction.

Indeed. We have developed the relevant constructs and carried out the BiFC assay in young, growing tobacco leaves. Figure 4B summarises our new data, according to which RBR and *AtBRCA1* can interact physically.

How does the phosphorylation status of RBR affect complex formation and function?

To analyse the phosphorylation status of RBR with known canonical and uncharacterized non-canonical phosphorylation sites, new antibodies or/and mutants with inactive or constitutively active phosphosites are needed and we feel that this is beyond the scope of our current work.

4. Page 10. Fig. 4E. PCNA1 is used as a positive control. In that case, do fragments 1 and 3, lacking E2F sites give a positive binding?

In order to study RBR enrichment on the *AtBRCA1* promoter, the sonication was optimized with the aim to gain DNA fragments mainly between ~300~500 bps. Region 2 containing a “perfect putative” E2F site (tttggcgctttt, -151, + strand) gave always the highest enrichment. Using primer pairs 1 and 3 for PCR, on the same DNA-population (~300- ~500bp long fragments), the enrichment could be related to less frequently amplified fragments which also had the E2F binding site at position -151. In addition, region 2, at position -234 (-strand) has an E2F-like element (ggggccaaaa) as well (Figure 5D and E).

What is the profile of RBR binding after MMC challenge?

We do show the binding of E2FA upon MMC treatment. RBR exerts its repressor function exclusively through E2FA on the *AtBRCA1* promoter, shown by the expressional studies in the different *e2fa* and *b* mutants; moreover, as E2FA can suppress RBR’s DDR related phenotype, we assumed that RBR would have also weaker binding to the *AtBRCA1* promoter.

5. PAGE 11. The use of *e2fa* mutants is interesting but requires (i) a detailed characterization of the various alleles, (ii) are they knock-out or knock-down in terms of mRNA produced? (iii) could a truncated protein be produced? (iv) Can this explain some of the results obtained (dominant negative effects after sequestering DP)? (v) What is the role of an E2F truncated protein bearing a marked box?

The *e2fa-1*, *e2fa-2* (MPIZ_244, GABI-348E09, respectively) was characterized earlier (Berckmans *et al*, 2011). Using a primer combination downstream of the relevant insertions, no mRNA was detected in either of the mutants, hence they do not produce the full-sized E2FA (see Figure below). However, the authors do not exclude that truncated protein can be synthesized. In our preliminary experiments, using a specific E2FA N-terminal related antibody, indeed we could detect a specific, lower mobility protein in the *e2fa-2* mutant (data not shown). Based on this, we assume that *e2fa-1* and *e2fa-2* are reduction-of-function rather than null mutants. The truncated *e2fa-1* and *e2fa-2* allele differ in their marked-box domain but both could dimerize with DPs. Thus, we suggest that the difference in the observed phenotype does not relate to a dominant negative effect sequestering DPs, but rather, the marked box domain plays an important role. The *e2fa-3* mutant was isolated by (Xiong *et al*, 2013) and described as a null allele.

6. Page 13. Lines 2-4. Explain suppression of the cell death response in amiRBR by *brca1* mutations.

Upon RBR silencing, the transcription of the *AtBRCA1* gene is induced together with other DDR genes. The elevated level of *AtBRCA1* should contribute to repair, however RBR silencing results

in cell death due to enhanced DNA damage. As the cell death response is totally suppressed by *e2fa* mutations, we argue that the RBR/E2FA transcriptional pathway is responsible for the cell death response. AtBRCA1 is one of its targets, presumably contributing to the cell death response – thus, in its absence, the cell death response is partially suppressed.

7. Page 14. Line 5. The differences between de-repression and activation needs further justification in this context.

For the transcriptional regulation of DDR genes, RBR acts as a repressor. Upon silencing of RBR, repression is released. Hence, we prefer to use de-repression:
“we analysed whether the de-repression of DDR genes upon RBR silencing is SOG1 dependent.”

8. It is already known that E2F colocalizes to DNA damaged sites (also in animal cells). Based on this, finding RBR in those sites could be also expected.

Although the interaction of RBR and E2FA during transcriptional repression, and co-localisation of RBR and E2F at damaged sites might be expected, hitherto it was not shown experimentally. In this resubmitted version we included our new result on the co-localisation. We show that E2FA foci indeed co-localise with RBR foci after both MMC and zeocin treatment. Morphology and number of foci per nucleus as well as high degree of co-localization (Figure 2C to F) confirmed the presence of E2FA and RBR in the same foci. γ H2AX also co-localised with RBR foci, though partially and with lower degree compared to RBR/E2FA co-localisation (Figure 2A and B). We found similar co-localisation patterns shown for RBR/ γ H2AX also for RBR/E2FA with γ H2AX (Figure 2, compare A and C to F), suggesting that RBR function together with E2FA at γ H2AX positive sites of DNA damage.

Referee #3:

General comment on the two manuscripts:

The authors B. M. Horvath et al. (Scheres and Boegre labs) submitted a manuscript with the title "Arabidopsis RETINOBLASTOMA RELATED is involved in repair and DNA damage response". The authors S. Biedermann et al. (Schnittger lab) submitted a manuscript with the title "The Retinoblastoma homolog RBR1 mediates localization of the DNA repair protein RAD51 to DNA lesions". The manuscripts should be considered for back-to-back publication. While the Scheres/Boegre paper has a lot of data, the Schnittger paper is much less substantial and appears often sloppy (no NGS mRNA analysis, no co-IP interaction data; missing size bars, some statistical analysis missing). Both manuscripts emphasize that RBR not only has a function in cell cycle regulation and transcription but also a direct function in DNA repair. Both studies underpin this latter point by co-localization data between RBR, DNA damage markers (γ H2AX) and DNA repair proteins (BRCA1 or RAD51). The Scheres/Boegre study also performed an additional experiment (co-IP) to demonstrate the co-existence of RBR and BRCA1 in the same complex. Unfortunately, both studies suffer from technical short-comings related to afore mentioned key experiments (detailed evaluation below) and it remains unclear if RBR is really targeted to DNA lesions, co-localizing with DNA repair factors and if it has a direct function and not an indirect one (via control of transcription of genes encoding DNA repair proteins and cell cycle factors). The accompanying experiments (cell death studies in root tips, sensitivity assays, epistatic analyses, mRNA expression and promoter control analyses) are not discriminating between an indirect or direct contribution of RBR to DNA damage response. It is important to highlight, that a principle involvement of RBR in DDR and DNA repair is unambiguously shown in both studies. The direct involvement of RBR in DNA repair in plants has already been hypothesized earlier (in a study related to meiotic DNA repair - Chen et al 2011, EMBO J.; in a study by the Scheres lab, Cruz-Ramirez et al., 2013 PLoS Biol.) but not conclusively answered back then. Furthermore, there are conflicting data comparing the given studies and the previous Chen et al. study: now Biedermann et al. report co-localization of RBR with RAD51 in mitotic nuclei, while the previous study of Chen et al. clearly showed no co-localization of these two factors during meiosis; Horvath et al, report co-localization of RBR with BRCA1, a factor needed for DNA repair and speculate about a failure of DNA repair in the mitotic nuclei with reduced RBR levels, yet the previous study by Chen et al. did not observe any DNA repair defects (only defects in connecting to the homologous partner). None of these conflicts are

further discussed in the two given manuscripts. In this sense, the two given manuscript fail to provide a strong and non-ambiguous answer for the interesting question if RBR in plants is directly involved in DNA damage repair.

The comparison between the observation (1) that there is no DNA repair defect in the *rbr-2* mutant in meiotic cells and (2) there is co-localisation of RBR and AtBRCA1 at the DNA lesions in meristematic cells is not straightforward due to a number of issues.

In depth molecular analysis of the *rbr-2* mutant revealed that in vegetative cells full-length “wild-type” RBR mRNA and protein were present. In contrast, in meiocytes, due to aberrant RBR mRNA splicing, a truncated RBR protein was detected. This truncated RBR lacked the B-box which is considered to be important in the interaction with LxCxE motif containing proteins. Likely due to the lack of this interaction, the expression of cell-cycle genes and *AtRAD51* in the *rbr-2* mutant was repressed in buds, compared to the formerly described transcriptional induction connected to E2F activity. Upon RBR silencing, however the full-length RBR mRNA is present resulting in functional RBR albeit at a reduced level.

We show here for the first time in plants that RBR and AtBRCA1 can interact. A detailed analysis using RBR and AtBRCA1 mutants with specific amino acid changes is beyond the scope of the current manuscript. Taking the evidence from humans, we can only assume whether the truncated transcript 2 in *rbr-2* can interact with AtBRCA1 and/or the reduced level of RBR reduce the possibility for interaction. Though the LxCxE motif in most cases is essential for the RBR binding, Aprelikova *et al.*, (1999) show that Rb and HsBRCA1 interaction in human cell lines depends on a 90 amino acid long region, between 304-394 of HsBRCA1. This region contains the LxCxE sequence at 358-362 but “when the LxCxE was mutated to LxPxE or when the penta-peptide was removed by an in-frame deletion, the HsBRCA1 binding domain was still able to interact with pRb.” Moreover, using *in vitro* translated full length HsBRCA1, the authors could show binding to GST-Rb ABC (amino acids 379-928) and not GST-Rb C (amino acids 780-928), showing that the AB domain of Rb was essential to AtBRCA1 binding.

According to motif homology studies, the B domain is not present in the truncated version of RBR in the *rbr-2* mutant. Nevertheless, in the *rbr-2* mutant, DSB formation is at or near wild-type level, and despite of reduction in cross-over formation, there is no evidence reported for chromosome fragmentation indicating that DSB are eventually and efficiently repaired (Chen *et al.*, 2011).

For the above reasons, to be able to conclude that the RBR and AtBRCA1 interaction facilitate repair in either mitotic or meiotic cells, would need further experimentation. In our resubmitted manuscript we focus on their common presence at the DNA lesions and their genotoxicity-induced physical interaction in plant cells, which is at least consistent with a joint function in DNA damage response.

In principle the addressed questions and the submitted findings are interesting and the authors should be given a chance to address all raised points of criticism. Special attention should be given to data quality and the key question of RBR co-localization and co-existence in the same complex together with established DNA repair factors.

General comments on the B. M. Horvath *et al.* manuscript:

The authors B. M. Horvath *et al.* (Scheres and Boegre labs) submitted a manuscript with the title “Arabidopsis RETINOBLASTOMA RELATED is involved in repair and DNA damage response”. The authors state that their data indicates that RBR not only is involved in cell cycle control but also in safeguarding DNA integrity. This latter insight is certainly new and has not been studied in depth before in plants. It should be noted though, that in 2011, a joint paper of the Franklin and Berger labs analyzed the importance of RBR in meiosis (Chen *et al.*, 2011; EMBO J.). Not much reference is given to this study, yet certain key findings in the given manuscript appear not in line with the previous study (see details above and below).

The importance of RBR in meiosis is highlighted now in the Introduction and its role in the mitotic and meiotic cell cycles compared in the Discussion.

Furthermore, the authors emphasize that their data indicate a direct involvement of RBR in plant DNA repair, acting together with DNA repair factors, localizing to chromatin/DNA to promote

DNA repair. In mammalian cells, the direct involvement of (the mammalian homologue of RBR) pRb in DNA repair has been suggested by co-IP experiments, especially highlighted in Cook et al. (2015 Cell Rep.) with evidence of pRb interacting with proteins involved in cNHEJ (Ku70/80/DNA-Pk; in Xiao and Goodrich (2005 Oncogene) with evidence of interaction between pRB and BRCA1 and Top2...etc... Conversely, Lang et al. (2012 New Phyt.) published that in Arabidopsis E2F, a binding partner of RBR involved in transcriptional control, co-localizes with γ H2AX. This latter results would rather suggest that RBR is not directly involved in DNA repair but possibly targeted together with E2F to DNA lesion sites (to integrate the DNA damage signals and release repression of genes encoding DNA repair factors globally). Indeed the authors provide very solid data on RBR dependent DNA repair gene de-repression upon genotoxic stress - rather supporting an indirect role of RBR in DNA damage response. No doubt, it is certainly intriguing to speculate about a direct role of RBR in DDR, but the data in the literature comes from different model systems, is partly conflicting and/or not convincing. In this sense, any statement on RBR's role in plant DDR has to be very solid and beyond any doubt. Unfortunately, the authors fail to make this point (see below).

We show that similarly to the studies carried out by (Coschi *et al*, 2014), RBR/E2F complex is targeted to sites of DNA damage marked by γ H2AX, frequently localized to regions with condensed DAPI (heterochromatin, Figure S3A). We immune-labelled the centromeric histone H3 (CenH3) to demonstrate that in accordance with previous findings in animal cells, RBR/E2F can be localized with γ H2AX positive sites in pericentromeric region of chromosomes (Figure S3B).

Specific comments on the B. M. Horvath et al. manuscript:

Title:

Change title to "Arabidopsis RETINOBLASTOMA RELATED is involved in DNA repair and damage response"

We rephrased the title to:

" *Arabidopsis* RETINOBLASTOMA RELATED regulates DNA damage response independently of the cell-cycle "

Introduction:

The first paragraph introduces the topic of cell cycle regulation via RBR and relates to prior research performed in plants.

The second paragraph introduces DNA damage response but exclusively quotes 4 non-plant reviews published between 2006 and 2011. Please up-date the paragraph with newer references and include references to DNA repair in plants.

We have included (Hu *et al*, 2015) and (Cools & De Veylder, 2009) as relevant plant reviews. We find the review on DDR by (Ciccina & Elledge, 2010) excellent and have not found in the literature more recent one comparable to its quality; we prefer to continue to cite this article.

The third paragraph starts with: "Depending on whether DNA damage results in single- or double strand breaks, different pathways are induced....etc.." Please also include the cell-cycle state as a determinant of DNA repair pathway choice. Furthermore, please include in the third and fourth paragraph plant-specific references when introducing "ATM" and "ATR". The reference to the review "Ciccina and Elledge, 2010" appears 4 times in the introduction....please find more appropriate and further references.

We followed the referee's advice and have included the relevant references and cell-cycle states.

With respect to ATM and ATR activation: the stimuli are dsDNA breaks (ATM) or free ssDNA (ATR)....pl. correct. (Certainly it is true that stalled replication forks may lead to ssDNA and activate ATR, but the context demands to name a more specific elicitor....). It may also be interesting to mention in this context that in mammalian cells there is extensive evidence for activation of ATM by oxidative stress and that in the context of cell cycle regulation/HU stress this has also been

proposed for plants (Yi et al 2014)...pl include these findings and according references.

Also these suggestions were incorporated .

Page 4:

"The ATM- and ATR activated WEE1 kinase mainly controls the replication checkpoint (Cools et al, 2011, Dissmeyer et al, 2009),..."

Please provide reference/evidence for WEE1 being ATM/ ATR activated...the given references are not including this information.

Here we cite the more relevant study by (De Schutter *et al*, 2007).

Page 5:

Pl. rephrase the entire last paragraph; first, as outlined above/below, the statements seem not justified; second the wording itself is problematic:

"..(i) sensing the DNA damage foci, where RBR can function together with AtBRCA1 in repair,..." . what is meant with "sensing the DNA damage foci"? This does not make sense. Furthermore, that "RBR can function together with AtBRCA1 in repair" is certainly not shown in the manuscript.

The entire paragraph is rephrased, suggestions are taken into account.

Results:

The first paragraph introduces the used RBR RNAi lines. Please clearly describe the lines used in the given study in relation to the ones used by Chen et al. (*rbr-2*) and by Borghi et al. (RBRi) so that in the "discussion" the discrepancies in results can be discussed.

A major focus of our manuscript is to study the spontaneous cell death response upon RBR silencing. Related to the above mentioned *RBRi* line, no cell death response was reported; the study by Borghi *et al*, (2010) focused on the effects of conditional reduction or loss of RBR during shoot development (RBRi) and no phenotypic changes were mentioned related to root development. For the *rbr-2* mutant allele neither hyper-proliferation nor cell death response in vegetative cells was described by Chen *et al*, (2011). Molecular analysis revealed that in vegetative cells wild-type RBR mRNA and protein were present. In contrast, in meiocytes, due to aberrant RBR mRNA splicing, a truncated RBR protein was detected. This truncated RBR lacked the B-box which is considered to be important in the interaction with LxCxE motif containing proteins, as E2Fs.

Based on this, we have to conclude that neither experimental systems, mutants or data are directly comparable with ours. For this reason we did not describe these mutants in the Results. In the Discussion, however, the role of RBR in meiosis is discussed.

Figure 1D: why is the signal of γ H2AX not overlapping the DAPI signal 100%? This signal should not be outside the DAPI-pos. area. Is the signal specific? Pl provide explanation or better picture.

Following this and other suggestions we have re-organised the content of Figures and provided new images.

We agree that the γ H2AX labelling presented in the earlier submitted version was not optimal; we modified our labelling protocol, tested different batches of cytohelicase and secondary antibodies. We performed wavelength correction for all images. New images are incorporated for γ H2AX labelling in Col-0, *amiRBR* without and upon MMC as well as zeocin treatments (Figure 1 to 3; Figure S2D and S3A). In addition, we also immuno-labelled Col-0(*CYCD3.1OE*) for γ H2AX (Figure 1F). In conclusion, the old images are replaced by higher quality illustrations showing γ H2AX labelling in Figure 1 to 3; Figure S2D and S3A to C. For further improvements see the following points.

Figure 1F: the *amiRBR* EDU pos/neg. cells are not shown (Suppl. Fig 1 shows only the *rRBR* data; overlay is also missing: pl. provide). Please provide the data and show channels individually and as an overlay.

Whole mount EdU labelling of root meristems from Col-0, *amiRBR*, Col-0(*CycD3.1OE*) lines was performed; data now presented in Figure 1A show channels individually and includes the overlay as well. The *rRBR* related images in Figure S6 are not altered, they serve to illustrate the properties of the material used for micro-array experiments.

In accordance with the data of - among others - (Ross *et al*, 2011) and based on our own experience finding that γ H2AX signal accumulates upon the incorporation of base analogues, such as EdU and BrdU, in the resubmitted version, we only present γ H2AX labelling in non-EdU treated material.

Page 7:

"...of this category also increased in *amiRBR* (Figure 1F), again indicating that RBR requirement to prevent DNA damage can be uncoupled from the restriction of S-phase entry." This statement is most likely not true, since the EdU pulse has been performed for 15min only (unclear when cells have been fixed - pl. provide experimental details) and the DNA damage from a prior S-phase may still be retained and yield a cell that is EdU negative but γ H2AX positive. Indeed, accumulation of DNA damage due to premature cell division without completing S-phase is very likely to occur in the constitutively silenced *amiRBR* transgenic plant line.

We agree with the reviewer; this experiment and conclusion was deleted.

Page 7:

"Next we studied whether lines with RBR silencing were more sensitive to genotoxic stress and found elevated γ H2AX labeling..."

Pl. rephrase: elevated γ H2AX foci are not measuring sensitivity to genotoxic stress...

We have rephrased this sentence: "To visualise early responses to DNA damage we followed the accumulation of the phosphorylated H2AX (γ H2AX) histone variant."

"The accumulation of DNA damage foci in the RBR silenced meristematic cells could indicate that, like Rb in animal cells, plant RBR maintains genome integrity.." Pl. provide according pictures and display together with the graph shown in Fig. S2A.

This sentence and conclusion was deleted. We have rephrased this chapter and incorporated new data.

The authors find more cell death and more cells with γ H2AX foci after 16h MMC treatment and conclude that DNA damage may accumulate due to compromised RBR function. The reviewer wants to emphasize that the most consistent interpretation would conversely be that in the *amiRBR* or *rRBR* lines pre-mature S-phase/cell division take place. This is concluded from the 16h time window of MMS treatment, and also from the fact that RBR depletion has been correlated (by the very same authors) with pre-mature entry into S etc... Furthermore, it should be noted that MMC acts as an intrastrand crosslinker. Only during S-phase (!) the DNA intrastrand crosslinks will compromise replication and need to be repaired and given sufficient time to do so. This has to be considered and integrated.

Besides MMC, we also used zeocin and HU to study the effect on cell death upon RBR silencing in two different RBR mutant lines. Our results were formerly shown in Figure S2 and S9. In response to the referee's advice we have extended our experimentation and summarised the data as a first section in the paragraph "DNA stress recruits RBR together with E2FA to γ H2AX-labelled heterochromatic foci". We did not observe any difference in cell death response upon treatment with genotoxic agent causing DNA intra-strand crosslinks or double strand breaks (Figure S2A to C). In contrast, HU treatment did not result in hypersensitive reaction (Figure S10).

The effect of zeocin (3 and 16 h) and MMC (16h) on RBR foci was compared also. Figure 2A shows that zeocin similarly to MMC also triggered RBR focus formation, showing that focus formation is independent of the nature of DNA damage caused upon different genotoxic agents.

In addition, we also studied both the cell death response and the presence of γ H2AX foci in the Col-0(*CYCD3.1 OE*) line. Although the set of transcriptionally induced genes related to DDR and replication overlaps (Table S1) and the number of EdU labelled nuclei was elevated to the same

extent in *amiRBR* and *Col-0(CYCD3.1 OE)* lines (Figure 1A and C), the level of γ H2AX foci formation was different in these two mutant lines (Figure 1E and F). Taking into account that the *Col-0(CYCD3.1 OE)* did not develop any cell death response (Figure 1B and D) despite of elevated S-phase entry, and had significantly less γ H2AX focus containing nuclei, we argue that the above consequences are the result of a different compromised RBR-related function. While both RBR and *CYCD3.1* control the S-phase checkpoint, RBR has an additional role in DNA damage response.

Figure 2A

16h MMC...see comment above.

Cytology of somatic nuclei exposed to MMC:

RBR localizes only as 1-4 large foci per nucleus but only in 17% of all cells (what about the remaining 83%(!!)?). The foci areas take up about 1 micrometer in width, which is about 25% of the entire width of the somatic nuclei. Similarly, the γ H2AX foci reside as few, large foci in the nuclei (the picture provided shows 4 foci) with only about 20% of all cells showing labeling altogether (Fig. S2A) (what about the other 80%?). Interestingly, the observed RBR and γ H2AX foci appear side-by-side but not overlapping, and also side-by-side with intensely DAPI-stained bodies. The authors also provide a graph of measured fluorescence intensity in the respective channels, to underline their statement of co-localization.

We show that RBR and E2FA co-localise in foci formed upon MMC or zeocine treatment. Focus formation was shown to be ATM and ATR dependent (Figure 2F). The percentage of nuclei having RBR foci is in a very good agreement with the data published for E2F foci formation by Lang et al. (2012). Co-localisation studies confirmed partial co-localisation of RBR foci with γ H2AX positive sites (Figure 2A, B and E) and co-localisation with E2FA (Figure 2C to F). Examining numerous nuclei, we observed that 80 % of the RBR foci localized side-by-side with intensely DAPI-stained condensed chromatin (Figure S3A).

The experimental section does not explain how the pictures are acquired: are these single stacks or are these (max. intensity?) projections? Why not performing a 3D re-construction with the (most likely) available z-layers. How is co-localization defined? Please provide a definition? Are these foci in the same z-level? Has the picture acquisition been done in a manner that wave length shifts has been considered (which could work potentially in favor of the author's arguments, since weirdly, the foci seem to cluster but not overlap)? Furthermore, to argue for co-localization (according to a definition yet to be provided) a statistical test and a comparison to a random situation is needed. Preferentially this test should be done in 3D (and not on a projection!) using the actually measured nuclei volume, exclude the volume of the nucleolus and use the average size of the foci volumes.....

We performed wavelength correction for all images shown in the resubmitted manuscript. In addition, Huygens deconvolution, 3D reconstruction, Imaris analysis and colocalization studies using JaCoBs ImageJ functions with statistics were applied.

We have extended our studies on RBR co-localisation with E2FA in foci formed upon genotoxic stress MMC and zeocin treatment. RBR and E2FA show strong colocalization into the same foci as confirmed by Imaris analysis after 3D reconstruction (Figure 2C to F) and by statistical analysis on co-localisation (Figure 2E). RBR and γ H2AX co-localisation was observed and shown from independent experiments (Figure 2A and B). As the images illustrate RBR foci co-localised with the γ H2AX signals; partial colocalization was shown by Imaris analysis after 3D reconstruction (Figure 2E). The degree of co-localization is lower compared to the co-localisation of RBR and E2FA signals (Figure 2E). The broader range of Pearson coefficient found for RBR and γ H2AX colocalisation is typical for dynamic transient interaction.

Please include in your analysis and discuss the adjacent DAPI-stained bodies: are these centromeric regions? The reviewer points out that Coschi et al. 2014 (Cancer discovery; not a plant study) found a protein complex associating with pericentromeric repeats comprised of E2F1, condensin and pRb. If this is also true in plants, the nature of the presented staining (a few massive foci of RBR) would be in accordance with previous findings in mammalian cells.

To analyse whether the RBR and DAPI-stained bodies co-localise we used the pericentromer-specific histone antibody (Histone H3, CenH3). Figure S3B illustrates that RBR and CenH3 foci indeed co-localised – although CenH3 is rarely detected in MMC treated plants compared to

centromeres of untreated controls. Figure S3A shows different examples of the co-localisation of the RBR foci and DAPI-stained bodies.

In general, the low amount of cells that show a staining altogether after 16h of MMC treatment and the low amount of γ H2AX foci in those few cells may reflect different technical short-comings: e.g. limited MMC stability and/or penetration; over-fixation of cells/proteins; limited permeability for antibodies to enter the cells/nuclei during the staining procedure...etc...

Before and during the re-submission, different labelling protocols were applied which all gave the same results related to immune-labelling. All experiments were repeated at least 5 times resulting in similar focus formation properties for RBR and γ H2AX.

As mentioned above, we performed new experiments partly to compare MMC and zeocin treatments. Both genotoxic agents resulted in similar RBR and E2FA co-localization in foci in an ATM/ATR dependent manner (Figure 2A and C for MMC), and co-localization with γ H2AX foci upon MMC and zeocin (Figure 2F). The high degree of E2FA and RBR co-localisation and partial co-localisation with γ H2AX suggests that specific labelling pattern indeed reflects targeting of E2FA and RBR to the sites of DNA damage visualized by γ H2AX. Moreover, the RBR and E2FA foci visualised are identical in number to the E2F foci upon bleomycin treatment described previously by (Lang *et al*, 2012).

Based on this, we are confident about our experimental conditions and results.

Please re-do and extend the analysis and re-write the paragraph accordingly.

The experiments were repeated and the related paragraph was rewritten accordingly.

In any case, please delete the following sentence:

"These observations supported a role for RBR in DNA damage sensing without excluding the possibility that in addition RBR protects DNA from damage."

This sentence is deleted.

How can these experiments possibly probe for RBR (or any other protein) sensing DNA damage? The experiment just probe for (co-)localization of proteins in cells. Phosphorylated γ H2AX is a downstream consequence of a prior detection of DNA lesions - co-localization would merely indicate that the two proteins reside at the same locus. Moreover, the peculiar nature of the γ H2AX foci make it uncertain if these foci are actually representing DNA damage. One way to test this would be to elevate the concentration of MMC, to see a concentration dependency of the foci numbers and the number of pos. cells. Please consider that MMC will only lead to DNA damage in S-phase and therefore an accumulation of DNA damage in amiRBR or rRBr upon MMC treatment is weakening the argument that RBR may act directly on DNA lesions and not indirectly by regulating cell cycle and transcription (as elegantly shown in the following parts of the given manuscript). Using a different drug, like bleomycin or a derivative, that chemically introduces ssDNA and dsDNA lesions irrespective of the DNA replication may be helpful, especially if co-treated with EdU to filter for those cells that have not undergone replication during the experimental window.

The authors probe for BRCA1, mentioning its pivotal role in mammals, but not mentioning its modest importance in plants (please include information and reference to work from H. Puchta).

Thank you for pointing out this omission. Studies by Puchta's group are included (Block-Schmidt *et al*, 2011, Trapp *et al*, 2011).

The authors argue for "up-regulation" of AtBRCA1:GFP in the meristematic cells of the rRBr line but more AtBRCA1:GFP foci could also represent more DNA damage after pre-mature S-phase entry and therefore more focused localization of AtBRCA1:GFP...etc....

The AtBRCA1-GFP protein is present only in restricted number of cells upon RBR silencing, likely due to damage either as a result of premature S-phase entry or genome instability. We do not have

data whether these cells are in G1/S-phase.

We expect, that if the AtBRCA1-GFP signal would be simply the result of silencing of RBR driven de-repression, it would appear equally dispersed and at low level in the majority of the meristematic cells; we did not observe this phenomenon. Based on the referee's suggestion the related description was rephrased: "Notably, silencing of RBR in the meristematic zone also resulted in the accumulation of AtBRCA-GFP localised in nuclear speckles in and around the stem cell niche (Figure 3B).

Again, it would be interesting to test G1 cells after bleomycin treatment, and see if presence or absence of RBR makes a difference for the observed "speckles".

Unfortunately, at the time of preparing this manuscript, *in planta* we could not differentiate/separate cells at different cell cycle stages. BY-2 tobacco cells could be used, but to synchronise cells, one need to apply aphidicolin. This drug is known to cause DNA damage. During the preparation of this resubmission, Yokoyama *et al*, (2016) published data on the dynamics of replication using the pgPCNA1-GFP marker line; with the help of which one could distinguish S-phase cells in the future, however we feel that it is beyond the scope of this resubmission.

The co-localization data suffers from the same shortcomings in technical implementation and analysis as outlined above. The quality of the cytological analysis presented in Figure 2D is poor. It is very hard to decide if the shown images represent genuine staining, technical artefacts or background (compare the different classes of γ H2AX staining: the authors believe they are meaningful, the reviewer is not convinced.) The low staining efficiency in general (see above) and the low quality of the pictures make it impossible to decide if these pictures are meaningful.

Please provide better data/pictures also for the BRCA1 analysis.

We performed new experiments and provided new images for the AtBRCA1/RBR/ γ H2AX localisation (Figure 3C). Wavelength shift was corrected on the images from the previous submission and presented in original quality concerning brightness/contrast/colour mode.

The experiment depicted in Figure 3A, aiming to show direct interaction between RBR and AtBRCA1 is not convincing at all. Please provide more convincing interaction data and ideally corroborate with a different experimental strategy.

Indeed. We have developed the relevant constructs and carried out the BiFC assay in young, growing tobacco leaves. Figure 4B summarises our new data, according to which RBR and AtBRCA1 can interact physically.

The experiment depicted in Figure S3C also suffers from the picture quality and the unclear interpretation of foci. Are they specific or background? Why is the partial lack of DAPI co-localization with γ H2AX shown in Figure 1 ignored but seriously interpreted with respect to RBR and DAPI in Figure S3C. Again, the settings of the microscope, the wavelength shift, the 3D vs the projection issue....etc... All this prohibits drawing convincing conclusions from these data.

As described earlier new images are shown in each of the microscopical and immuno-co-localisation -related studies. Former Figure S3C was deleted.

The following sentence does not make sense: "If RBR function was solely required to maintain genome integrity, while AtBRCA1 was involved in repair....", pl. re-phrase.

The entire paragraph was rewritten, and now it includes further data on the phenotypic features of the *amiRBR;Atbrca1-1* mutant. It starts as:

"To study whether RBR and AtBRCA1 may have a function together, we followed the genetic interaction of *amiRBR* and *Atbrca1-1* cross in the homozygous *amiRBR;Atbrca1-1* line."

Page 12:

AtBRCA1 genepl correct
Done.

Page 13:

Please give an explanation why the *brca1-3* mutant line has been introduced.

Atbrca1-3 allele was introduced in order to strengthen our results on the *Atbrca1-1* and on the *amiRBR;brca1-1* mutants phenotype. From the expressional analysis, it seems that *Atbrca1-3* allele has a truncated, non-functional version of *AtBRCA1* mRNA; however its transcription is not inducible via genotoxic stress (Figure S4B).

Both *Atbrca1* alleles developed same reactions upon genotoxic stress also studying the cell death response (Figure S4C to E), and similarly partially suppressed the cell death developed in the *amiRBR* background (Figure 6C and D).

Page 13:

While it is interesting to learn about the (partial) suppression of the *amiRBR* cell death phenotype by *brca1* mutants it would also be interesting to learn about survival and fitness of the entire plant.

After crossing, in the F3 generation the segregation of the “healthy” vs “ill or deformed” plants was 42/47 (Figure S5A to C). The development of the “ill” plants in some cases did not reach the 2-cotyledon stage. In contrast, the healthy category grew normally, reached maturity, and could be tested for the presence of the silencing construct carrying the GFP marker and for the loss of *AtBRCA1* function (Figure S5D). In this category, at normal growth conditions, the homozygous lines were similar to the parents. RBR silencing in the double mutants was followed by analyzing the phenotypes of the same individual seedlings also studied for cell death response. Thus QC and stem cell meristem maintenance was followed by quantifying cell layers in the distal meristem; columella stem, daughter and differentiated cells and quantifying S-phase entry on the same batch of seedlings (Figure S5E to H). The loss of *AtBRCA1* gene was controlled by its MMC-inducibility ((Figure S6D).

Page 13:

"We detected a remarkable overlap between genes induced upon genotoxic stress in an ATM/SOG1 dependent manner and..." Pl correct to "We observed..." and furthermore change the word "remarkable" to "significant", if true ...if so, then pl provide the numbers (statistical test against random...etc..!)

We observed significant overlap between genes annotated to DNA repair and having defect in their induction by irradiation in the *sog1-1* mutant and genes de-repressed upon RBR silencing (Table S2, DDR related genes: 8 *sog1-1/10 amiRBR*).

Discussion:

The Discussion was reorganized in order to emphasise the cell-cycle dependent and independent functions of RBR during DDR. Accordingly, our findings are discussed in the “RBR mediated DNA damage control at γ H2AX foci” and “RBR mediated transcriptional response to DNA damage”.

Pl rephrase sentence: "Cells are protected from DNA damage by imposing quiescence..."

This sentence was deleted.

Pl revise sentence after additional data has been added:

"We show that upon genotoxic stress RBR accumulates in large nuclear foci, which significantly overlap with DNA damage sites labelled for γ H2AX....". Define "overlap" and "co-localisation", give values for "significant"....

In response to the referee's comment, we avoid to use different expressions. Sentence is deleted and rewritten.

“We reveal that RBR is homogenously distributed in somatic meristematic nuclei but accumulates in a limited number of large nuclear foci upon genotoxic stress, which co-localise with DNA damage sites labelled for γ H2AX and with E2FA.”

Delete the following sentence:

"During homologous recombination of meiotic chromosomes, RBR is recruited to DSBs,".....since it is not clear if true (certainly they do not coincide with RAD51 or DMC1).

This sentence was rewritten:

Also consistent with non-transcriptional roles for RBR is the finding that, during meiosis, RBR is recruited to the chromosomes in a DNA DSB-dependent manner, where it was suggested to facilitate the assembly of chromatin modifiers, repair proteins and condensin complexes for homologous recombination through their LxCxE motifs (Chen *et al*, 2011).”

Page 18:

PI rephrase last paragraph according to the new data. The following statement needs re-phrasing in any case: "(i) protection and sensing of DNA integrity".

In conclusion, RBR, mainly known as a regulator of cell cycle and asymmetric cell division in plant meristems, is also involved in maintaining genome integrity in these growth zones through two functions, (i) assembly at a limited number of H2AX foci together with E2F and, possibly, AtBRCA1; (ii) transcriptional regulation of important DDR genes including *AtBRCA1*. It will be interesting to investigate in the future whether and how assembly of E2F-RBR complexes at particular γ H2AX foci is coupled to the transcriptional role of these complexes in DDR gene regulation.

References

Aprelikova ON, Fang BS, Meissner EG, Cotter S, Campbell M, Kuthiala A, Bessho M, Jensen RA, Liu ET (1999) BRCA1-associated growth arrest is RB-dependent. *Proc Natl Acad Sci U S A* 96: 11866-71

Berckmans B, Vassileva V, Schmid SP, Maes S, Parizot B, Naramoto S, Magyar Z, Alvim Kamei CL, Koncz C, Bogre L, Persiau G, De Jaeger G, Friml J, Simon R, Beeckman T, De Veylder L (2011) Auxin-dependent cell cycle reactivation through transcriptional regulation of Arabidopsis E2Fa by lateral organ boundary proteins. *Plant Cell* 23: 3671-83

Block-Schmidt AS, Dukowic-Schulze S, Wanieck K, Reidt W, Puchta H (2011) BRCC36A is epistatic to BRCA1 in DNA crosslink repair and homologous recombination in Arabidopsis thaliana. *Nucleic Acids Res* 39: 146-54

Borghi L, Gutzat R, Fütterer J, Laizet Y, Hennig L, Gruissem W (2010) Arabidopsis RETINOBLASTOMA-RELATED is required for stem cell maintenance, cell differentiation, and lateral organ production. *Plant Cell* 22: 1792-811

Chen Z, Higgins JD, Hui JT, Li J, Franklin FC, Berger F (2011) Retinoblastoma protein is essential for early meiotic events in Arabidopsis. *EMBO J* 30: 744-55

Ciccia A, Elledge SJ (2010) The DNA damage response: making it safe to play with knives. *Mol Cell* 40: 179-204

Cools T, De Veylder L (2009) DNA stress checkpoint control and plant development. *Curr Opin Plant Biol* 12: 23-8

Coschi CH, Ishak CA, Gallo D, Marshall A, Talluri S, Wang J, Cecchini MJ, Martens AL, Percy V, Welch I, Boutros PC, Brown GW, Dick FA (2014) Haploinsufficiency of an RB-E2F1-Condensin II complex leads to aberrant replication and aneuploidy. *Cancer Discov* 4: 840-53

- Cruz-Ramirez A, Diaz-Trivino S, Wachsman G, Du Y, Arteaga-Vazquez M, Zhang H, Benjamins R, Blilou I, Neef AB, Chandler V, Scheres B (2013) A SCARECROW-RETINOBLASTOMA protein network controls protective quiescence in the Arabidopsis root stem cell organizer. *PLoS Biol* 11: e1001724
- De Schutter K, Joubes J, Cools T, Verkest A, Corellou F, Babiychuk E, Van Der Schueren E, Beeckman T, Kushnir S, Inze D, De Veylder L (2007) Arabidopsis WEE1 kinase controls cell cycle arrest in response to activation of the DNA integrity checkpoint. *Plant Cell* 19: 211-25
- Dewitte W, Riou-Khamlichi C, Scofield S, Healy JM, Jacquard A, Kilby NJ, Murray JA (2003) Altered cell cycle distribution, hyperplasia, and inhibited differentiation in Arabidopsis caused by the D-type cyclin CYCD3. *Plant Cell* 15: 79-92
- Hu Z, Cools T, De Veylder L (2015) Mechanisms Used by Plants to Cope with DNA Damage. *Annu Rev Plant Biol*
- Lang J, Smetana O, Sanchez-Calderon L, Lincker F, Genestier J, Schmit AC, Houlne G, Chaboute ME (2012) Plant gammaH2AX foci are required for proper DNA DSB repair responses and colocalize with E2F factors. *New Phytol* 194: 353-63
- Naouar N, Vandepoele K, Lammens T, Casneuf T, Zeller G, van Hummelen P, Weigel D, Ratsch G, Inze D, Kuiper M, De Veylder L, Vuylsteke M (2009) Quantitative RNA expression analysis with Affymetrix Tiling 1.0R arrays identifies new E2F target genes. *Plant J* 57: 184-94
- Reidt W, Wurz R, Wanieck K, Chu HH, Puchta H (2006) A homologue of the breast cancer-associated gene BARD1 is involved in DNA repair in plants. *EMBO J* 25: 4326-37
- Riou-Khamlichi C, Huntley R, Jacquard A, Murray JA (1999) Cytokinin activation of Arabidopsis cell division through a D-type cyclin. *Science* 283: 1541-4
- Ross HH, Rahman M, Levkoff LH, Millette S, Martin-Carreras T, Dunbar EM, Reynolds BA, Laywell ED (2011) Ethynyldeoxyuridine (EdU) suppresses in vitro population expansion and in vivo tumor progression of human glioblastoma cells. *J Neurooncol* 105: 485-98
- Trapp O, Seeliger K, Puchta H (2011) Homologs of breast cancer genes in plants. *Front Plant Sci* 2: 19
- Xiong Y, McCormack M, Li L, Hall Q, Xiang C, Sheen J (2013) Glucose-TOR signalling reprograms the transcriptome and activates meristems. *Nature* 496: 181-6
- Wang, S. *et al.* A noncanonical role for the CKI-RB-E2F cell-cycle signaling pathway in plant effector-triggered immunity. *Cell Host Microbe* 16, 787-794, doi:10.1016/j.chom.2014.10.005

2nd Editorial Decision

30 January 2017

Thank you again for your patience during the re-review of your manuscript on Arabidopsis RBR involvement in DNA repair. We have now received the below comments from two of the original referees, and given that they consider the key experimental issues for the most part satisfactorily addressed, we should in principle be happy to proceed further with eventual publication in The EMBO Journal. Nevertheless, the referees (in particular referee 2) still point out a number of substantial presentational shortcomings that would need to be significantly improved prior to acceptance. I am therefore returning the manuscript to you once more for a final round of revision, inviting you together with your (senior) co-authors to carefully go through the reports and to determine how the manuscript should be reorganized and streamlined. As indicated by the reviewers, the manuscript is currently very difficult to understand even for experts, and for a broad general journal such as The EMBO Journal, it will be essential to make it more easily accessible.

In this respect, I encourage you to study our revised guidelines for supplementary materials <http://emboj.embopress.org/authorguide#expandedview> which offer a number of hierarchical layers

for structuring your study so as to be attractive for more general readers (main text and figures), expert readers (up to five extendable "Expanded View" figures with legends in the main text, as well as other Expanded View items), and specialists looking for specific details (supplementary material including additional figures and their legends, tables, material and methods, references etc.; headed by a brief Table of Contents).

REFeree REPORTS

Referee #1:

RBR1 is known to play a critical role in progression into S phase- acting as an inhibitor of E2F-dependent gene expression until all conditions are correct for cell cycle progression. These conditions include the absence of detectable DNA damage. Here the authors present data indicating that

1) spontaneous cell death occurs among the mitotic cells of a line partially defective in RBR1 function- as expected for a line defective in a cell cycle checkpoint. This death is described as SOG1-independent (ATR and ATM's effects are not investigated), at least for the very specific cell populations scored (which suggests this death may be mitotic, rather than programmed cell death). Suppression of cell cycle progression is shown to suppress cell death, also consistent with the notion that unchecked progression is producing dead cells (regardless of whether this is programmed). So this result is not very surprising. However, enhancement of cell cycle progression (which is monitored as EdU incorporation) though other modifications does not reproduce this enhanced cell death, suggesting that RBR is doing something that protects the cells from death- something beyond slowing the cell cycle. This is interesting.

2) gH2AX foci are found to be enhanced in an *rbr* line, and enhanced to a lesser degree in the CYCD3.1 line, I think this difference may be a little overinterpreted, its not as striking as the cell death data, and sort of makes the point that hasty progression through the cell cycle does produce DSBs- this is not an effect specific to RBR. Not surprising.

3) The authors demonstrate that RBR can colocalize (sometimes) with gH2AX, and with and with BRCA1. It is generally found at these foci with EF2A. This is consistent with observations in animal models. Focus formation is ATR/ATM dependent, suggesting it has something to do with damage detection or repair.

4) The authors identify transcripts up- or down-regulated in an *rbr* suppressed line. The majority of these have a canonical E2F binding site "in the promoter region" (please let us know if this is statistically significant- I don't know how long this sequence is). This suggests they might be directly regulated by an E2F/RBR complex. This is followed up for BRCA1 only via ChIP. Both E2FA (specially) and the *rbr* inhibited mutant derepress BRCA1- consistent with the notion that RBR is acting as a classical repressor protein at this promoter. Although we know that RBR acts as a transcriptional repressor, this is the first indication, at least in plants, that a DDR gene is a target. However, the induction in *rbr* is small- not at all on the scale observed in response to DNA damage. DDR-related and (SOG1-dependent) induction and RBR suppression appear to be unrelated pathways. Which is interesting- there are two different forms of regulation for this gene.

The current title is not quite right, as I'm sure RBR also regulates genomic stability via the cell cycle- this suggests it doesn't do this. I think the authors mean to say RBR has additional functions beyond cell cycle regulation?

The comments below are not major and mostly have to do with writing.

DDR is well-introduced but cell cycle regulation is not. The effects of CYCD3.1 OE on, for example, cell death are described without telling the reader what CYCD3.1 does or what we might expect. It is nice that the authors have two ways of suppressing RBR expression, but it is (a little) troubling that only one suppression method is used for each type of experiment. I would really appreciate more of an introduction to each of these suppressed lines. One is expressing a micro RNA, but driven by 35S. Would we expect expression in the root meristem? Or not? The other is expressing an RNAi, but only in "the root meristem". Do you mean the entire meristematic zone? And the effects somehow don't spread beyond the zone of expression? I realize these mutants are

published elsewhere, but it would save the reader a lot of time if you could discuss them briefly here.

Top of p9 2nd paragraph: CENH3 is a constitutive component of functional centromeres- it is present throughout the cell cycle, not just during M phase (it is generally not present at high levels in cells that have permanently exited the mitotic cycle). I have no idea what the authors are trying to get at in this paragraph- why they did this or what this result means to them. I'd drop this pointless paragraph and the figure that goes with it.

Bottom p 9- give the reader a little help- tell us what BRCA1 phenotype is complemented by the GFP fusion

Top of p12- is RBR up or downregulated? Please make this clear. Similarly, at the end of the paragraph, I think you mean AtBRCA is upregulated in the rBRr mutant, rather than upregulated by RBR1. Please make this clear.

Top of p 113- remind the reader you're talking about data for animals- not clear (yet) that the same sequences are important for apoptosis in plants.

1st paragraph p 15- figure 5F does not show this data. I don't know where it is- perhaps in the supplement?

Last paragraph p 15:

Fig. 8B needs a key, so we know what the difference is between the striped, solid, and white columns.

More general comments on this section: sog1 mutants do not completely suppress clastogen-induced cell death- they suppress the death that occurs early after treatment, which is focused around the QC in the stem cell niche. Cell death also occurs after clastogen treatment in the absence of SOG1, but it occurs later, throughout the mitotic population of the RAM. Interestingly, very little amiRBR cell death occurs in this SOG1-dependent pattern, most of it occurs below the QC in the "extra" columellar layers, and what does occur above the QC is not encircling the QC (like SOG1-dependent death)- consistent with this being a different pathway. I don't know if the authors want to discuss this- just a thought. Similarly, the pictures in 6C suggest that brca1 is defective in amiRBR spontaneous cell death specifically in the proximal cell files- there's plenty of death elsewhere. Of course I have very few images to look at, I may be over-generalizing.

Referee #2:

This is revised version of a manuscript entitled "Arabidopsis RETINOBLASTOMA RELATED regulates DNA damage response independently of the cell-cycle". Authors have made a significant effort to address the points included in my previous report, although in some cases with not too much success. Overall, however, the main conclusions are supported by the data presented: (1) RBR1 has a direct role in DNA damage response (DDR), (2) this is cell cycle-independent, (3) RBR1 colocalize with BRCA1 at damaged sites (containing H2A.X), (4) BRCA1 expression depends (primarily) on E2FA in the absence of DNA damage and on SOG1 after DNA damage.

From a formal point of view, I expected to find in this revised version a thorough modification of the display items and reorganization of the text, which is extremely difficult to follow. I found that the revised version is made unnecessarily complex, mixing together sets of (unrelated) data, and combining descriptions of data presented in different figures. The logic of data organization and description is difficult to understand. The following comments illustrate this:

- Panels Fig 4A,B describes RBR1-BRCA1 interactions and this is not significantly related to data in Fig 4C that refer to H2A.X in brca1 and brca1,amiRb mutants, which in turn are also used in Fig. 6C to describe cell death. This forces to move back and forth in the manuscript instead of having an integrative description of the topic.
- Results in Fig 6A are largely coincident (repeated?) with those in Fig S7B, which contains the

(small) addition of e2fb mutant

- Measurements of BRCA1 expression with and without MMC treatment appear in Fig 6A and 7C, making again difficult the comparison
- Any reason to justify that Figs 6B and 6D are separated in two different panels?

Therefore, I am convinced that the manuscript will significantly benefit from a profound reorganization of the display items and a clarification of the text. I understand that the authors are tempted to include the lot of results that they have generated, but doing that clearly damages the clarity of the manuscript and the strength of the message. In some cases it is clear that those (extra) results are contributing nothing or very little to support the main message:

- Table 1 with the expression data needs a description of the symbols used and the conclusions derived from the comparison made. Otherwise this information is not mentioned/used/discussed anywhere else in the manuscript.
- Suppl Table S5. What's the relevance of this study on DREAM for the main message of the work relating RBR1 and DNA damage response?
- Suppl Fig S6. The phenotype of rRBr (cell division) is published, and the cell death phenotype is already shown in Suppl Fig S2A.
- I quote "The transcriptional changes in a set of genes representing different functional categories were confirmed by qRT-PCR in rRBr root-tips (Figure 5B, C) and amiRBR seedlings (Figure 7C, Figure S7B and E)". In the absence of a justification for choosing these genes (which are not specified), this information is useless. This sentence also illustrates the mixed description that I referred to above (a qRT-PCR validation in three panels of main figures and two panels of a Suppl Fig.).

Very briefly, regarding the authors' point-by-point reply to my comments.

- Explain the phenotype of the double amiRBR,brca1 mutant (growth arrest).
- Authors would agree that their answer to my point on the ChIP experiment is poor. Fig 5E shows very similar values for the positive control (PCNA1), for fragment 1 containing E2F sites and for fragment 3 lacking them. Based on data available is likely that E2F is bound to the BRCA1 promoter but certainly the results in 5E are not very clean.

2nd Revision - authors' response

20 February 2017

RESPONSE TO REFEREES

We appreciated the referees' useful suggestions for our revised manuscript **EMBOJ-2016-94561R**. We have done our best now to rearrange both text and figures so that the 'Results' section reads more smoothly. The 'Results' section has been extensively re-written to better clarify the logic of our experimentation. By rearranging the figures, we synchronised their description in the text. Now, it follows exactly the figure order. In this re-written revision, we specifically addressed the following issues raised by the referees.

Referee #1:

1. Significance of E2F sequence: the presence of the E2F consensus motif (8bp long) was analysed genome wide in *Arabidopsis* published by (Naouar *et al.*, 2009). We compared the published dataset of genes with E2F motif in their 1kb promoter region (Appendix Table S1, (Naouar *et al.*, 2009) to our differentially expressed genes.

2. Title: changed in the spirit of the referees' request, according to the Editor's suggestion (119 characters).

3. Now, cell-cycle regulation is also introduced (p3, first paragraph) and the function of CYCD3.1 is included (p6, second paragraph). Difference between *amiRBR* (constitutive artificial microRNA against RBR) and *rRBr* (meristem-specific antisense RNA) now explained in first paragraph of Results (p.6).

4. CenH3: We think it will turn out to be relevant for the readers that RBR foci partially co-localised with centromeric heterochromatin immunolabelled with CenH3. Similar co-localisation was shown in animal studies (Coschi *et al.*, 2014, Hinds, 2014). We kept the information on CenH3 and RBR

co-localisation very brief in the Results; just one sentence in the current version (p.8, first paragraph, last sentence).

5. The criterium for AtBRCA1 complementation, the cell death response, is now presented in a single main and EV figures together (Figs 4A and EV3D and E) and explicitly explained in the text (p.9, first paragraph).

6. RBR regulation and AtBRCA1 regulatory interactions are now explicitly clarified (p.12, last sentence of first paragraph).

7. We shortened the description and clarified that the presence of the marked box “domain” described for animals but not yet verified in plants.(p.13, second paragraph).

8. Previous text: “*Atbrca1-3* allele also showed hypersensitivity to MMC (Figure S4C-E) and partially suppressed the effect of RBR silencing (Figure 5F).” Citation of Figure 5F was a typing mistake. Now it is corrected and shown in Fig 6C and D.

9. The referee is right to notice that in the current Fig 9 the MMC induced cell death suppressed by SOG1 is possibly more extensive in the vasculature than the spontaneous cell death in *amiRBR*. It is difficult to interpret the developmental specificity of cell death in current time, and therefore we did not expand on this observation.

10. The same applies for the suppression of spontaneous cell death by *Atbrca1-1* in the *amiRBR* that might be more pronounced in the proximal meristem.

Referee # 2:

We have extensively rearranged the data and thus the figures, reorganised and edited the text especially in the ‘Results’ section to clarify the main message.

1.The legend and explanation of symbols of Table S1 (micro-array data) is now more detailed (Appendix Table S1)

2. Appendix Table S4 and S5: In response to previous referees’ comments we were asked to expand on E2F specificities. We did this by

(i). analysing *e2fa* and *e2fb* mutants: here we found that E2FA can be functionally linked to AtBRCA1 regulation and cell death,

(ii). by ChIP showing that E2FA binding to *AtBRCA1* promoter is increased upon MMC treatment,

(iii). by quantitative mass spec where we showed that E2FA interaction with RBR is increased upon MMC treatment (Appendix Table S4) and

(iv) by comparing protein complexes formed by E2FA compared to E2FB. Mass spec analysis of E2FA-GFP and E2FB-GFP pull downs showed that E2FA interacts with RBR and DPs, as expected; but is not part of the DREAM complex in contrast to E2FB and E2FC (Kobayashi *et al.*, 2015). All these data contribute to the understanding of E2F specificities in plants and the latter two strengthen our data related to the difference found between E2FA and E2FB in terms of AtBRCA1 regulation and cell death response. Hence, we are of opinion that the mass spec data support the message of the manuscript.

3. *rRBR* phenotypes are shown to reveal where and when the root specific silencing is/was effective in the quantification experiments. To follow cell division was especially important during the isolation of the root-tip material for the microarray experiment. At each biological replica, we used seedlings for the different parallel experiments, such as lugol staining, cell death and RNA isolation in order to follow the effect of *RBR* silencing in time and to synchronise the studies. We prefer to keep this information. The images are now shown in Fig EV1A-C.

4. The genes chosen for qRT-PCR validation of data were selected to represent well known DDR genes and one from other functional categories, as explained in the Supplementary Information, now Appendix Supplementary Methods, “Analysis of the differentially expressed genes”. Their identity is shown in the Figs 7 and S2 and Table S1. It is now explicitly mentioned that we validated the micro-array data both in the antisense RNA (*rRBr*) and artificial microRNA (*amiRBR*) lines shown within a single figure (p.12 first paragraph, Fig 7B,C and D, respectively).

5. We now provide a tentative conclusion for the *amiRBR,Atbrca1-1* strong phenotype “*indicating a variably penetrant window of sensitivity for the lack of AtBRCA1 and compromised RBR level during embryogenesis.*” (p.11. first paragraph).

6. We also explain explicitly how the enrichment of RBR on *AtBRCA1* promoter, on DNA segments flanking the 2 E2F motif-bearing fragment with (much higher) signal can either be due to sonication inhomogeneity or (less likely) cryptic E2F binding sites (p.13, first paragraph) and resulting similar enrichment as in the positive control.

References

Coschi CH, Ishak CA, Gallo D, Marshall A, Talluri S, Wang J, Cecchini MJ, Martens AL, Percy V, Welch I, Boutros PC, Brown GW, Dick FA (2014) Haploinsufficiency of an RB-E2F1-Condensin II complex leads to aberrant replication and aneuploidy. *Cancer Discov* 4: 840-53

Hinds PW (2014) A little pRB can lead to big problems. *Cancer Discov* 4: 764-5

Kobayashi K, Suzuki T, Iwata E, Nakamichi N, Suzuki T, Chen P, Ohtani M, Ishida T, Hosoya H, Muller S, Leviczky T, Pettko-Szandtner A, Darula Z, Iwamoto A, Nomoto M, Tada Y, Higashiyama T, Demura T, Doonan JH, Hauser MT et al. (2015) Transcriptional repression by MYB3R proteins regulates plant organ growth. *EMBO J*

Naouar N, Vandepoele K, Lammens T, Casneuf T, Zeller G, van Hummelen P, Weigel D, Ratsch G, Inze D, Kuiper M, De Veylder L, Vuylsteke M (2009) Quantitative RNA expression analysis with Affymetrix Tiling 1.0R arrays identifies new E2F target genes. *Plant J* 57: 184-94

3rd Editorial Decision

23 February 2017

Thank you for submitting your final revised manuscript and updated files for our consideration. I am pleased to inform you that we have now accepted it for publication in The EMBO Journal, and will be happy to coordinate its publication with the related paper by Biedermann et al.

Corresponding Author Name: Beatrix Horvath/ Ben Scheres

Journal Submitted to: The EMBO Journal

Manuscript Number: EMBOJ-2016-94561